# Convergence of denoising diffusion models under the manifold hypothesis

**Valentin De Bortoli**
*Department of Computer Science*
*ENS, CNRS, PSL University*
*Paris, France*

**Reviewed on OpenReview:** *https://openreview.net/forum?id=MhK5aXo3gB&*

## Abstract

Denoising diffusion models are a recent class of generative models exhibiting state-of-the-art performance in image and audio synthesis. Such models approximate the time-reversal of a forward noising process from a target distribution to a reference measure, which is usually Gaussian. Despite their strong empirical results, the theoretical analysis of such models remains limited. In particular, all current approaches crucially assume that the target density admits a density w.r.t. the Lebesgue measure. This does not cover settings where the target distribution is supported on a lower-dimensional manifold or is given by some empirical distribution. In this paper, we bridge this gap by providing the first convergence results for diffusion models in this setting. In particular, we provide quantitative bounds on the Wasserstein distance of order one between the target data distribution and the generative distribution of the diffusion model.

## 1 Introduction

Diffusion modeling, also called score-based generative modeling, is a new paradigm for generative modeling which exhibits state-of-the-art performance in image and audio synthesis (Song and Ermon, 2019; Song et al., 2021b; Ho et al., 2020; Nichol and Dhariwal, 2021; Dhariwal and Nichol, 2021). Such models first consider a forward stochastic process, adding noise to the data until a Gaussian distribution is reached. The model then approximates the backward process associated with this forward noising process. It can be shown, see (Haussmann and Pardoux, 1986) for instance, that in order to compute the drift of the backward trajectory, the gradient of the forward logarithmic density (Stein score) must be estimated. Such an estimator is then obtained using score matching techniques (Hyvärinen, 2005; Vincent, 2011) and leveraging neural network techniques. At sampling time, the backward process is initialized with a Gaussian and run backward in time using the approximation of the Stein score. Despite impressive empirical results, theoretical understanding and convergence analysis of diffusion models remain limited. De Bortoli et al. (2021b) establish the convergence of diffusion models in total variation under the assumption that the target distribution admits a density w.r.t. the Lebesgue measure and under dissipativity conditions. More recently Lee et al. (2022) obtained convergence results for diffusion models, including predictor-corrector schemes, under the assumption that the target distribution admits a density w.r.t. the Lebesgue measure and satisfies a log-Sobolev inequality.

However, these works implicitly assume that the score does not explode as $t \to 0$, by imposing that the score of the data distribution is Lipschitz continuous or satisfies some growth property. This is not observed in practice and experimentally the norm of the score blows up when $t \to 0$, see (Kim et al., 2022) for instance. Indeed, the assumptions that the target distribution admits a density w.r.t. the Lebesgue measure and has a Lipschitz logarithmic gradient does not hold if one assumes the *manifold hypothesis* (Tenenbaum et al., 2000; Fefferman et al., 2016; Goodfellow et al., 2016; Brown et al., 2022) or if the target measure is an empirical measure. In this setting, the target distribution is supported on a lower dimensional compact set. In the case of image processing, this hypothesis is supported by empirical evidence (Weinberger and Saul,

2006; Fefferman et al., 2016). Under this hypothesis, even though the forward process admits a density for all $t > 0$ its logarithmic gradient explodes for small $t \to 0$. Consequently, previous theoretical analyses of diffusion models do not apply to this setting. To our knowledge, (Pidstrigach, 2022) is the only existing work investigating the convergence of diffusion models under such manifold assumptions by showing that the limit of the continuous backward process with approximate score is well-defined and that its distribution is equivalent to the one of the target distribution under integrability conditions on the error of the score. In particular, Pidstrigach (2022) show that these distributions have the same support.

In this work, we complement these results and study the convergence rate of diffusion models under the manifold hypothesis. More precisely, we derive quantitative convergence bounds in Wasserstein distance of order one between the target distribution and the generative distribution of the diffusion model. The rest of the paper is organized as follows. In Section 2, we recall the basics of diffusion models. We present our main results and discuss links with the existing literature in Section 3. The rest of the paper is dedicated to the proof of Theorem 1 in Section 4. We conclude and explore future avenues in Section 5.

## 2   Diffusion models for generative modeling

In this section, we recall the basics of diffusion models. Henceforth, let $\pi \in \mathscr{P}(\mathbb{R}^d)$ denote the target distribution, also known as the data distribution, and $\pi_\infty = \mathrm{N}(0, \mathrm{Id})$ the $d$-dimensional Gaussian distribution with zero mean and identity covariance matrix. In what follows, we let $T > 0$ and consider the forward noising process $(\mathbf{X}_t)_{t \in [0,T]}$ given by an Ornstein–Uhlenbeck[1] process as follows

$$\mathrm{d}\mathbf{X}_t = -\beta_t \mathbf{X}_t \mathrm{d}t + \sqrt{2\beta_t} \mathrm{d}\mathbf{B}_t, \qquad \mathbf{X}_0 \sim \pi. \tag{1}$$

where $(\mathbf{B}_t)_{t \geq 0}$ is a $d$-dimensional Brownian motion and $t \mapsto \beta_t$ is a (positive) weight function. In practice, setting $\beta_0 \leq \beta_T$ allows for better control of the backward diffusion near the target distribution, see (Nichol and Dhariwal, 2021; Song et al., 2021b) for instance. In what follows, we assume that (1) admits a strong solution. Under mild assumptions on the target distribution (Haussmann and Pardoux, 1986; Cattiaux et al., 2021), the backward process $(\mathbf{Y}_t)_{t \in [0,T]} = (\mathbf{X}_{T-t})_{t \in [0,T]}$ satisfies

$$\mathrm{d}\mathbf{Y}_t = \beta_{T-t}\{\mathbf{Y}_t + 2\nabla \log p_{T-t}(\mathbf{Y}_t)\}\mathrm{d}t + \sqrt{2\beta_{T-t}}\mathrm{d}\mathbf{B}_t, \tag{2}$$

where $\{p_t\}_{t \in (0,T]}$ the family of densities of $\{\mathcal{L}(\mathbf{X}_t)\}_{t \in (0,T]}$ [2] w.r.t. the Lebesgue measure. In order to define (2) we do not need to assume that $\pi$ admits a density w.r.t. the Lebesgue measure. In practice, instead of sampling from $\mathbf{Y}_0 \sim \mathcal{L}(\mathbf{X}_T)$ we sample from $\mathbf{Y}_0 \sim \pi_\infty = \mathrm{N}(0, \mathrm{Id})$. For large $T > 0$ the mismatch between the distribution of $\mathbf{X}_T$ and $\pi_\infty$ is small due to geometric convergence of the Ornstein–Uhlenbeck process.

In practice, $\{\nabla \log p_t\}_{t \in [0,T]}$ cannot be computed exactly and is approximated by a family of estimators $\{\boldsymbol{s}(t, \cdot)\}_{t \in [0,T]}$. Those estimators minimize the denoising score matching loss function $\ell$ given by

$$\ell(\boldsymbol{s}) = \int_0^T \phi(t) \mathbb{E}[\|\boldsymbol{s}(t, \mathbf{X}_t) - \nabla \log p_{t|0}(\mathbf{X}_t|\mathbf{X}_0)\|^2]\mathrm{d}t, \tag{3}$$

with $p_{t|0}$ is the density of $\mathbf{X}_t$ given $\mathbf{X}_0$, $i.e.$ the density of the transition kernel associated with (1) and $\phi : [0, T] \to \mathbb{R}_+$ is a weighting function. In practice, (3) is approximated using Monte Carlo samples and the loss function is minimized over the parameters of a neural network.

Once the score estimator $\boldsymbol{s}$ is learned, we introduce a continuous-time backward process $(\hat{\mathbf{Y}}_t)_{t \in [0,T]}$ approximating $(\mathbf{Y}_t)_{t \in [0,T]}$ and given by

$$\mathrm{d}\hat{\mathbf{Y}}_t = \beta_{T-t}\{\hat{\mathbf{Y}}_t + 2\boldsymbol{s}(T - t, \hat{\mathbf{Y}}_t)\}\mathrm{d}t + \sqrt{2\beta_{T-t}}\mathrm{d}\mathbf{B}_t, \qquad \hat{\mathbf{Y}}_0 \sim \pi_\infty = \mathrm{N}(0, \mathrm{Id}). \tag{4}$$

In practice, one needs to discretize (4) in order to define an algorithm which can be implemented. We consider a sequence of stepsizes $\{\gamma_k\}_{k \in \{0,\dots,K\}}$ such that $\sum_{k=0}^K \gamma_k = T$. In what follows, for any $k \in \{0,\dots,K\}$ we

---

[1]Also called Variance Preserving Stochastic Differential Equation (VPSDE) in Song et al. (2021b).
[2]For any $\mathbb{R}^d$-valued random variable $X$, $\mathcal{L}(X)$ is the distribution of $X$.

denote $t_{k+1} = \sum_{j=0}^{k} \gamma_j$ and $t_0 = 0^3$. Given this sequence of stepsizes, we consider the interpolation process $(\bar{\mathbf{Y}}_t)_{t \in [0,T]}$ defined for any $k \in \{0, \ldots, K\}$ and $t \in [t_k, t_{k+1}]$ by

$$d\bar{\mathbf{Y}}_t = \beta_{T-t}\{\bar{\mathbf{Y}}_t + 2\boldsymbol{s}(T - t_k, \bar{\mathbf{Y}}_{t_k})\}dt + \sqrt{2\beta_{T-t}}d\mathbf{B}_t, \qquad \bar{\mathbf{Y}}_0 \sim \pi_\infty.$$

This process is an Ornstein–Uhlenbeck process on the interval $[t_k, t_{k+1}]$. Denoting $(Y_k)_{k \in \{0, \ldots, K+1\}}$ such that for any $k \in \{0, \ldots, K+1\}$, $Y_k = \bar{\mathbf{Y}}_{t_k}$, we have for any $k \in \{0, \ldots, K\}$

$$Y_{k+1} = Y_k + \gamma_{1,k}(Y_k + 2\boldsymbol{s}(T - t_k, Y_k)) + \sqrt{2\gamma_{2,k}}Z_k, \tag{5}$$

$$\gamma_{1,k} = \exp[\int_{T-t_{k+1}}^{T-t_k} \beta_s ds] - 1, \qquad \gamma_{2,k} = (\exp[2\int_{T-t_{k+1}}^{T-t_k} \beta_s ds] - 1)/2.$$

where $\{Z_k\}_{k \in \mathbb{N}}$ is a sequence of independent $d$-dimensional Gaussian random variables with zero mean and identity covariance matrix. The discretization (5) approximately corresponds to the discrete-time scheme introduced in (Ho et al., 2020), see Appendix B.2. We call this discretization scheme the *exponential integrator* (EI) discretization, similarly to Zhang and Chen (2022) who introduced a similar scheme in accelerated deterministic diffusion models. Lee et al. (2022) analyze a slightly different scheme corresponding to replacing $\beta_{T-t}\bar{\mathbf{Y}}_t$ by $\beta_{T-t}\bar{\mathbf{Y}}_{t_k}$ in (4). We summarize the processes we have introduced in Table 1 and discuss the links between (5) and the classical Euler–Maruyama discretization in Appendix B.1. As emphasized in the

| Description | Evolution equation |
|---|---|
| Forward process | $d\mathbf{X}_t = -\beta_t \mathbf{X}_t dt + \sqrt{2\beta_t}d\mathbf{B}_t$ |
| Backward process (BP) | $d\mathbf{Y}_t = \beta_{T-t}\{\mathbf{Y}_t + 2\nabla \log p_{T-t}(\mathbf{Y}_t)\}dt + \sqrt{2\beta_{T-t}}d\mathbf{B}_t$ |
| Score approximate BP (SBP) | $d\hat{\mathbf{Y}}_t = \beta_{T-t}\{\hat{\mathbf{Y}}_t + 2\boldsymbol{s}(T - t, \hat{\mathbf{Y}}_t)\}dt + \sqrt{2\beta_{T-t}}d\mathbf{B}_t$ |
| EI interpolation of SBP | $d\bar{\mathbf{Y}}_t = \beta_{T-t}\{\bar{\mathbf{Y}}_t + 2\boldsymbol{s}(T - t_k, \bar{\mathbf{Y}}_{t_k})\}dt + \sqrt{2\beta_{T-t}}d\mathbf{B}_t$ |
| EI discretization of SBP | $Y_{k+1} = Y_k + \gamma_{1,k}(Y_k + 2\boldsymbol{s}(T - t_k, Y_k)) + \sqrt{2\gamma_{2,k}}Z_k$ |

Table 1: Different processes considered in this paper.

introduction, under the manifold hypothesis or in the case where the target distribution is an empirical measure, the true score $\nabla \log p_t$ explodes when $t \to 0$. This behavior has been observed in practice for image synthesis (Kim et al., 2022; Song and Ermon, 2020). One way to deal with this explosive behavior is to truncate the integration of the backward diffusion, *i.e.* instead of running $(\mathbf{Y}_t)_{t \in [0,T]}$ we consider $(\mathbf{Y}_t)_{t \in [0, T-\varepsilon]}$ for a small hyperparameter $\varepsilon > 0$, (Vahdat et al., 2021; Song and Ermon, 2020). Translating this condition on the associated discretized process, we assume that $t_K = T - \varepsilon$ and study $\{Y_k\}_{k \in \{0, \ldots, K\}}$ by disregarding the last sample $Y_{K+1}$. We note that versions of diffusion models defined in discrete time do not suffer from such shortcomings as the truncation is embedded in the discretization scheme, see (Song et al., 2021b; Song and Ermon, 2020; 2019; Ho et al., 2020) for instance. Recently Kim et al. (2022) have proposed a soft probabilistic truncation to replace the proposed hard threshold.

## 3 Main results

We first start by introducing and discussing our main assumptions. The only assumption we consider on the data distribution $\pi$ is that it is supported on a compact set $\mathcal{M} \subset \mathbb{R}^d$ (*i.e.* a bounded and closed subset of $\mathbb{R}^d$).

**A1.** *$\pi$ is supported on a compact set $\mathcal{M}$ and $0 \in \mathcal{M}$.*

The assumption $0 \in \mathcal{M}$ can be omitted but is kept to simplify the proofs. We denote $\mathrm{diam}(\mathcal{M})$ the diameter of the manifold defined by $\mathrm{diam}(\mathcal{M}) = \sup\{\|x - y\| : x, y \in \mathcal{M}\}$.

An assumption of compactness is natural in image processing as images are encoded on a finite range (typically $[0, 255]$ for each channel). We emphasize that this assumption encompasses not only all distributions which admit a continuous density on a lower dimensional manifold but also all empirical densities of the form $(1/N)\sum_{i=1}^{N} \delta_{X^i}$. Next, we turn to the temperature schedule $t \mapsto \beta_t$ and make the following assumption.

---

$^3$Note that $t_{K+1} = T$.

**A2.** $t \mapsto \beta_t$ *is continuous, non-decreasing and there exists* $\bar{\beta} > 0$ *such that for any* $t \in [0, T]$, $1/\bar{\beta} \leq \beta_t \leq \bar{\beta}$.

Under this assumption, the integral of $t \mapsto \beta_t$ is well-defined and for any $t \in [0, T]$ we have that

$$\mathbf{X}_t = m_t \mathbf{X}_0 + \sigma_t Z, \qquad m_t = \exp[-\int_0^t \beta_s \mathrm{d}s], \qquad \sigma_t^2 = 1 - \exp[-2\int_0^t \beta_s \mathrm{d}s],$$

where the first equality holds in distribution and $Z$ is a Gaussian random variable with zero mean and identity covariance. Note that **A2** is satisfied for every schedule used in practice, see Appendix G. Finally, we make the following assumption on the score network.

**A3.** *There exist* $\boldsymbol{s} \in \mathrm{C}([0, T] \times \mathbb{R}^d, \mathbb{R}^d)$ *and* $\mathtt{M} \geq 0$ *such that for any* $t \in [0, T]$ *and* $x_t \in \mathbb{R}^d$,

$$\|\boldsymbol{s}(t, x_t) - \nabla \log p_t(x_t)\| \leq \mathtt{M}(1 + \|x_t\|)/\sigma_t^2.$$

Contrary to De Bortoli et al. (2021b), we do not assume a uniform bound in time and space as we allow growth as $t \to 0$ and $\|x\| \to 0$. This assumption is more realistic as $\|\nabla \log p_t(x_t)\| \sim_{t \to 0} c_0(x_t)/\sigma_t^2$ and $\|\nabla \log p_t(x_t)\| \sim_{\|x_t\| \to +\infty} c_1(t)\|x_t\|$ as we will show in Appendix C. This explosive behavior as $t \to 0$ is accounted for in practical implementations. For example Song et al. (2021b) used a parameterization of the score of the form $\boldsymbol{s}(t, x) = \boldsymbol{n}(t, x)/\sigma_t$, where $\boldsymbol{n}$ is a neural network with learnable parameters. Our assumption is notably different from the one of Lee et al. (2022) which assume a uniform in time $\mathrm{L}^2$ bound between the score estimator and the true score. Nevertheless, in Appendix I we derive Theorem I.1 which is the counterpart to our main result under a $\mathrm{L}^2$ error assumption, using the theory of Lee et al. (2022) to derive an $\mathrm{L}^\infty$ error from a $\mathrm{L}^2$ one. However our $\mathrm{L}^2$ error bounds are weaker than the ones of Lee et al. (2022) as they are estimated w.r.t. to the distribution of the *algorithm* and not w.r.t. the true backward distribution. We highlight that $\mathrm{L}^2$ bounds are more realistic than $\mathrm{L}^\infty$ as the score is estimated on the data.

Finally, we make the following assumption on the sequence of stepsizes. Recall that for any $k \in \{0, \ldots, N\}$ we have $t_{k+1} = \sum_{j=0}^k \gamma_j$ and $t_0 = 0$.

**A4.** *For any* $k \in \{0, \ldots, K-1\}$, *we have* $\gamma_k \sup_{v \in [T-t_{k+1}, T-t_k]} \beta_v/\sigma_v^2 \leq \delta \leq 1/2$.

In the case where $\beta_t = \beta_0$ for any $t \in [0, T]$, **A4** is implied by the following condition: for any $k \in \{0, \ldots, K-1\}$

$$\gamma_k(\beta_0 + (2\sum_{j=k+1}^K \gamma_j)^{-1}) \leq \delta. \tag{6}$$

In the next section, we fix $\gamma_K = \varepsilon$ and in this case, the condition (6) is satisfied if $\gamma_k \leq \delta\varepsilon/(2 + \beta_0\varepsilon)$.

### 3.1 Convergence bounds

We are now ready to state our main result.

**Theorem 1.** *Assume* **A1**, **A2**, **A3**, **A4** *that* $T \geq 2\bar{\beta}(1 + \log(1 + \mathrm{diam}(\mathcal{M})))$, $\gamma_K = \varepsilon$ *and* $\varepsilon, \mathtt{M}, \delta \leq 1/32$. *Then, there exists* $\mathtt{D}_0 \geq 0$ *such that*

$$\mathbf{W}_1(\mathcal{L}(Y_K), \pi) \leq \mathtt{D}_0(\exp[\kappa/\varepsilon](\mathtt{M} + \delta^{1/2})/\varepsilon^2 + \exp[\kappa/\varepsilon]\exp[-T/\bar{\beta}] + \varepsilon^{1/2}),$$

*with* $\kappa = \mathrm{diam}(\mathcal{M})^2(1 + \bar{\beta})/2$ *and*

$$\mathtt{D}_0 = D(1 + \bar{\beta})^7(1 + d + \mathrm{diam}(\mathcal{M})^4)(1 + \log(1 + \mathrm{diam}(\mathcal{M}))), \tag{7}$$

*and* $D$ *is a numerical constant.*

First, we note that letting $T \to +\infty$, $\delta, \mathtt{M} \to 0$ and then $\varepsilon \to 0$ we get that $\mathbf{W}_1(\mathcal{L}(Y_K), \pi) \to 0$. This consequence is to be expected since $\lim_{\varepsilon \to 0} \mathcal{L}(\mathbf{Y}_{T-\varepsilon}) = \pi$. An explicit dependency of the bound on these parameters is given in Corollary 2. More generally the error bound depends on four variables (a) $\varepsilon$ which corresponds to the truncation of the backward process, (b) $T$ the integration time of the forward process, (c) $\delta$ which is related to a condition on the stepsizes of the backward discretization, see **A4** (d) $\mathtt{M}$ which controls the score approximation, see **A3**. The dependence w.r.t. $\delta^{1/2}$ and $\mathtt{M}$ is linear, whereas the dependence w.r.t. $T$ is of the form $\exp[-T/\bar{\beta}]$. These two terms are multiplied by a quantity depending on the truncation

bound $\varepsilon$ which is exponential of the form $\exp[\kappa/\varepsilon]$. We conjecture that under additional assumptions on $\mathcal{M}$ this dependence can be improved to also be polynomial, see Theorem 3 for an extension of Theorem 1 under general Hessian assumptions. Additional remarks and comments on Theorem 1 and its assumptions are considered in Appendix F.

*Proof.* We provide a sketch of the proof. The detailed proof is postponed to Section 4. The distribution of $Y_K$ is given by $\pi_\infty \mathrm{R}_K$, where $\mathrm{R}_K$ is the transition kernel associated with $Y_K|Y_0$. In order to control $\mathbf{W}_1(\pi_\infty \mathrm{R}_K, \pi)$, we consider the following inequality

$$\mathbf{W}_1(\pi_\infty \mathrm{R}_K, \pi) \leq \mathbf{W}_1(\pi_\infty \mathrm{R}_K, \pi_\infty \mathrm{Q}_{t_K}) + \mathbf{W}_1(\pi_\infty \mathrm{Q}_{t_K}, \pi \mathrm{P}_{T-t_K}) + \mathbf{W}_1(\pi \mathrm{P}_{T-t_K}, \pi), \tag{8}$$

where $(\mathrm{P}_t)_{t \in [0,T]}$ is the semi-group associated with $(\mathbf{X}_t)_{t \in [0,T]}$ and $(\mathrm{Q}_t)_{t \in [0,T]}$ is the semi-group associated with $(\mathbf{Y}_t)_{t \in [0,T]}$. We then control each one of these terms. The first term corresponds to the discretization error and the score approximation. It is upper bounded by a term of the form $\mathcal{O}(\exp[\kappa/\varepsilon](\mathtt{M} + \delta^{1/2})/\varepsilon^2)$. The second term corresponds to the convergence of the continuous-time exact backward process and is of order $\mathcal{O}(\exp[\kappa/\varepsilon]\exp[-T/\bar{\beta}])$. The last term corresponds to the error between the data distribution and a slightly noisy version of this distribution and is of order $\mathcal{O}(\varepsilon^{1/2})$. $\qquad\square$

As an immediate corollary of Theorem 1, we have the following result.

**Corollary 2.** *Assume* **A**1, **A**2, **A**3, **A**4. *Let* $\eta \in (0, 1/32)$, $T \geq 2\bar{\beta}(1 + \log(1 + \mathrm{diam}(\mathcal{M})))$ *and*

$$T \geq \bar{\beta}(\kappa + 1)/\eta^2, \quad \mathtt{M} \leq \exp[-\kappa/\eta^2]\eta^5, \quad \delta \leq \exp[-2\kappa/\eta^2]\eta^{10}, \quad \gamma_K = \eta^2.$$

*Then,*

$$\mathbf{W}_1(\mathcal{L}(Y_K), \pi) \leq 4\mathtt{D}_0\eta,$$

*with* $\kappa = \mathrm{diam}(\mathcal{M})^2(1 + \bar{\beta})/2$ *and* $\mathtt{D}_0$ *given in* (7).

The constant $\mathtt{D}_0$ appearing in Theorem 1 and Corollary 2 does not depend on $\varepsilon$, $T$, $\delta$ and $\mathtt{M}$ but only on $\bar{\beta}$, $\mathrm{diam}(\mathcal{M})$ and $d$. In particular, we highlight that the dependence of $\mathtt{D}_0$ w.r.t. the dimension is $\mathcal{O}(d)$ and the dependence w.r.t. the diameter of $\mathcal{M}$ is $\mathcal{O}(\mathrm{diam}(\mathcal{M})^4)$ up to logarithmic term. Note that the diameter might only depend on *intrisic* dimension $p$ of $\mathcal{M}$ which satisfies $p \ll d$ in some settings. For example in the case of an hypercube of dimension $p$ we have $\mathrm{diam}(\mathcal{M}) = \sqrt{p}$.

Contrary to De Bortoli et al. (2021b); Lee et al. (2022); Pidstrigach (2022), our results are stated w.r.t. the Wasserstein distance and not the total variation distance or the Kullback-Leibler divergence. We emphasize that studying the total variation or Kullback-Leibler divergence between the distribution of $Y_K$ and the one of $\pi$ under **A**1 with $\mathcal{M}$ lower dimensional than $\mathbb{R}^d$ lead to *vacuous bounds* as these quantities are lower bounded by 1 in the case of the total variation and $+\infty$ in the case of the Kullback-Leibler divergence since the densities we are comparing are not supported on the same set.[4] This is not the case with the Wasserstein distance of order one. To the best of our knowledge Theorem 1 is the first convergence result for diffusion models w.r.t. $\mathbf{W}_1$. We note that our result could be extended to $\mathbf{W}_p$ for any $p \geq 1$, since we do not rely on any property specific to $\mathbf{W}_1$ among all $\mathbf{W}_p$ distances for any $p \geq 1$. In particular, our analysis does not use the fact that $\mathbf{W}_1$ is an integral probability metric, (Sriperumbudur et al., 2009).

We conclude this section, with an improvement upon Theorem 1 in the case where tighter bounds on the Hessian $\nabla^2 \log p_t$ are available.

**Theorem 3.** *Assume* **A**1, **A**2, **A**3, **A**4 *that* $T \geq 2\bar{\beta}(1 + \log(1 + \mathrm{diam}(\mathcal{M})))$, $\gamma_K = \varepsilon$ *and* $\varepsilon, \mathtt{M}, \delta \leq 1/32$. *In addition, assume that there exists* $\Gamma \geq 0$ *such that for any* $t \in (0, T]$ *and* $x_t \in \mathbb{R}^d$

$$\|\nabla^2 \log p_t(x_t)\| \leq \Gamma/\sigma_t^2. \tag{9}$$

*Then, there exists* $\mathtt{D}_0 \geq 0$ *such that*

$$\mathbf{W}_1(\mathcal{L}(Y_K), \pi) \leq \mathtt{D}_0((\mathtt{M} + \delta^{1/2})/\varepsilon^{\Gamma+2} + \exp[-T/\bar{\beta}]/\varepsilon^\Gamma + \varepsilon^{1/2}),$$

---

[4]We emphasize however that total variation bounds smaller than 1 and finite Kullback-Leibler divergence have strong implications, namely the generative model has same support as the target distribution. However, such property is not satisfied in practice, see Appendix F or (Jolicoeur-Martineau et al., 2021, Figure 2) for instance.

*with*

$$\mathtt{D}_0 = D(1 + d + (1 + \mathrm{diam}(\mathcal{M}))^4) \exp[3(1 + \bar{\beta})^2 (\Gamma + 2)(1 + \log(1 + \mathrm{diam}(\mathcal{M})))].$$

*and $D$ is a numerical constant.*

*Proof.* The complete proof is postponed to Appendix J.1. The crux of the proof is to derive an improved version of Proposition 6 which provides controls on some tangent process. Indeed, in Proposition 6, we use an upper bound of the form $\|\nabla^2 \log p_t(x_t)\| \leq \Gamma/\sigma_t^4$ which is a loose upper bound derived under **A**1. $\qquad\square$

Theorem 3 improves the bounds of Theorem 1, since the *exponential* dependency w.r.t. $\varepsilon$ is replaced by a *polynomial* dependency with exponent $\Gamma$. At first sight, it is not clear when (9) is satisfied. However, in special cases we can verify this condition explicitly. For example, in Appendix J.2, we show that this condition is satisfied if $\pi$ is the uniform distribution on the hypercube, with $p \in \{1, \dots, d\}$. The condition (9) has strong geometrical implications on $\mathcal{M}$. In particular, under appropriate smoothness assumptions on $\mathcal{M}$, it implies that $\mathcal{M}$ is convex, Appendix J.3.

### 3.2 Statistical guarantees and empirical measure targets

We emphasize that the results of Theorem 1 hold under the general assumption **A**1 which only requires the target measure to be supported on a compact set. This includes measures which are supported on a smooth manifold of dimension $p \leq d$ but also all empirical measures of the form $(1/N) \sum_{i=1}^{N} \delta_{X^i}$ with $\{X^i\}_{i=1}^{N} \sim \pi^{\otimes N}$. In particular if we assume that the underlying target measure $\pi$ is supported on a manifold of dimension $p \leq d$ and that the diffusion models are trained w.r.t. some empirical measure associated with $\pi$ then we have the following result.

**Proposition 4.** *Assume **A**1, **A**2, **A**3, **A**4 that $T \geq 2\bar{\beta}(1 + \log(1 + \mathrm{diam}(\mathcal{M})))$, $\gamma_K = \varepsilon$ and $\varepsilon, \mathtt{M}, \delta \leq 1/32$. Then, for any $\eta > 0$ there exist $\mathtt{D}_0, \mathtt{D}_1 \geq 0$ such that*

$$\mathbb{E}[\mathbf{W}_1(\mathcal{L}(Y_K), \pi)] \leq \mathtt{D}_0(\exp[\kappa/\varepsilon](\mathtt{M} + \delta^{1/2})/\varepsilon^2 + \exp[\kappa/\varepsilon]\exp[-T/\bar{\beta}] + \varepsilon^{1/2}) + \mathtt{D}_1 N^{-1/(d_M(\mathcal{M})+\eta)},$$

*with $d_M(\mathcal{M})$ the Minkowski dimension of $\mathcal{M}$, see (11), $\kappa = \mathrm{diam}(\mathcal{M})^2(1+\bar{\beta})/2$, $\mathtt{D}_1$ given in (Weed and Bach, 2019, Theorem 1) and*

$$\mathtt{D}_0 = D(1 + \bar{\beta})^7(1 + d + \mathrm{diam}(\mathcal{M})^4)(1 + \log(1 + \mathrm{diam}(\mathcal{M}))),$$

*with $D$ a numerical constant.*

*Proof.* For any $N \in \mathbb{N}$, we denote $\pi^N = (1/N) \sum_{i=1}^{N} \delta_{X^i}$. Using Theorem 1, we have that for any $N \in \mathbb{N}$

$$\mathbf{W}_1(\mathcal{L}(Y_K), \pi^N) \leq \mathtt{D}_0(\exp[\kappa/\varepsilon](\mathtt{M} + \delta^{1/2})/\varepsilon^2 + \exp[\kappa/\varepsilon]\exp[-T/\bar{\beta}] + \varepsilon),$$

with a constant $\mathtt{D}_0$ which does not depend on $\{X^i\}_{i=1}^{N}$ and $N$. Therefore, we have that for any $N \in \mathbb{N}$

$$\mathbb{E}[\mathbf{W}_1(\mathcal{L}(Y_K), \pi^N)] \leq \mathtt{D}_0(\exp[\kappa/\varepsilon](\mathtt{M} + \delta^{1/2})/\varepsilon^2 + \exp[\kappa/\varepsilon]\exp[-T/\bar{\beta}] + \varepsilon). \tag{10}$$

Using (Weed and Bach, 2019, Theorem 1) and (Weed and Bach, 2019, Proposition 2), for any $\eta > 0$, there exists $\mathtt{D}_1 \geq 0$ such that

$$\mathbb{E}[\mathbf{W}_1(\pi^N, \pi)] \leq \mathtt{D}_1 N^{-1/(d_M(\mathcal{M})+\eta)},$$

which concludes the proof upon combining this result, (10) and the triangle inequality. $\qquad\square$

The Minkowski dimension $d(\mathcal{M})$ is defined as follows:

$$d(\mathcal{M}) = d - \liminf_{\varepsilon \to 0} \log(\mathrm{Vol}(\mathcal{M}_\varepsilon))/\log(1/\varepsilon), \tag{11}$$

with $\mathrm{Vol}(\mathsf{A})$ the volume of a (measurable) set $\mathsf{A}$ and $\mathcal{M}_\varepsilon$ the $\varepsilon$-fattening of $\mathcal{M}$, *i.e.* for any $\varepsilon > 0$, $\mathcal{M}_\varepsilon = \{x \in \mathbb{R}^d : d(x, \mathcal{M}) \leq \varepsilon\}$. For example if $\mathcal{M}$ is a topological manifold of dimension $p \leq d$ then its Minkowski dimension is $p$, *i.e.* $d_M(\mathcal{M}) = p$. Hence, in this case the error term in Proposition 4 depends *exponentially* on

the dimension of $\mathcal{M}$ and on its diameter but depends only *linearly* on $d$, the dimension of the ambient space. Note again that $\mathrm{diam}(\mathcal{M})$ might depend on the dimension of $\mathcal{M}$. For example in the case of the hypercube $\mathcal{M} = [-1/2, 1/2]^p$, we have $\mathrm{diam}(\mathcal{M}) = \sqrt{p}$. Hence, the results of Proposition 4 show that diffusion models exploit the lower-dimensional structure of the target. We highlight that this result does not quantify the *diversity* of diffusion models, *i.e.* their ability to produce samples which are distinct from the ones of the training dataset Alaa et al. (2022); Zhao et al. (2018). There is empirical evidence that denoising diffusion models yield generative models with good diversity properties Xiao et al. (2021); Dhariwal and Nichol (2021) and we leave the theoretical study of the diversity of denoising diffusion models for future work.

### 3.3 Related works

To the best of our knowledge, (De Bortoli et al., 2021b) is the first quantitative convergence results for denoising diffusion models. More precisely, De Bortoli et al. (2021b) show a bound in total variation between the distribution of the diffusion model and the target distribution of the form

$$\|\mathcal{L}(Y_{K+1}) - \pi\|_{\mathrm{TV}} \leq A(\exp[-T] + \exp[T](\mathtt{M}^{1/2} + \delta^{1/2})). \tag{12}$$

This result holds under the assumption that $\pi$ admits a density w.r.t. Lebesgue measure which satisfies some dissipativity conditions. Again we emphasize that such results in total variation are vacuous under the manifold hypothesis. The upper bound in (12) is obtained using a similar splitting of the error as in Theorem 1. However the control of the discretization error is handled using Girsanov formula in (De Bortoli et al., 2021b) and relies on similar techniques as (Dalalyan, 2017; Durmus and Moulines, 2017). In the present work, this error is controlled using the interpolation formula from Del Moral and Singh (2019) which, combined with controls on stochastic flows, allows for tighter controls of the discretization error w.r.t. $\mathbf{W}_1$.

Lee et al. (2022) study the convergence of diffusion models under (uniform in time) $\mathrm{L}^2$ controls on the score approximation. Their result is given w.r.t. the total variation and therefore suffers from the same shortcoming as the ones of De Bortoli et al. (2021b). In particular it is assumed that the data distribution admits a density w.r.t. the Lebesgue measure which satisfies some regularity conditions as well as a logarithmic Sobolev inequality. Additionally, it is required that $\nabla^2 \log p_t$ is bounded uniformly in time and in space which is not true under the manifold hypothesis and is hard to verify in practice even in simple cases.

Closer to our line of work are the results of Pidstrigach (2022) who proves that the approximate backward process (4) converges to a random variable whose distribution is supported on the manifold of interest. In this work, we complement these results by studying the discretization scheme and providing quantitative bounds between the output of the diffusion model and the target distribution.

Related to the manifold hypothesis and the study of convergence of diffusion models De Bortoli et al. (2022) study the convergence of a Riemannian counterpart of diffusion models. Result are given w.r.t. the total variation (defined on the manifold of interest). Even though such diffusion models directly incorporate the manifold information they require the knowledge of the geodesics and the Riemannian metric of the manifold. In the case of the manifold hypothesis these quantities are not known and therefore cannot be used in practice. In particular, De Bortoli et al. (2022) focus on manifolds which have a well-known structure such as $\mathbb{S}^1$, $\mathbb{T}^2$ or $\mathrm{SO}_3(\mathbb{R})$.

Franzese et al. (2022) show that there exists a trade-off between long and short time horizons $T$. Their analysis is based on a rearrangement of the Evidence Lower Bound (ELBO) obtained by Huang et al. (2021). This ELBO can be decomposed in the sum of two terms: one which decreases with $T$ (controlling the bias between $\mathcal{L}(\mathbf{X}_T)$ and $\pi_\infty$) and one which increases with $T$ (corresponding to the loss term (3)). Their decomposition of the ELBO is in fact equivalent to (Song et al., 2021a, Theorem 1). In Appendix H we include a short derivation of this result.

Finally, we highlight the earlier results of Block et al. (2020a). In this work, the authors study a version of the Langevin algorithm in which the score term is approximated. This is different from the diffusion model [5] setting and is closer to the setting of Plug-and-Play (PnP) approaches (Venkatakrishnan et al., 2013; Arridge

---

[5]Even though the authors provide a discussion on an annealed version of the algorithm they study which corresponds to the original framework of Song and Ermon (2019).

et al., 2019; Zhang et al., 2017). Related to our manifold assumptions, Block et al. (2020b) show that in a setting similar to PnP approaches, the corresponding Langevin dynamics enjoys fast convergence rates if the target distribution is supported on a manifold with curvature assumptions. In particular, they show that a noisy version of the target distribution satisfies a logarithmic Sobolev inequality with constant which only depends on the intrisic dimension of the manifold.

## 4  Proof of Theorem 1

In this section, we present a proof of Theorem 1. More precisely, we control each term on the right hand side of (8). The bottleneck of the proof resides in the control of the discretization and approximation error $\mathbf{W}_1(\pi_\infty R_K, \pi_\infty Q_{t_K})$ which is dealt with in Section 4.1. Then, we turn to the convergence of the backward process $\mathbf{W}_1(\pi_\infty Q_{t_K}, \pi P_{T-t_K})$ in Section 4.2. Finally, we control the noising error $\mathbf{W}_1(\pi P_{T-t_K}, \pi)$ and conclude in Section 4.3. Technical results are postponed to the appendix.

### 4.1  Control of $\mathbf{W}_1(\pi_\infty R_K, \pi_\infty Q_{t_K})$

In this section, we control $\mathbf{W}_1(\pi_\infty R_K, \pi_\infty Q_{t_K})$. To do so we are going to use the backward formula introduced in Del Moral and Singh (2019). First, we recall the definition of the stochastic flows $(\mathbf{Y}_{s,t}^x)_{s,t \in [0,T]}$ and the interpolation of its discretization $(\bar{\mathbf{Y}}_{s,t}^x)_{s,t \in [0,T]}$, for any $x \in \mathbb{R}^d$ and $s, t \in [0, T]$ with $t \geq s$

$$d\mathbf{Y}_{s,t}^x = \beta_{T-t}\{\mathbf{Y}_{s,t}^x + 2\nabla \log p_{T-t}(\mathbf{Y}_{s,t}^x)\}dt + \sqrt{2\beta_{T-t}}d\mathbf{B}_t, \qquad \mathbf{Y}_{s,s}^x = x,$$

and for any $k \in \{0, \ldots, K\}$ and $t \in [s_k, t_{k+1})$

$$d\bar{\mathbf{Y}}_{s,t}^x = \beta_{T-t}\{\bar{\mathbf{Y}}_{s,t}^x + 2\boldsymbol{s}(T - s_k, \bar{\mathbf{Y}}_{s,s_k}^x)\}dt + \sqrt{2\beta_{T-t}}d\mathbf{B}_t, \qquad \bar{\mathbf{Y}}_{s,s}^x = x,$$

where $s_k = \max(s, t_k)$. We also introduce the tangent process $(\mathbf{Y}_{s,t}^x)_{t \in [s,T]}$

$$d\nabla\mathbf{Y}_{s,t}^x = \beta_{T-t}\{\mathrm{Id} + 2\nabla^2 \log p_{T-t}(\mathbf{Y}_{s,t}^x)\}\nabla\mathbf{Y}_{s,t}^x dt, \qquad \nabla\mathbf{Y}_{s,s}^x = \mathrm{Id}. \tag{13}$$

Note that $(\mathbf{Y}_{s,t}^x)_{t \in [s,T]}$ is a $d \times d$ stochastic process. The tangent process $(\nabla\mathbf{Y}_{s,t}^x)_{s,t \in [0,T]}$ can also be defined as follows. Under mild regularity assumption, for any $s, t \in [0, T]$ with $t \geq s$, $x \mapsto \mathbf{Y}_{s,t}^x$ is a diffeomorphism, see (Kunita, 1981), and we denote $x \mapsto \nabla\mathbf{Y}_{s,t}^x$ its differential. Then, (Kunita, 1981, Section 2) shows under mild assumptions that $(\nabla\mathbf{Y}_{s,t}^x)_{s,t \in [0,T]}$ satisfies (13). Hence, $(\nabla\mathbf{Y}_{s,t}^x)_{s,t \in [0,T]}$ encodes the local variation of the process $(\mathbf{Y}_{s,t}^x)_{s,t \in [0,T]}$ w.r.t. its initial condition. Our bound on the approximation/discretization error relies on the following proposition which was first proven by Del Moral and Singh (2019).

**Proposition 5.** *Assume* **A**1*. Then, for any $s, t \in [0, T)$ with $s < t$ and $x \in \mathbb{R}^d$*

$$\mathbf{Y}_{s,t}^x - \bar{\mathbf{Y}}_{s,t}^x = \int_s^t (\nabla\mathbf{Y}_{u,t}^{\bar{\mathbf{Y}}_{s,u}^x})^\top \Delta b_u((\bar{\mathbf{Y}}_{s,v}^x)_{v \in [s,T]})du,$$

*where for any $u \in [0, T)$ with $u \in [s_k, t_{k+1})$ for some $k \in \{0, \ldots, K\}$ and $(\omega_v)_{v \in [s,T]} \in \mathrm{C}([s,T], \mathbb{R}^d)$ we have*

$$b_u(\omega) = \beta_{T-u}(\omega_u + 2\nabla \log p_{T-u}(\omega_u)), \quad \bar{b}_u(\omega) = \beta_{T-u}(\omega_u + 2\boldsymbol{s}(T - s_k, \omega_{s_k})), \quad \Delta b_u(\omega) = b_u(\omega) - \bar{b}_u(\omega).$$

*where $s_k = \max(s, t_k)$.*

*Proof.* The proof of this proposition is postponed to Appendix E. $\qquad\square$

Using Proposition 5 our goal is now to control $\|\nabla\mathbf{Y}_{s,t}^x\|$ and $\|\Delta b_s((\bar{\mathbf{Y}}_{s,t}^x)_{t \in [s,T]})\|$ for any $s, t \in [0, T]$ and $x \in \mathbb{R}^d$. To do so, we introduce the time $t^\star$ which is a lower bound on the supremum time so that the backward process is contractive on $[0, t^\star]$,

$$t^\star = T - 2\bar{\beta}(1 + \log(1 + \mathrm{diam}(\mathcal{M}))). \tag{14}$$

We then obtain the following bound.

**Proposition 6.** *Assume* **A**1 *and* $T \geq 2\bar{\beta}(1 + \log(1 + \text{diam}(\mathcal{M})))$. *Let* $t_K \in [0, T)$. *Then, for any* $s \in [0, t_K]$ *and* $x \in \mathbb{R}^d$ *we have*

$$\|\nabla \mathbf{Y}^x_{s,t_K}\| \leq \exp[-(1/2) \int_{T-t^\star}^{T-s} \beta_u \mathrm{d}u \mathbb{1}_{[0,t^\star)}(s)] \exp[(\text{diam}(\mathcal{M})^2/2)\sigma^{-2}_{T-t_K}].$$

*Proof.* Let $x \in \mathbb{R}^d$. First, using (13) and Lemma C.2 we have that for any $s, t \in [0, T]$ with $s \leq t$

$$\mathrm{d}\|\nabla \mathbf{Y}^x_{s,t}\|^2 \leq 2\beta_{T-t}(\|\nabla \mathbf{Y}^x_{s,t}\|^2 - 2(1 - m^2_{T-t}\text{diam}(\mathcal{M})^2/(2\sigma^2_{T-t}))/\sigma^2_{T-t}\|\nabla \mathbf{Y}^x_{s,t}\|^2)\mathrm{d}t.$$

First, assume that $s \leq t^\star$ and that $t \geq t^\star$. In that case, using Lemma D.8 we have that

$$\int_s^{t^\star} \beta_{T-u}(1 - 2/\sigma^2_{T-u} + m^2_{T-u}\text{diam}(\mathcal{M})^2/\sigma^4_{T-u})\mathrm{d}u \leq -(1/2) \int_s^{t^\star} \beta_{T-u}\mathrm{d}u.$$

Therefore, using that result and the fact that $\nabla \mathbf{Y}^x_{s,s} = \text{Id}$, we get that

$$\|\nabla \mathbf{Y}^x_{s,t^\star}\| \leq \exp[-(1/2) \int_{T-t^\star}^{T-s} \beta_u \mathrm{d}u]. \tag{15}$$

In addition, using Lemma D.8 we have that

$$\int_{t^\star}^t \beta_{T-u}(1 - 2/\sigma^2_{T-u} + m^2_{T-u}\text{diam}(\mathcal{M})^2/\sigma^4_{T-u})\mathrm{d}u \leq (\text{diam}(\mathcal{M})^2/2)(\sigma^{-2}_{T-t} - \sigma^{-2}_{T-t^\star}).$$

Therefore, we get that

$$\|\nabla \mathbf{Y}^x_{s,t}\| \leq \exp[(\text{diam}(\mathcal{M})^2/2)\sigma^{-2}_{T-t}]\|\nabla \mathbf{Y}^x_{s,t^\star}\|.$$

Hence, combining this result and (15), in the case where $s \leq t^\star$ we have

$$\|\nabla \mathbf{Y}^x_{s,t}\| \leq \exp[-(1/2) \int_{T-t^\star}^{T-s} \beta_s \mathrm{d}s] \exp[(\text{diam}(\mathcal{M})^2/2)\sigma^{-2}_{T-t}].$$

The proof in the cases where $s \geq t^\star$, $t \geq t^\star$ and $s \leq t^\star$, $t \leq t^\star$ are similar and left to the reader. $\qquad\square$

Our next goal is to control $\|\Delta b\|$. We recall that $b, \bar{b} : [0, T] \times \text{C}([0, T], \mathbb{R}^d) \to \mathbb{R}^d$ where for any $u \in [0, T)$ such that $u \in [s_k, t_{k+1})$ for some $k \in \{0, \ldots, K\}$ and $\omega = (\omega_v)_{v \in [s,T]} \in \text{C}([s, T], \mathbb{R}^d)$ [6] we have

$$b_u(\omega) = \beta_{T-u}(\omega_u + 2\nabla \log p_{T-u}(\omega_u)), \qquad \bar{b}_u(\omega) = \beta_{T-u}(\omega_u + 2\boldsymbol{s}(T - s_k, \omega_{s_k})),$$
$$\Delta b_u(\omega) = b_u(\omega) - \bar{b}_u(\omega),$$

where $s_k = \max(s, t_k)$. We now provide upper bounds on $\Delta b$. We introduce the intermediate drift functions $b^{(a)}, b^{(b)}, b^{(c)}, b^{(d)}$ such that $b^{(a)} = b$ and $b^{(d)} = \bar{b}$. In addition, for any $s, u \in [0, T)$ such that $u \geq s$, $u \in [s_k, t_{k+1})$ for some $k \in \{0, \ldots, K\}$ and for any $\omega = (\omega_v)_{v \in [s,T]} \in \text{C}([s, T], \mathbb{R}^d)$ we have

$$b^{(b)}_u(\omega) = \beta_{T-u}(\omega_u + 2\nabla \log p_{T-s_k}(\omega_u)), \qquad b^{(c)}_u(\omega) = \beta_{T-u}(\omega_u + 2\nabla \log p_{T-s_k}(\omega_{s_k})),$$
$$\Delta^{(a,b)}b = b^{(a)} - b^{(b)}, \qquad \Delta^{(b,c)}b = b^{(b)} - b^{(c)}, \qquad \Delta^{(c,d)}b = b^{(c)} - b^{(d)},$$

where $s_k = \max(s, t_k)$. We have that

$$\|\Delta b\| \leq \|\Delta^{(a,b)}b\| + \|\Delta^{(b,c)}b\| + \|\Delta^{(c,d)}b\|. \tag{16}$$

In the rest of this section, we control each term on the right hand side of (16).

**Lemma 7.** *For any* $s, u \in [0, T)$ *such that* $u \geq s$, $u \in [s_k, t_{k+1})$ *for some* $k \in \{0, \ldots, K\}$ *and* $\omega = (\omega_v)_{v \in [s,T]} \in \text{C}([s, T], \mathbb{R}^d)$ *we have*

$$\|\Delta^{(a,b)}b_u(\omega)\| \leq 2 \sup_{v \in [T-u, T-t_k]}(\beta^2_v/\sigma^6_v)(2 + \text{diam}(\mathcal{M})^2)(\text{diam}(\mathcal{M}) + \|\omega_u\|)\gamma_k.$$

---

[6]With a slight abuse of notation we assume that each process on $\text{C}([s, T])$ is extended on $\text{C}([0, T])$ by setting $\omega_u = \omega_s$ for any $u \in [0, s]$.

*Proof.* Assume that $s \leq t_k$. Then, we have

$$\|\Delta^{(a,b)}b_u(\omega)\| \leq 2\beta_{T-u}\|\nabla \log p_{T-u}(\omega_u) - \nabla \log p_{T-t_k}(\omega_u)\|$$
$$\leq 2\beta_{T-u}\gamma_k \sup_{v \in [T-u, T-t_k]} \|\partial_v \nabla \log p_{T-v}(\omega_u)\|.$$

Using Lemma C.3, we have that

$$\|\Delta^{(a,b)}b_u(\omega)\| \leq 2\beta_{T-u} \sup_{v \in [T-u, T-t_k]}(\beta_v/\sigma_v^6)(2 + \mathrm{diam}(\mathcal{M})^2)(\mathrm{diam}(\mathcal{M}) + \|\omega_u\|)\gamma_k,$$

which concludes the proof in the case where $s \leq t_k$. The case where $s \geq t_k$ is similar and left to the reader. $\square$

**Lemma 8.** *For any $s, u \in [0, T)$ such that $u \geq s$, $u \in [s_k, t_{k+1})$ for some $k \in \{0, \ldots, K\}$ and $\omega = (\omega_v)_{v \in [s,T]} \in$ $\mathrm{C}([s,T], \mathbb{R}^d)$ we have*

$$\|\Delta^{(b,c)}b_u(\omega)\| \leq 2(\beta_{T-u}/\sigma_{T-u}^4)(1 + \mathrm{diam}(\mathcal{M})^2)\|\omega_u - \omega_{s_k}\|,$$

*where $s_k = \max(s, t_k)$.*

*Proof.* Assume that $s \leq t_k$. We have

$$\|\Delta^{(b,c)}b_u(\omega)\| \leq 2\beta_{T-u}\|\nabla \log p_{T-t_k}(\omega_{t_k}) - \nabla \log p_{T-t_k}(\omega_u)\|$$
$$\leq 2\beta_{T-u} \sup_{v \in [u, T-t_k]} \|\nabla^2 \log p_{T-t_k}(\omega_v)\|\|\omega_u - \omega_{t_k}\|.$$

Using Lemma C.2 we have that

$$\|\Delta^{(b,c)}b_u(\omega)\| \leq 2(\beta_{T-u}/\sigma_{T-u}^4)(1 + \mathrm{diam}(\mathcal{M})^2)\|\omega_u - \omega_{t_k}\|,$$

which concludes the proof in the case where $s \leq t_k$. The case where $s \geq t_k$ is similar and left to the reader. $\square$

Finally, combining Lemma 7, Lemma 8 and **A3** in (16), we get that for any $s, u \in [0, T)$ such that $u \geq s$, $u \in [s_k, t_{k+1})$ for some $k \in \{0, \ldots, K\}$ and $(\omega_v)_{v \in [s,T]} \in \mathrm{C}([s,T], \mathbb{R}^d)$ we have

$$\begin{aligned}\|\Delta b_u(\omega)\| &\leq 2 \sup_{v \in [T-t_{k+1}, T-t_k]}(\beta_v^2/\sigma_v^6)(2 + \mathrm{diam}(\mathcal{M})^2)(\mathrm{diam}(\mathcal{M}) + \|\omega_u\|)\gamma_k \\ &\quad + 2(\beta_{T-u}/\sigma_{T-u}^4)(1 + \mathrm{diam}(\mathcal{M})^2)\|\omega_u - \omega_{s_k}\| \\ &\quad + 2\beta_{T-u}\mathtt{M}(1 + \|\omega_{s_k}\|)/\sigma_{T-u}^2,\end{aligned} \tag{17}$$

where $s_k = \max(s, t_k)$.

The following proposition controls the local error between the continuous-time backward process and the interpolation of the discretized one where the true score is replaced by the approximation $s$.

**Proposition 9.** *Assume* **A1**, **A2**, **A3**, **A4**. *In addition, assume that $\delta, \mathtt{M}, \gamma_K \leq 1/32$. Then, we have for any $s, u \in [0, t_K]$ with $u \geq s$*

$$\mathbb{E}[\|\Delta b_u((\bar{\mathbf{Y}}_{s,v})_{v \in [s,T]})\|] \leq \mathtt{C}_0(T - t_K + \bar{\beta})^2(\mathtt{M} + \delta^{1/2})/(T - t_K)^2,$$

*where $\bar{\mathbf{Y}}_{s,s} \sim \mathrm{N}(0, \mathrm{Id})$ and*

$$\mathtt{C}_0 = (1 + \bar{\beta})^{7/2}(4 + 256d + 43664(1 + \mathrm{diam}(\mathcal{M}))^4). \tag{18}$$

*Proof.* Let $s, u \in [0, t_K]$ with $u \geq s$. In what follows, for ease of notation, we denote for any $k \in \{0, \ldots, K\}$

$$\kappa_k = \sup_{v \in [T-t_{k+1}, T-t_k]} \beta_v/\sigma_v^2.$$

There exists $k \in \{0, \ldots, K-1\}$ such that $u \in [t_k, t_{k+1}]$. Assume that $s \leq t_k$. Recall that using (17), we have that for any $\omega = (\omega_v)_{v \in [s,T]} \in \mathrm{C}([s,T], \mathbb{R}^d)$

$$\|\Delta b_u(\omega)\| \leq 2 \sup_{v \in [T-t_{k+1}, T-t_k]}(\beta_v^2/\sigma_v^6)(2 + \mathrm{diam}(\mathcal{M})^2)(\mathrm{diam}(\mathcal{M}) + \|\omega_u\|)\gamma_k$$

$$+ 2(\beta_{T-u}/\sigma_{T-u}^4)(1 + \mathrm{diam}(\mathcal{M})^2)\|\omega_u - \omega_{t_k}\|$$
$$+ 2\beta_{T-u}\mathtt{M}(1 + \|\omega_{s_k}\|)/\sigma_{T-u}^2$$
$$\leq 2(\kappa_k^2/\sigma_{T-t_{k+1}}^2)\gamma_k(2 + \mathrm{diam}(\mathcal{M})^2)(\mathrm{diam}(\mathcal{M}) + \|\omega_u\|)$$
$$+ 2\kappa_k^2(1 + \mathrm{diam}(\mathcal{M})^2)\|\omega_u - \omega_{t_k}\|/\beta_{T-u} + 2\kappa_k\mathtt{M}(1 + \|\omega_{s_k}\|).$$

Combining this result with Lemma D.5, its following remark and Lemma D.6, we get that

$$\mathbb{E}[\|\Delta b_u((\bar{\mathbf{Y}}_{s,v})_{v\in[s,T]})\|] \leq 2(\kappa_k^2/\sigma_{T-t_{k+1}}^2)\gamma_k(2 + \mathrm{diam}(\mathcal{M})^2)(\mathrm{diam}(\mathcal{M}) + \mathtt{K}_0^{1/2})$$
$$+ 2\kappa_k^2(1 + \mathrm{diam}(\mathcal{M})^2)\mathtt{L}_0^{1/2}\bar{\beta}^{3/2}\gamma_k^{1/2} + 2\kappa_k\mathtt{M}(1 + \mathtt{K}_0^{1/2}).$$

Denoting $\mathtt{C} = 2(2 + \mathrm{diam}(\mathcal{M})^2)(\mathrm{diam}(\mathcal{M}) + \mathtt{K}_0^{1/2}) + 2\mathtt{L}_0^{1/2}\bar{\beta}^{3/2}(1 + \mathrm{diam}(\mathcal{M})^2) + 2(1 + \mathtt{K}_0^{1/2})$, we get that

$$\mathbb{E}[\|\Delta b_u((\bar{\mathbf{Y}}_{s,v})_{v\in[s,T]})\|] \leq \mathtt{C}((\kappa_k^2/\sigma_{T-t_{k+1}}^2)\gamma_k + \kappa_k^2\gamma_k^{1/2} + \mathtt{M}\kappa_k).$$

Combining this result, **A**4 and Lemma D.3 we have

$$\mathbb{E}[\|\Delta b_u((\bar{\mathbf{Y}}_{s,v})_{v\in[s,T]})\|] \leq \mathtt{C}(1+\bar{\beta})^2(1 + \bar{\beta}/(T-t_K))^2(\delta^{1/2} + \mathtt{M})$$
$$+ \mathtt{C}(1+\bar{\beta})(1 + \bar{\beta}/(T-t_K))(\delta/\sigma_{T-t_K}^2)$$
$$\leq \mathtt{C}(1+\bar{\beta})^2(T-t_K+\bar{\beta})^2(\delta^{1/2} + \mathtt{M})/(T-t_K)^2 + \mathtt{C}(1+\bar{\beta})(T-t_K+\bar{\beta})(\delta/\sigma_{T-t_K}^2)/(T-t_K).$$

Finally, using Lemma D.2, we have $\sigma_{T-t_K}^{-2} = (1 - \exp[-2\int_0^{T-t_K}\beta_s\mathrm{d}s])^{-1} \leq 1 + \bar{\beta}/(2(T-t_K))$ Therefore, using that $\gamma_K = T - t_K < 1$ we get that

$$\mathbb{E}[\|\Delta b_u((\bar{\mathbf{Y}}_{s,v})_{v\in[s,T]})\|] \leq \mathtt{C}(1+\bar{\beta})^2(T-t_K+\bar{\beta})^2(\delta + \delta^{1/2} + \mathtt{M})/(T-t_K)^2$$
$$\leq 2\mathtt{C}(1+\bar{\beta})^2(T-t_K+\bar{\beta})^2(\delta^{1/2} + \mathtt{M})/(T-t_K)^2$$

which concludes the first part of the proof in the case where $s \leq t_k$. The same bound holds in the case where $s \geq t_k$. Finally, we conclude upon noticing that $2\mathtt{C}(1+\bar{\beta})^2 \leq \mathtt{C}_0$ with $\mathtt{C}_0$ given by (18). □

We are now ready to control the global error between the backward process and the interpolation of the associated discrete-time process where the true score has been replaced by its approximation $\boldsymbol{s}$.

**Proposition 10.** *Assume* **A**1, **A**2, **A**3, **A**4 *and* $\gamma_K = \varepsilon$. *In addition, assume that* $\varepsilon, \delta, \mathtt{M} \leq 1/32$. *Then*

$$\mathbf{W}_1(\pi_\infty\mathrm{Q}_{t_K}, \pi_\infty\mathrm{R}_K) \leq \mathtt{D}_0\exp[\mathrm{diam}(\mathcal{M})^2(1+\bar{\beta})/(2\varepsilon)](\mathtt{M} + \delta^{1/2})/\varepsilon^2,$$

*where* $\mathtt{D}_0 = (1+\bar{\beta})^7(8 + 512d + 87328(1 + \mathrm{diam}(\mathcal{M}))^4)(1 + \log(1 + \mathrm{diam}(\mathcal{M})))$.

*Proof.* Using Proposition 5, we have

$$\|\mathbf{Y}_{t_K} - Y_K\| = \|\mathbf{Y}_{t_K} - \bar{\mathbf{Y}}_{t_K}\| \leq \int_0^{t_K} \|\nabla\mathbf{Y}_{u,t_K}^{\bar{\mathbf{Y}}_{0,u}}\|\|\Delta b_u((\bar{\mathbf{Y}}_{0,v})_{v\in[0,T]})\|\mathrm{d}u.$$

Combining this result, recalling that $t^\star$ is defined in (14) and Proposition 6, we get

$$\|\mathbf{Y}_{t_K} - Y_K\| \leq \int_0^{t_K} \exp[-(1/2)\int_{T-t^\star}^{T-u}\beta_s\mathrm{d}s\mathbb{1}_{[0,t^\star)}(u)]\exp[(\mathrm{diam}(\mathcal{M})^2/2)\sigma_{T-t_K}^{-2}]\|\Delta b_u((\bar{\mathbf{Y}}_{0,v})_{v\in[0,T]})\|\mathrm{d}u$$
$$\leq \exp[(\mathrm{diam}(\mathcal{M})^2/2)\sigma_{T-t_K}^{-2}](\int_0^{t^\star}\exp[-(1/2)\int_{T-t^\star}^{T-u}\beta_s\mathrm{d}s]\|\Delta b_u((\bar{\mathbf{Y}}_{0,v})_{v\in[0,T]})\|\mathrm{d}u$$
$$+ \int_{t^\star}^{t_K}\|\Delta b_u((\bar{\mathbf{Y}}_{0,v})_{v\in[0,T]})\|\mathrm{d}u).$$

Using this result and Proposition 9 we get

$$\mathbf{W}_1(\pi_\infty\mathrm{Q}_{t_K}, \pi_\infty\mathrm{R}_K) \leq \mathbb{E}[\|\mathbf{Y}_{t_K} - Y_K\|]$$
$$\leq \exp[(\mathrm{diam}(\mathcal{M})^2/2)\sigma_{T-t_K}^{-2}](\int_0^{t^\star}\exp[-(1/2)\int_{T-t^\star}^{T-u}\beta_s\mathrm{d}s]\mathbb{E}[\|\Delta b_u((\bar{\mathbf{Y}}_{0,v})_{v\in[0,T]})\|]\mathrm{d}u$$

$$+ \int_{t^\star}^{t_K} \mathbb{E}[\|\Delta b_u((\bar{\mathbf{Y}}_{0,v})_{v \in [0,T]})\|] \mathrm{d}u).$$

$$\leq \exp[(\mathrm{diam}(\mathcal{M})^2/2)\sigma_{T-t_K}^{-2}]\mathtt{C}_0(T - t_K + \bar{\beta})^2(\mathtt{M} + \delta^{1/2})/(T - t_K)^2$$

$$\times (\int_0^{t^\star} \exp[-(1/2)\int_{T-t^\star}^{T-u} \beta_s \mathrm{d}s]\mathrm{d}u + t_K - t^\star). \tag{19}$$

We have that

$$\int_0^{t^\star} \exp[-(1/2)\int_{T-t^\star}^{T-u} \beta_s \mathrm{d}s]\mathrm{d}u \leq \int_0^{t^\star} \exp[-(t^\star - u)/(2\bar{\beta})]\mathrm{d}u \leq 2\bar{\beta}. \tag{20}$$

In addition, using (14) we have

$$t_K - t^\star = T - \varepsilon - T + 2\bar{\beta}(1 + \log(1 + \mathrm{diam}(\mathcal{M}))) \leq 2\bar{\beta}(1 + \log(1 + \mathrm{diam}(\mathcal{M}))). \tag{21}$$

Using Lemma D.2, we have that $\sigma_{T-t_K}^{-2} \leq (1 + \bar{\beta})/\varepsilon$. Combining this result, (20) and (21) in (19) we get

$$\mathbf{W}_1(\pi_\infty \mathrm{Q}_{t_K}, \pi_\infty \mathrm{R}_K) \leq 2\mathtt{C}_0 \exp[\mathrm{diam}(\mathcal{M})^2(1 + \bar{\beta})/(2\varepsilon)](1 + \bar{\beta})^3(1 + \log(1 + \mathrm{diam}(\mathcal{M})))(\mathtt{M} + \delta^{1/2})/\varepsilon^2,$$

which concludes the proof. $\qquad\square$

## 4.2 Control of $\mathbf{W}_1(\pi_\infty \mathrm{Q}_{t_K}, \pi \mathrm{P}_{T-t_K})$

In this section, we focus on the error $\mathbf{W}_1(\pi_\infty \mathrm{Q}_{t_K}, \pi \mathrm{P}_{T-t_K})$. First, note that $\pi \mathrm{P}_{T-t_K} = \pi \mathrm{P}_T \mathrm{Q}_{t_K}$. Therefore, using Proposition D.9, we have

$$\mathbf{W}_1(\pi_\infty \mathrm{Q}_{t_K}, \pi \mathrm{P}_{T-t_K}) = \mathbf{W}_1(\pi_\infty \mathrm{Q}_{t_K}, \pi \mathrm{P}_T \mathrm{Q}_{t_K}) \leq \exp[(1/2)\sigma_{T-t_K}^{-2}]\mathbf{W}_1(\pi \mathrm{P}_T, \pi_\infty). \tag{22}$$

To control $\mathbf{W}_1(\pi \mathrm{P}_T, \pi_\infty)$, we use a synchronous coupling, i.e. we set $(\mathbf{Y}_t, \mathbf{Z}_t)_{t \in [0,T]}$ such that

$$\mathrm{d}\mathbf{Y}_t = -\beta_t \mathbf{Y}_t \mathrm{d}t + \sqrt{2\beta_t}\mathrm{d}\mathbf{B}_t, \qquad \mathrm{d}\mathbf{Z}_t = -\beta_t \mathbf{Z}_t \mathrm{d}t + \sqrt{2\beta_t}\mathrm{d}\mathbf{B}_t,$$

where $(\mathbf{B}_t)_{t \in [0,T]}$ is a $d$-dimensional Brownian motion and $\mathbf{Y}_0 \sim \pi$, $\mathbf{Z}_0 \sim \pi_\infty$. We have that for any $t \in [0,T]$, $\mathbf{Z}_t \sim \pi_\infty$. In addition, denoting $u_t = \mathbb{E}[\|\mathbf{Y}_t - \mathbf{Z}_t\|]$ for any $t \in [0,T]$, we have that $u_t \leq u_0 \exp[-\int_0^t \beta_s \mathrm{d}s]$. Therefore, combining this result and (22), we get that

$$\mathbf{W}_1(\pi_\infty \mathrm{Q}_{t_K}, \pi \mathrm{P}_{T-t_K}) \leq \exp[(1/2)\sigma_{T-t_K}^{-2}]\exp[-\int_0^T \beta_t \mathrm{d}t]\mathbf{W}_1(\pi, \pi_\infty). \tag{23}$$

Therefore, using Lemma D.2, we have

$$\mathbf{W}_1(\pi_\infty \mathrm{Q}_{t_K}, \pi \mathrm{P}_{T-t_K}) \leq \exp[(1 + \bar{\beta})\mathrm{diam}(\mathcal{M})^2/(2\varepsilon)]\exp[-T/\bar{\beta}](\sqrt{d} + \mathrm{diam}(\mathcal{M})).$$

## 4.3 Control of $\mathbf{W}_1(\pi \mathrm{P}_{T-t_K}, \pi)$ and conclusion

In this section, we focus on the error $\mathbf{W}_1(\pi, \pi \mathrm{P}_{T-t_K})$ and conclude the proof. We have that $\mathbf{W}_1(\pi, \pi \mathrm{P}_{T-t_K}) \leq \mathbb{E}[\|X - m_{T-t_K}X + \sigma_{T-t_K}Z\|]$, with $X \sim \pi$ and $Z \sim \mathrm{N}(0, \mathrm{Id})$. Hence, using $1 - m_{T-t_K} \leq \sigma_{T-t_K}$, we have

$$\mathbf{W}_1(\pi, \pi \mathrm{P}_{T-t_K}) \leq \mathrm{diam}(\mathcal{M})(1 - m_{T-t_K}) + \sigma_{T-t_K}\sqrt{d} \leq (\mathrm{diam}(\mathcal{M}) + \sqrt{d})\sigma_{T-t_K}.$$

Using Lemma D.2 and this result we have

$$\mathbf{W}_1(\pi, \pi \mathrm{P}_{T-t_K}) \leq (2\bar{\beta})^{1/2}(\mathrm{diam}(\mathcal{M}) + \sqrt{d})\varepsilon^{1/2}. \tag{24}$$

We conclude the proof upon combining this result, (23) and Proposition 10 in (8)

# 5 Conclusion

In this work, we have studied the convergence of diffusion models under the manifold hypothesis and provided convergence guarantees w.r.t. the Wasserstein distance of order one. Our theoretical results show that diffusion models are able to recover target distributions defined on low-dimensional manifolds. One current

limitation of our results lies in the dependency w.r.t. $1/\varepsilon$ which is exponential in the general case and might be overly pessimistic. This dependency can be improved at the cost of imposing conditions on the Hessian of $\log p_t$ but further investigations are needed to establish similar results in realistic settings.

Our results can be extended in several directions. First, in this work we focused on the Ornstein–Uhlenbeck process as a forward noising process. It would be interesting to analyze other forward diffusions such as the critically-damped one (Dockhorn et al., 2021). Another extension would be to study other discretization frameworks such as predictor-corrector schemes (Song et al., 2021a) and to extend our analysis to more realistic statistical settings. Finally, it is a challenge to derive similar bounds for target distributions with $\mathbb{R}^d$ support and tail constraints.

Finally, we would like to deepen our study of the relationship between the geometry of the manifold $\mathcal{M}$ and the properties of the score function. Preliminary results from Appendix J.3 indicate that the convexity of $\mathcal{M}$ can be recovered from the properties of the score but it remains unclear if more can be said on the geometry of the manifold.

## Acknowledgements

We thank Arnaud Doucet, Émile Mathieu and James Thornton for providing feedback on an early version of the paper. We thank George Deligiannidis, Alain Durmus and Éric Moulines for useful discussions. Finally, we are indebted to Pierre Del Moral who pointed us toward his work on stochastic interpolation formulae. This work has been supported by The Alan Turing Institute through the Theory and Methods Challenge Fortnights event "Accelerating generative models and nonconvex optimisation", which took place on 6-10 June 2022 and 5-9 Sep 2022 at The Alan Turing Institute headquarters.

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

# A    Organization of the appendix

The appendix is organized as follows. We start by discussing our discretization scheme in Appendix B. In Appendix C, we provide upper bounds on the gradient and Hessian of the logarithmic gradient of the density of the forward process under the manifold assumption. In Appendix D, we control the stability of several backward processes. In Appendix E, we recall and adapt a stochastic interpolation formula from Del Moral and Singh (2019). We check the different assumptions on the noise schedule in Appendix G. A short proof of the results of Franzese et al. (2022) is presented in Appendix H. In Appendix I, we present an extension of our results in the case where error is controlled w.r.t. the $L^2$ norm, following the work of Lee et al. (2022). We improve on Theorem 1 in Appendix J under some Hessian conditions.

# B    Discretization of backward processes

In Appendix B.1, we briefly describe the links between our proposed discretization and the classical Euler-Maruyama discretization. In Appendix B.2, we show that the discretization (5) is associated to the one of Ho et al. (2020) under specific settings

## B.1    Link with Euler-Maruyama discretization

First, we recall the Euler-Maruyama discretization. Given a sequence of stepsizes a discretization of (4) is given by the so-called Euler-Maruyama approximation, *i.e.* we define for any $k \in \{0, \dots, K\}$ and $t \in [t_k, t_{k+1}]$

$$\mathrm{d}\bar{\mathbf{Y}}_t^{\mathrm{EM}} = \beta_{T-t_k}\{\bar{\mathbf{Y}}_{t_k}^{\mathrm{EM}} + 2\boldsymbol{s}(T - t_k, \bar{\mathbf{Y}}_{t_k}^{\mathrm{EM}})\}\mathrm{d}t + \sqrt{2\beta_{T-t_k}}\mathrm{d}\mathbf{B}_t, \qquad \bar{\mathbf{Y}}_0^{\mathrm{EM}} \sim \pi_\infty. \tag{25}$$

The associated discrete process $(Y_k^{\mathrm{EM}})_{k \in \{0, \dots, K+1\}}$ is given for any $k \in \{0, \dots, K+1\}$ by $Y_k^{\mathrm{EM}} = \bar{\mathbf{Y}}_{t_k}^{\mathrm{EM}}$ and we have for any $k \in \{0, \dots, K\}$

$$Y_{k+1}^{\mathrm{EM}} = Y_k^{\mathrm{EM}} + \gamma_k \beta_{T-t_k}\{Y_k^{\mathrm{EM}} + 2\boldsymbol{s}(T - t_k, Y_k^{\mathrm{EM}})\} + \sqrt{2\beta_{T-t_k}\gamma_k}Z_k, \tag{26}$$

where $\{Z_k\}_{k \in \mathbb{N}}$ is a sequence of independent $d$-dimensional Gaussian random variables with zero mean and identity covariance matrix.

Note that (5) describes the same update as (25) up to the first order w.r.t. $\gamma_k$. In practice, there is no additional cost to replace the classical Euler-Maruyama discretization with the discretization defined in (5), provided that the integral of the temperature schedule $t \mapsto \beta_t$ can be computed in close form, which is the case in all the cases considered experimentally, see Appendix G.

However, in our theoretical analysis we found out that (5) introduces less error than (26) when compared to the approximate backward process (4). In our study we only consider the discretization scheme $(Y_k)_{k \in \{0, \dots, K+1\}}$ but emphasize that our analysis could be readily extended to derive similar discretization errors for the process $(Y_k^{\mathrm{EM}})_{k \in \{0, \dots, K+1\}}$.

## B.2    Equivalence with Ho et al. (2020)

In this section, we show that the discretization scheme introduced in Ho et al. (2020) and the one of (5) are equivalent up to the first order in some parameter.

**Setting of Ho et al. (2020)** We start by recalling the setting of Ho et al. (2020). Since, there is a conflict between our notations and the ones of Ho et al. (2020), we write our constants in red and the constants of Ho et al. (2020) in blue. The forward process in Ho et al. (2020) is given for any $t \in \{1, \ldots, T\}$[7]

$$q(x_t|x_0) = \mathrm{N}(x_t; \sqrt{\bar{\alpha}_t} x_0, (1 - \bar{\alpha}_t) \, \mathrm{Id}), \tag{27}$$

and we define

$$\beta_t = 1 - \alpha_t, \qquad \bar{\alpha}_t = \textstyle\prod_{s=1}^t \alpha_s. \tag{28}$$

In that case the loss function is given by

$$\ell(\theta) = \textstyle\sum_{t=1}^T \mathbb{E}[\|\epsilon_t - \epsilon_\theta(\sqrt{\bar{\alpha}_t} x_0 + \sqrt{1 - \bar{\alpha}_t} \epsilon_t, t)\|^2], \tag{29}$$

with $\{\epsilon_t\}_{t=1}^T$ a collection of independent Gaussian random variables with zero mean and identity covariance matrix. The backward sampling is given by the following recursion

$$x_{t-1} = \alpha_t^{-1/2}(x_t - (\beta_t/\sqrt{1 - \bar{\alpha}_t})\epsilon_\theta(x_t, t)) + \beta_t z_t, \tag{30}$$

with $\{z_t\}_{t=1}^T$ a collection of independent Gaussian random variables with zero mean and identity covariance matrix[8]. Note that using these notations, there is a conflict of notation between the forward process and the backward process. To clarify our identification, we denote $y_t = x_t$ for any $t \in \{0, \ldots, T\}$, with $x_t$ given by (30), in what follows.

**Identification** In what follows, we set $t = k + 1$, $T = K$ and for any $t \in \{1, \ldots, T\}$

$$\alpha_t = \exp[-2 \textstyle\int_{T-t_{K+1-k}}^{T-t_{K+1-(k+1)}} \beta_s \mathrm{d}s] = \exp[-2 \textstyle\int_{T-t_{K+1-k}}^{T-t_{K-k}} \beta_s \mathrm{d}s].$$

For instance, we have $\alpha_1 = \exp[-2 \int_0^{T-t_K} \beta_s \mathrm{d}s]$ and $\alpha_T = \exp[-2 \int_{T-t_1}^T \beta_s \mathrm{d}s]$. Note that in this case, using (28), we have

$$\bar{\alpha}_t^{1/2} = \exp[-\textstyle\int_0^{T-t_{K-k}} \beta_s \mathrm{d}s] = m_{T-t_{K-k}}.$$

Similarly, $\sqrt{1 - \bar{\alpha}_t} = \sigma_{T-t_{K-k}}$. In what follows, we identify the distribution of the forward process (27) with the one of (1), the loss function (29) with the one of (3) and the time reversal (30) with the one of (5).

(a) The distribution $q(x_t|x_0)$ given in (27) is the distribution of $\mathbf{X}_{T-t_{K-k}}|\mathbf{X}_0$ where $(\mathbf{X}_t)_{t \in [0,T]}$ is given in (1), since $\mathbf{X}_{T-t_{K-k}} = m_{T-t_{K-k}} \mathbf{X}_0 + \sigma_{T-t_{K-k}} Z$ with $Z \sim \mathrm{N}(0, \mathrm{Id})$. Therefore, we identify $x_t$ and $\mathbf{X}_{T-t_{K-k}}$ for any $t \in \{1, \ldots, T\}$. Similarly, for any $t \in \{1, \ldots, T\}$, we identify $t$ and $T - t_{K-k}$.

(b) Using that $\mathbf{X}_t = m_t \mathbf{X}_0 + \mathbf{B}_{\sigma_t}$ for any $t \in \{0, \ldots, T\}$, the loss is given by

$$\begin{aligned} \ell(\boldsymbol{s}) &= \textstyle\int_0^T \kappa(t) \mathbb{E}[\|\boldsymbol{s}(t, \mathbf{X}_t) - \nabla \log p_{t|0}(\mathbf{X}_t|\mathbf{X}_0)\|^2] \mathrm{d}t \\ &= \textstyle\int_0^T \kappa(t) \mathbb{E}[\|\boldsymbol{s}(t, \mathbf{X}_t) + \mathbf{B}_{\sigma_t}/\sigma_t^2\|^2] \mathrm{d}t \\ &= \textstyle\int_0^T \kappa(t)/\sigma_t^2 \mathbb{E}[\| - \sigma_t \boldsymbol{s}(t, \mathbf{X}_t) - \mathbf{B}_{\sigma_t}/\sigma_t\|^2] \mathrm{d}t \\ &= \textstyle\int_0^T \kappa(T-t)/\sigma_{T-t}^2 \mathbb{E}[\| - \sigma_{T-t} \boldsymbol{s}(T-t, \mathbf{X}_{T-t}) - \mathbf{B}_{\sigma_{T-t}}/\sigma_{T-t}\|^2] \mathrm{d}t. \end{aligned}$$

With a slight abuse of notation we assume that $\kappa(T - t) = \sum_{k=0}^K \delta_{t_{K-k}}(t) \sigma_{T-t}^2$ for any $t \in [0, T]$. Hence, we get that

$$\begin{aligned} \ell(\boldsymbol{s}) &= \textstyle\sum_{k=0}^K \mathbb{E}[\|\epsilon_t - (-\sigma_{T-t_{K-k}} \boldsymbol{s}(T - t_{K-k}, \mathbf{X}_{T-t_{K-k}}))\|^2] \\ &= \textstyle\sum_{k=0}^K \mathbb{E}[\|\epsilon_t - (-\sqrt{1 - \bar{\alpha}_t} \boldsymbol{s}(T - t_{K-k}, x_t))\|^2] \\ &= \textstyle\sum_{t=1}^T \mathbb{E}[\|\epsilon_t - (-\sqrt{1 - \bar{\alpha}_t} \boldsymbol{s}(t, \sqrt{\bar{\alpha}_t} x_0 + \sqrt{1 - \bar{\alpha}_t} \epsilon_t))\|^2] \end{aligned}$$

Hence, identifying $\epsilon_\theta(\cdot, t)$ and $-\sqrt{1 - \bar{\alpha}_t} \boldsymbol{s}(t, \cdot)$ for any $t \in \{1, \ldots, T\}$, we recover (29).

---

[7]Note that in Ho et al. (2020), $T$ is a number of steps and not the total time of the forward.
[8]We consider the case where $\sigma_t = \beta_t$.

(c) We now aim at recovering (30) from (5). Using the change of variable $k \to K - k$ and noting that for any $t \in (0, T]$, $\mathbf{B}_{\sigma_t}/\sigma_t$ is a Gaussian random variable with zero mean and identity covariance matrix we have

$$\begin{aligned}
Y_{K-k+1} &= Y_{K-k} + (\exp[\int_{T-t_{K-k+1}}^{T-t_{K-k}} \beta_s \mathrm{d}s] - 1)(Y_{K-k} + 2s(T - t_{K-k}, Y_{K-k})) \\
&\quad + (\exp[2\int_{T-t_{K-k+1}}^{T-t_{K-k}} \beta_s \mathrm{d}s] - 1)^{1/2} Z_{K-k} \\
&= Y_{K-k} + (\alpha_t^{-1/2} - 1)(Y_{K-k} + 2s(T - t_{K-k}, Y_{K-k})) + \sqrt{\beta_t} Z_{K-k} \\
&= \alpha_t^{-1/2} Y_{K-k} + 2(\alpha_t^{-1/2} - 1)s(T - t_{K-k}, Y_{K-k}) + \sqrt{\beta_t} Z_{K-k} \\
&= \alpha_t^{-1/2} Y_{K-k} - 2((\alpha_t^{-1/2} - 1)/\sqrt{1 - \bar{\alpha}_t})\epsilon_\theta(Y_{K-k}, t) + \sqrt{\beta_t} Z_{K-k}. \tag{31}
\end{aligned}$$

Finally, since $\alpha_t = 1 - \beta_t$ we have $\alpha_t^{1/2} = 1 - \beta_t/2 + o(\beta_t)$. This implies that $2(\alpha_t^{-1/2} - 1) = -\beta_t/\sqrt{\alpha_t} + o(\beta_t/\sqrt{\alpha_t})$. Therefore, combining this result and (31), we get that

$$Y_{K-k+1} = \alpha_t^{-1/2}(Y_{K-k} + (\beta_t/\sqrt{1 - \bar{\alpha}_t})\epsilon_\theta(t, Y_{K-k})) + \sqrt{\beta_t} Z_{K-k} + o(\beta_t/\sqrt{\alpha_t}).$$

This corresponds to (30) up to a term of the form $o(\beta_t/\sqrt{\alpha_t})$.

## C  Gradient and Hessian controls on the logarithmic density

Let $\pi \in \mathscr{P}(\mathbb{R}^d)$ be the target probability measure. We denote $(p_t)_{t>0}$ such that for any $t > 0$ the density w.r.t. the Lebesgue measure of the distribution of $\mathbf{X}_t$ (with initialization $\mathbf{X}_0^N \sim \pi$) is given by $p_t$. Similarly, $\pi^N \in \mathscr{P}(\mathbb{R}^d)$ be an empirical version of $\pi$, i.e. $\pi^N = (1/N)\sum_{k=1}^N X^k$, with $\{X^k\}_{k=1}^N \sim \pi^{\otimes N}$. We denote $(p_t^N)_{t>0}$ such that for any $t > 0$ the density w.r.t. the Lebesgue measure of the distribution of $\mathbf{X}_t^N$ (with initialization $\mathbf{X}_0^N \sim \pi^N$) is given by $p_t$. In order to show the stability and growth of the processes at hand we need to control quantities related to the gradient and Hessian of $\log q_t$ where $q_t = p_t$ or $p_t^N$. We first show a dissipativity condition on the gradient. We recall that for any $t \in [0, T]$

$$m_t = \exp[-\int_0^t \beta_s \mathrm{d}s], \qquad \sigma_t^2 = 1 - \exp[-2\int_0^t \beta_s \mathrm{d}s].$$

Such dissipativity conditions will allow us to control the moments of the introduced backward processes.

**Lemma C.1.** *Assume* **A**1. *Then for any $t \in (0, T]$ and $x_t \in \mathbb{R}^d$ we have that*

$$\langle \nabla \log q_t(x_t), x_t \rangle \leq -\|x_t\|^2/\sigma_t^2 + m_t \mathrm{diam}(\mathcal{M})\|x_t\|/\sigma_t^2,$$

*with $q_t = p_t^N$ or $p_t$. In addition, we have*

$$\|\nabla \log q_t(x_t)\|^2 \leq 2\|x_t\|^2/\sigma_t^4 + 2m_t^2 \mathrm{diam}(\mathcal{M})^2/\sigma_t^4. \tag{32}$$

*Proof.* Let $N \in \mathbb{N}$. We have that for any $t \in [0, T]$ and $x_t \in \mathbb{R}^d$

$$p_t^N(x_t) = (1/N)\sum_{k=1}^N \exp[-\|x_t - m_t X^k\|^2/2\sigma_t^2]/(2\pi\sigma_t^2)^{d/2},$$

Therefore, we get that for any $t \in [0, T]$ and $x_t \in \mathbb{R}^d$

$$\nabla \log p_t^N(x_t) = (-1/N)\sum_{k=1}^N (x_t - m_t X^k)\exp[-\|x_t - m_t X^k\|^2/2\sigma_t^2]/((2\pi\sigma_t^2)^{d/2}\sigma_t^2 p_t^N(x_t)).$$

Hence, we have

$$\langle \nabla \log p_t^N(x_t), x_t \rangle \leq -\|x_t\|^2/\sigma_t^2 + m_t \mathrm{diam}(\mathcal{M})\|x_t\|/\sigma_t^2.$$

Therefore taking the limit $N \to +\infty$, the same conclusion holds for $p_t$. The proof of (32) follows the same lines and is left to the reader. $\square$

We now provide controls on the Hessian $\nabla^2 \log q_t$. Such bounds allow to control the growth (or contraction) of the tangent process. This will also allow us to control the growth (or contraction) of the distance between backward processes w.r.t. the Wasserstein distance of order one.

**Lemma C.2.** *Assume* **A**1 *then we have for any* $t \in (0, T]$, $x_t \in \mathbb{R}^d$ *and* $\mathrm{M} \in \mathcal{M}_d(\mathbb{R}^d)$

$$\langle \mathrm{M}, \nabla^2 \log q_t(x_t)\mathrm{M} \rangle \leq -(1 - m_t^2 \mathrm{diam}(\mathcal{M})^2/(2\sigma_t^2))/\sigma_t^2 \|\mathrm{M}\|^2.$$

*In addition, we have*

$$\|\nabla^2 \log q_t(x_t)\| \leq (1 + \mathrm{diam}(\mathcal{M})^2)/\sigma_t^4.$$

*More generally, we have*

$$\nabla^2 \log p_t(x_t) = -\mathrm{Id}/\sigma_t^2$$
$$+ (2\sigma_t^4)^{-1} \int_{\mathcal{M} \times \mathcal{M}} (x_0 - x_0')^{\otimes 2} \exp[-\|x_t - m_t x_0\|^2/(2\sigma_t^2)] \exp[-\|x_t - m_t x_0'\|^2/(2\sigma_t^2)] \mathrm{d}\pi(x_0)\mathrm{d}\pi(x_0')$$
$$/(\int_{\mathcal{M}} \exp[-\|x_t - m_t x_0\|^2/(2\sigma_t^2)]\mathrm{d}\pi(x_0))^2.$$

*Proof.* Let $N \in \mathbb{N}$. For any $t \in (0, T]$ and $x \in \mathbb{R}^d$, we let $\bar{p}_t^N = p_t^N (2\pi\sigma_t^2)^{d/2}$ and we have

$$\bar{p}_t^N(x) = (1/N) \sum_{k=1}^N \exp[-\|x - m_t X^k\|^2/2\sigma_t^2],$$

Hence, we have

$$\nabla \log \bar{p}_t^N(x) = (-1/N) \sum_{k=1}^N (x - m_t X^k) \exp[-\|x - m_t X^k\|^2/2\sigma_t^2]/(\sigma_t^2 \bar{p}_t^N(x)).$$

Hence, we get that

$$\nabla^2 \log \bar{p}_t^N(x) = -\mathrm{Id}/\sigma_t^2$$
$$+ (1/N) \sum_{k=1}^N (x - m_t X^k) \otimes (x - m_t X^k) \exp[-\|x - m_t X^k\|^2/2\sigma_t^2]/(\sigma_t^4 \bar{p}_t^N(x))$$
$$- (1/N^2)(\sum_{k=1}^N (x - m_t X^k) \exp[-\|x - m_t X^k\|^2/2\sigma_t^2])$$
$$\otimes (\sum_{k=1}^N (x - m_t X^k) \exp[-\|x - m_t X^k\|^2/2\sigma_t^2])/(\sigma_t^2 \bar{p}_t^N(x))^2.$$

For any $k \in \{0, \dots, N-1\}$, denote $f_t^k = -(x - m_t X^k)/\sigma_t^2$ and $e_t^k = \exp[-\|f_t^k\|^2]$. Using the previous result, we have

$$\nabla^2 \log \bar{p}_t^N(x) = -\mathrm{Id}/\sigma_t^2 + \sum_{k=1}^N f_t^k \otimes f_t^k e_t^k / \sum_{k=1}^N e_t^k$$
$$- (\sum_{k=1}^N f_t^k e_t^k / \sum_{k=1}^N e_t^k) \otimes (\sum_{k=1}^N f_t^k e_t^k / \sum_{k=1}^N e_t^k)$$
$$= -\mathrm{Id}/\sigma_t^2 + (1/2) \sum_{j,k=1}^N (f_t^k - f_t^j) \otimes (f_t^k - f_t^j) e_t^k e_t^j / \sum_{k,j=1}^N e_t^k e_t^j. \tag{33}$$

In addition, using that for any $\ell \in \{1, \dots, N\}$, $X^\ell \in \mathcal{M}$ we have that

$$\|f_t^k - f_t^j\| = m_t \|X^k - X^j\|/\sigma_t^2 \leq m_t \mathrm{diam}(\mathcal{M})/\sigma_t^2.$$

Therefore, we get that

$$\langle \mathrm{M}, \nabla^2 \log \bar{p}_t^N(x)\mathrm{M} \rangle \leq -(1 - m_t^2 \mathrm{diam}(\mathcal{M})^2/(2\sigma_t^2))/\sigma_t^2 \|\mathrm{M}\|^2.$$

Using (33), the fact that $\mathcal{M}$ is compact and the strong law of large numbers we have that

$$\lim_{N \to +\infty} \nabla^2 \log \bar{p}_t^N(x) = -\mathrm{Id}/\sigma_t^2$$
$$+ \int_{\mathbb{R}^d} (x - m_t x_0) \otimes (x - m_t \bar{x}_0) \exp[-\|x - m_t x_0\|^2/(2\sigma_t^2)] \exp[-\|x - m_t \bar{x}_0\|^2/(2\sigma_t^2)]\mathrm{d}\pi(x_0)\mathrm{d}\pi(\bar{x}_0)$$
$$/(\int_{\mathbb{R}^d} \exp[-\|x - m_t \bar{x}_0\|^2/(2\sigma_t^2)]\mathrm{d}\pi(x_0))^2.$$

Hence, we get that $\lim_{N \to +\infty} \nabla^2 \log p_t^N(x) = \nabla^2 \log p_t$, which concludes the proof. $\square$

Finally, in order to control the local error of the time discretization, we also need to control the time derivative of the gradient, *i.e.* $\partial_t \nabla \log q_t$.

**Lemma C.3.** *Assume* **A**1. *Then for any $t \in (0, T]$ and $x_t \in \mathbb{R}^d$ we have that*

$$\|\partial_t \nabla \log q_t(x_t)\| \leq (\beta_t/\sigma_t^6)(2 + \mathrm{diam}(\mathcal{M})^2)(\mathrm{diam}(\mathcal{M}) + \|x\|).$$

*Proof.* Let $N \in \mathbb{N}$ and $t \in (0, T]$. Recall that for any $x \in \mathbb{R}^d$, $p_t^N(x) = \bar{p}_t^N(x)/(2\pi\sigma_t^2)^{d/2}$ with

$$\bar{p}_t^N(x) = (1/N)\sum_{k=1}^N e_t^k(x), \qquad e_t^k(x) = \exp[-\|x - m_t X^k\|^2/(2\sigma_t^2)].$$

In what follows, we denote $f_t^k = \log e_t^k$ for any $k \in \{1, \ldots, N\}$. For any $x \in \mathbb{R}^d$ we have

$$\partial_t \log \bar{p}_t^N(x) = \sum_{k=1}^N \partial_t f_t^k(x) e_t^k(x)/\sum_{k=1}^N e_t^k(x).$$

Therefore, we have for any $x \in \mathbb{R}^d$

$$\begin{aligned}
\partial_t \nabla \log \bar{p}_t^N(x) &= \sum_{k=1}^N \partial_t \nabla f_t^k(x) e_t^k(x)/\sum_{k=1}^N e_t^k(x) \\
&\quad + \sum_{k=1}^N \partial_t f_t^k(x) \nabla f_t^k(x) e_t^k(x)/\sum_{k=1}^N e_t^k(x) \\
&\quad - \sum_{k,j=1}^N \partial_t f_t^k(x) \nabla f_t^j(x) e_t^k(x) e_t^j(x)/\sum_{k,j=1}^N e_t^k(x) e_t^j(x) \\
&= \sum_{k=1}^N \partial_t \nabla f_t^k(x) e_t^k(x)/\sum_{k=1}^N e_t^k(x) \\
&\quad + (1/2)\sum_{k,j=1}^N (\partial_t f_t^k(x) - \partial_t f_t^j(x))(\nabla f_t^k(x) - \nabla f_t^j(x)) e_t^k(x) e_t^j(x)/\sum_{k,j=1}^N e_t^k(x) e_t^j(x). \quad (34)
\end{aligned}$$

In what follows, we fix $k, j \in \{1, \ldots, N\}$ and provide upper bounds for $|\partial_t f_t^k - \partial_t f_t^j|$, $\|\nabla f_t^k - \nabla f_t^j\|$ and $\partial_t \nabla f_t^k$. First, we have that for any $x \in \mathbb{R}^d$, $\nabla f_t^k(x) = -(x - m_t X^k)/\sigma_t^2$. Hence, using that $m_t \leq 1$, we get that for any $x \in \mathbb{R}^d$

$$\|\nabla f_t^k(x) - \nabla f_t^j(x)\| \leq m_t \mathrm{diam}(\mathcal{M})/\sigma_t^2 \leq \mathrm{diam}(\mathcal{M})/\sigma_t^2. \quad (35)$$

In addition, we have that for any $x \in \mathbb{R}^d$

$$\partial_t f_t^k(t) = \partial_t \sigma_t^2/(2\sigma_t^4)\|x - m_t X^k\|^2 + \partial_t m_t/\sigma_t^2 \langle X^k, x - m_t X^k \rangle.$$

Combining this result, the fact that $\partial_t \sigma_t^2 = -2m_t \partial_t m_t$ and that $\partial_t m_t = -\beta_t m_t$, we get that

$$\begin{aligned}
\partial_t f_t^k(t) &= -\beta_t m_t/\sigma_t^2[-(m_t/\sigma_t^2)\|x - m_t X^k\|^2 + \langle x - m_t X^k, X^k \rangle] \\
&= -\beta_t m_t/\sigma_t^2 \langle x - m_t X^k, -(m_t/\sigma_t^2)(x - m_t X^k) + X^k \rangle \\
&= -\beta_t m_t/\sigma_t^4 \langle x - m_t X^k, -m_t x + X^k \rangle \\
&= \beta_t m_t/\sigma_t^4(m_t\|x\|^2 + m_t\|X^k\|^2 + (1 + m_t^2)\langle x, X^k \rangle). \quad (36)
\end{aligned}$$

Using this result and that $m_t \leq 1$, we have that for any $x \in \mathbb{R}^d$

$$\begin{aligned}
|\partial_t f_t^k(x) - \partial_t f_t^j(x)| &\leq 2\beta_t m_t^2 \mathrm{diam}(\mathcal{M})^2/\sigma_t^4 + \beta_t m_t(1 + m_t^2)\mathrm{diam}(\mathcal{M})\|x\|/\sigma_t^4 \quad (37) \\
&\leq 2(\beta_t/\sigma_t^4)\mathrm{diam}(\mathcal{M})(\mathrm{diam}(\mathcal{M}) + \|x\|).
\end{aligned}$$

Using (36), we have for any $x \in \mathbb{R}^d$

$$\nabla \partial_t f_t^k(x) = 2\beta_t m_t^2/\sigma_t^4 x + (\beta_t m_t/\sigma_t^4)(1 + m_t^2)X^k.$$

Therefore, combining this result and the fact that $m_t \leq 1$, we get that for any $x \in \mathbb{R}^d$

$$\|\partial_t \nabla f_t^k(x)\| \leq 2(\beta_t/\sigma_t^4)(\mathrm{diam}(\mathcal{M}) + \|x\|). \quad (38)$$

Combining (35), (37) and (38) in (34), we get that for any $x \in \mathbb{R}^d$

$$\begin{aligned}
\|\partial_t \nabla \log \bar{p}_t^N(x)\| &\leq 2(\beta_t/\sigma_t^4)(\mathrm{diam}(\mathcal{M}) + \|x\|) + (\beta_t/\sigma_t^6)\mathrm{diam}(\mathcal{M})^2(\mathrm{diam}(\mathcal{M}) + \|x\|) \\
&\leq (\beta_t/\sigma_t^6)(2 + \mathrm{diam}(\mathcal{M})^2)(\mathrm{diam}(\mathcal{M}) + \|x\|),
\end{aligned}$$

which concludes the proof using that $\lim_{N \to +\infty} \partial_t \nabla \log p_t^N(x_t) = \partial_t \nabla \log p_t$. □

We conclude this section with bounds on the higher-order differentials of $\log p_t$. To compute higher derivatives we will use the following lemma.

**Lemma C.4.** *Let $E = \{e_i\}_{i=1}^M$ be a family of functions such that for any $i \in \{1, \ldots, M\}$, $e_i \in \mathrm{C}^\infty(\mathbb{R}^d, \mathbb{R})$. Similarly, let $G = \{g_i\}_{i=1}^M$ be a family of functions such that for any $i \in \{1, \ldots, M\}$, $g_i \in \mathrm{C}^\infty(\mathbb{R}^d, \mathbb{R}^p)$. Let $F(E, G)$ such that for any $x \in \mathbb{R}^d$*

$$F(E, G) = \textstyle\sum_{i=1}^M g_i e_i / \sum_{i=1}^M e_i.$$

*Then, we have*

$$\nabla F(E, G) = F(E, \nabla G) + (1/2)F(E \otimes E, (G \ominus G) \odot (\nabla \log E \ominus \nabla \log E)),$$

*where $\otimes$ is the tensor product, $\odot$ the pointwise product and $\ominus$ the tensor substraction.*

*Proof.* We assume that $p = 1$. The proof in the general case is similar and left to the reader. We have that

$$
\begin{aligned}
\nabla F(E, G) &= \textstyle\sum_{i=1}^M \nabla g_i e_i / \sum_{i=1}^M e_i + \sum_{i,j=1}^M g_i \nabla \log(e_i) e_i e_j / \sum_{i,j=1}^M e_i e_j \\
&\quad - \textstyle\sum_{i,j=1}^M g_i \nabla \log(e_j) e_i e_j / \sum_{i,j=1}^M e_i e_j \\
&= \textstyle\sum_{i=1}^M \nabla g_i e_i / \sum_{i=1}^M e_i \\
&\quad + (1/2) \textstyle\sum_{i,j=1}^M (g_i \nabla \log(e_i) + g_j \nabla \log(e_j) - g_i \nabla \log(e_j) - g_j \nabla \log(e_i)) e_i e_j / \sum_{i,j=1}^M e_i e_j \\
&= \textstyle\sum_{i=1}^M \nabla g_i e_i / \sum_{i=1}^M e_i \\
&\quad + (1/2) \textstyle\sum_{i,j=1}^M (g_i - g_j)(\nabla \log e_i - \nabla \log e_j) / \sum_{i,j=1}^M e_i e_j,
\end{aligned}
$$

which concludes the proof. $\square$

**Lemma C.5.** *Assume* **A**1*. Then, there exists $C \geq 0$ such that for any $t \in (0, T]$ we have*

$$\|\nabla^2 \log p_t(x)\| + \|\nabla^3 \log p_t(x)\| + \|\nabla^4 \log p_t(x)\| \leq C/\sigma_t^8.$$

*Proof.* Let $t \in (0, T]$. First, remark that for any $x \in \mathbb{R}^d$

$$\nabla^2 \log p_t(x) = -\operatorname{Id}/\sigma_t^2 + F(E^{\otimes 2}, (\nabla \log E \ominus \nabla \log E)^{\odot 2}),$$

where $E = \{e_i\}_{i=1}^N$ and for any $i \in \{1, \ldots, N\}$, $e_i(x) = \exp[-\|x - m_t X^i\|^2/(2\sigma_t^2)]$. Note that $\nabla \log E \ominus \nabla \log E$ does not depend on $x$ and there exists $C_0 \geq 0$ such that for any $i, j \in \{1, \ldots, N\}$, $\|\nabla \log e_i(x) - \nabla \log e_j(x)\| \leq C_0/\sigma_t^2$. Hence, using Lemma C.4 we have

$$\nabla^3 \log p_t(x) = F(E^{\otimes 4}, (\nabla \log E \ominus \nabla \log E)^{\odot 2} \odot (\nabla \log(E \otimes E) \ominus \nabla \log(E \otimes E))).$$

Again, note that $G_1 = (\nabla \log E \ominus \nabla \log E)^{\odot 2} \odot (\nabla \log(E \otimes E) \ominus \nabla \log(E \otimes E))$ does not depend on $x$, upon remarking that

$$\nabla \log(E \otimes E) \ominus \nabla \log(E \otimes E) = (\nabla \log E \ominus \nabla \log E) \oplus (\nabla \log E \ominus \nabla \log E).$$

Finally, we have $\nabla^4 \log p_t(x) = F(E^{\otimes 8}, G_2)$, where

$$
\begin{aligned}
G_2 = &[((\nabla \log E \ominus \nabla \log E)^{\odot 2} \odot (\nabla \log(E \otimes E) \ominus \nabla \log(E \otimes E))) \\
&\ominus ((\nabla \log E \ominus \nabla \log E)^{\odot 2} \odot (\nabla \log(E \otimes E) \ominus \nabla \log(E \otimes E)))] \odot (\nabla \log(E^{\otimes 4}) \ominus \nabla \log(E^{\otimes 4})).
\end{aligned}
$$

Therefore, we get that there exists $C \geq 0$ such that for any $x \in \mathbb{R}^d$

$$\|\nabla \log p_t^3(x)\| \leq C/\sigma_t^6, \qquad \|\nabla \log p_t^4(x)\| \leq C/\sigma_t^8.$$

We conclude the proof upon using that $\sigma_t \leq 1$. $\square$

# D   Control of the backward processes

We start by introducing the different processes in Appendix D.1. We gather a few technical results in Appendix D.2. Then, we turn to the stability and Lipschitz properties of the backward processes in Appendix D.3. Finally, we control the growth of the backward tangent process in Appendix D.4.

## D.1   Introduction of the processes

In this section, we study the stability of the backward process given by

$$\mathrm{d}\mathbf{Y}_t = \beta_{T-t}\{\mathbf{Y}_t + 2\nabla \log q_{T-t}(\mathbf{Y}_t)\}\mathrm{d}t + \sqrt{2\beta_{T-t}}\mathrm{d}\mathbf{B}_t, \tag{39}$$

where $q_t$ is either $p_t$ or $p_t^N$. We are also going to consider the following approximate continuous-time process

$$\mathrm{d}\hat{\mathbf{Y}}_t = \beta_{T-t}\{\hat{\mathbf{Y}}_t + 2\boldsymbol{s}(T-t, \hat{\mathbf{Y}}_t)\}\mathrm{d}t + \sqrt{2\beta_{T-t}}\mathrm{d}\mathbf{B}_t, \tag{40}$$

where $\boldsymbol{s}(t, \cdot)$ is an approximation of either $p_t$ or $p_t^N$. Note that since $q_t > 0$ and $q \in \mathrm{C}^\infty((0, T] \times \mathbb{R}^d, \mathbb{R}^d)$ and that $\boldsymbol{s} \in \mathrm{C}^1((0, T] \times \mathbb{R}^d, \mathbb{R}^d)$ we have that (39) and (40) admit strong solutions up to an explosion time. Finally, we also consider the following interpolating process: for any $t \in [t_k, t_{k+1})$

$$\mathrm{d}\bar{\mathbf{Y}}_t = \beta_{T-t}\{\bar{\mathbf{Y}}_t + 2\boldsymbol{s}(T-t_k, \bar{\mathbf{Y}}_{t_k})\}\mathrm{d}t + \sqrt{2\beta_{T-t}}\mathrm{d}\mathbf{B}_t. \tag{41}$$

This process is an interpolation of a modified Euler–Maruyama discretization of (40). Note that the classical Euler–Maruyama discretization would be associated with the following interpolation

$$\mathrm{d}\bar{\mathbf{Y}}_t^{\mathrm{EM}} = \beta_{T-t_k}\{\bar{\mathbf{Y}}_{t_k}^{\mathrm{EM}} + 2\boldsymbol{s}(T-t_k, \bar{\mathbf{Y}}_{t_k}^{\mathrm{EM}})\}\mathrm{d}t + \sqrt{2\beta_{T-t_k}}\mathrm{d}\mathbf{B}_t.$$

In (41), we take advantage of the linear part of the drift. Indeed on the interval $[t_k, t_{k+1}]$, the process (41) is a simple Ornstein–Uhlenbeck which can be integrated explicitly. In particular for any $k \in \{0, \dots, N-1\}$ and $t \in [t_k, t_{k+1}]$ we have

$$\bar{\mathbf{Y}}_t = \bar{\mathbf{Y}}_{t_k} + (\exp[\textstyle\int_{T-t}^{T-t_k} \beta_s \mathrm{d}s] - 1)(\bar{\mathbf{Y}}_{t_k} + 2\boldsymbol{s}(T-t_k, \bar{\mathbf{Y}}_{t_k})) + (\exp[2\textstyle\int_{T-t}^{T-t_k} \beta_s \mathrm{d}s] - 1)^{1/2} Z,$$

where $Z$ is a Gaussian random variable with zero mean and identity covariance and the equality holds in distribution independent from $\bar{\mathbf{Y}}_{t_k}$. Denoting $\{Y_k\}_{k\in\{0,\dots,N-1\}}$, we get that for any $k \in \{0, \dots, N-1\}$

$$Y_{k+1} = Y_k + (\exp[\textstyle\int_{T-t_{k+1}}^{T-t_k} \beta_s \mathrm{d}s] - 1)(Y_k + 2\boldsymbol{s}(T-t_k, Y_k)) + (\exp[2\textstyle\int_{T-t_{k+1}}^{T-t_k} \beta_s \mathrm{d}s] - 1)^{1/2} Z_k,$$

where $\{Z_k\}_{k\in\mathbb{N}}$ is a collection of independent Gaussian random variables with zero mean and identity covariance matrix. Using this scheme instead of the classical Euler–Maruyama simplifies the analysis of the discretization. Up to the first order this scheme is equal to the classical Euler–Maruyama discretization. Once again, we emphasize that computing this scheme is as expensive as computing the classical Euler–Maruyama discretization provided that the integral $\int_s^t \beta_u \mathrm{d}u$ are available in close form for all $s, t \in [0, T]$ which is the case for all the discretization schemes used in practice. We refer to Table 1 for a list of all processes used in the proof. In what follows, we control the stability of these processes.

## D.2   Some useful technical lemmas

We gather in this section some technical results.

**Lemma D.1.** *For any* $s, t \in [0, T]$ *we have*

$$\textstyle\int_s^t \beta_{T-u}/\sigma_{T-u}^2 \mathrm{d}u = [(-1/2)\log(\exp[2\int_0^{T-u} \beta_v \mathrm{d}v] - 1)]_s^t, \tag{42}$$

$$\textstyle\int_s^t \beta_{T-u} m_{T-u}^2/\sigma_{T-u}^4 \mathrm{d}u = [(1/2)/(1 - \exp[-2\int_0^{T-u} \beta_v \mathrm{d}v])]_s^t. \tag{43}$$

*In particular, if* $\beta = \beta_0$ *then*

$$\textstyle\int_s^t \beta_{T-u}/\sigma_{T-u}^2 \mathrm{d}u = [(-1/2)\log(\exp[2\beta_0(T-u)] - 1)]_s^t,$$

$$\textstyle\int_s^t \beta_{T-u} m_{T-u}^2/\sigma_{T-u}^4 \mathrm{d}u = [(1/2)/(1 - \exp[-2\beta_0(T-u)])]_s^t.$$

*Proof.* We have

$$
\begin{aligned}
\int_s^t \beta_{T-u}/\sigma_{T-u}^2 \mathrm{d}u &= \int_s^t \beta_{T-u}/(1 - \exp[-2\int_0^{T-u} \beta_v \mathrm{d}v])\mathrm{d}u \\
&= \int_s^t \beta_{T-u} \exp[2\int_0^{T-u} \beta_v \mathrm{d}v]/(\exp[2\int_0^{T-u} \beta_v \mathrm{d}v] - 1)\mathrm{d}u \\
&= (-1/2)\int_s^t \partial_u \log(\exp[2\int_0^{T-u} \beta_v \mathrm{d}v] - 1)\mathrm{d}u.
\end{aligned}
$$

This concludes the proof of (42). We have

$$
\begin{aligned}
\int_s^t \beta_{T-u} m_{T-u}^2/\sigma_{T-u}^4 \mathrm{d}u &= \int_s^t \beta_{T-u} m_{T-u}^2/(1 - \exp[-2\int_0^{T-u} \beta_v \mathrm{d}v])^2 \mathrm{d}u \\
&= \int_s^t \beta_{T-u} \exp[-2\int_0^{T-u} \beta_v \mathrm{d}v]/(1 - \exp[-2\int_0^{T-u} \beta_v \mathrm{d}v])^2 \mathrm{d}u \\
&= (1/2)\int_s^t \partial_u (1 - \exp[-2\int_0^{T-u} \beta_v \mathrm{d}v])^{-1} \mathrm{d}u.
\end{aligned}
$$

This concludes the proof of (43). □

**Lemma D.2.** *Assume* **A**2. *We have* $\sigma_t^2 \leq 2t\bar{\beta}$ *and* $\sigma_t^{-2} \leq 1 + \bar{\beta}/(2t)$.

*Proof.* First, using that for any $a \geq 0$, $\exp[-a] \geq 1 - a$ we have

$$
\sigma_t^2 = 1 - \exp[-2\int_0^t \beta_s \mathrm{d}s] \leq 1 - \exp[-2\bar{\beta}t] \leq 2\bar{\beta}t.
$$

Second, using that for any $a \geq 0$, $1/(1 + \exp[-a]) \leq 1 + 1/a$ we have

$$
\sigma_t^{-2} = (1 - \exp[-2\int_0^t \beta_s \mathrm{d}s])^{-1} \leq 1 + (2\int_0^t \beta_s \mathrm{d}s)^{-1},
$$

which concludes the proof. □

For any $k \in \{0, \ldots, K\}$ we introduce $\kappa_k = \sup_{u \in [T-t_{k+1}, T-t_k]} \beta_u/\sigma_u^2$.

**Lemma D.3.** *Assume* **A**2. *Then, we have that for any* $k \in \{0, \ldots, N-1\}$

$$
\kappa_k \leq \bar{\beta}(1 + \bar{\beta}/t).
$$

*Proof.* Recall that for any $s \in [0, T]$, $\sigma_s^2 = 1 - \exp[-2\int_0^s \beta_u \mathrm{d}u]$. Using that for any $v \geq 0$, $(1 - e^{-2v})^{-1} \leq 1 + 1/(2v)$ we have for any $t \in (0, T]$

$$
\beta_t/\sigma_t^2 \leq \beta_t + \beta_t(2\int_0^t \beta_s \mathrm{d}s)^{-1} \leq \beta_t(1 + \bar{\beta}/t),
$$

which concludes the proof. □

### D.3 Stability and Lipschitz properties of the backward processes

The controls derived in Appendix C allow for uniform control of the moments of the backward processes. Note that such Lyapunov techniques were used in Fontaine et al. (2021) to control energy functionals in convex optimization problems. The following lemma is not used directly in our final analysis but provides intuitive controls on the backward process.

**Lemma D.4.** *Assume* **A**1, **A**3 *and that there exists* $\eta > 0$ *such that* $\mathtt{M} + \eta \mathrm{diam}(\mathcal{M}) \leq 1/4$. *Then, for any* $t \in [0, T]$ *we have*

$$
\mathbb{E}[\|\hat{\mathbf{Y}}_t\|^2] \leq d + 8(1 + \mathtt{M} + \mathrm{diam}(\mathcal{M})/\eta).
$$

*In particular if* $\mathtt{M} \leq 1/8$ *then for any* $t \in [0, T]$ *we have*

$$
\mathbb{E}[\|\hat{\mathbf{Y}}_t\|^2] \leq d + 8(1 + \mathtt{M} + 8\mathrm{diam}(\mathcal{M})^2).
$$

*Proof.* First, using Lemma C.1, we have that for any $t \in [0, T)$, $\mathbb{E}[\|\hat{\mathbf{Y}}_t\|^2] < +\infty$. Hence, using Itô's lemma, we have

$$\mathrm{d}(1/2)\|\hat{\mathbf{Y}}_t\|^2 = \beta_{T-t}\{\|\hat{\mathbf{Y}}_t\|^2 + 2\langle \boldsymbol{s}(T-t, \hat{\mathbf{Y}}_t), \hat{\mathbf{Y}}_t\rangle + 2\}\mathrm{d}t + \sqrt{2\beta_{T-t}}\langle \hat{\mathbf{Y}}_t, \mathrm{d}\mathbf{B}_t\rangle.$$

Therefore, using **A3**, Lemma C.1 and that for any $a, b, \eta > 0$, $2ab \leq a^2\eta + b^2/\eta$ we get that for any $t \in [0, T)$ we have that $(1/2)\mathbb{E}[\|\hat{\mathbf{Y}}_t\|^2] \leq u_t$, where $u_0 = d/2$ and

$$\begin{aligned}
\mathrm{d}u_t &= \beta_{T-t}(1 - 2/\sigma_{T-t}^2 + 2(\mathtt{M} + \eta m_{T-t}\mathrm{diam}(\mathcal{M}))/\sigma_{T-t}^2)u_t \\
&\quad + \beta_{T-t}(2 + 2\mathtt{M} + 2m_{T-t}\mathrm{diam}(\mathcal{M})/\eta)/\sigma_{T-t}^2 \\
&= \beta_{T-t}(\sigma_{T-t}^2 - 2 + 2(\mathtt{M} + \eta m_{T-t}\mathrm{diam}(\mathcal{M})))/\sigma_{T-t}^2 u_t \\
&\quad + \beta_{T-t}(2 + 2\mathtt{M} + 2m_{T-t}\mathrm{diam}(\mathcal{M})/\eta)/\sigma_{T-t}^2.
\end{aligned}$$

For any $t \in [0, T]$ and $x \in \mathbb{R}$ define $F(t, x)$ given by

$$F(t, x) = -(2 - \sigma_{T-t}^2 - 2(\mathtt{M} + m_{T-t}\eta\mathrm{diam}(\mathcal{M})))/\sigma_{T-t}^2 x + (2 + 2\mathtt{M} + 2m_{T-t}\mathrm{diam}(\mathcal{M})/\eta)/\sigma_{T-t}^2.$$

Using that for any $t \in [0, T]$, $m_t, \sigma_t^2 \in [0, 1]$, we have that for any $t \in [0, T]$

$$2 - \sigma_{T-t}^2 - 2(\mathtt{M} + \eta m_{T-t}\mathrm{diam}(\mathcal{M})) \geq 1 - 2(\mathtt{M} + \eta\mathrm{diam}(\mathcal{M})) \geq 1/2.$$

Using (Fontaine et al., 2021, Lemma 3) we get that for any $t \in [0, T]$

$$u_t \leq d/2 + 2(1 + \mathtt{M} + \mathrm{diam}(\mathcal{M})/\eta)/(1 - 2(\mathtt{M} + \eta\mathrm{diam}(\mathcal{M}))) \leq d/2 + 4(1 + \mathtt{M} + \mathrm{diam}(\mathcal{M})/\eta),$$

which concludes the proof. $\qquad\square$

**Lemma D.5.** *Assume **A1**, **A2** and **A3**. Assume that there exists $\delta > 0$ such that for any $k \in \{0, \ldots, K\}$, $\gamma_k\beta_{T-t_k}/\sigma_{T-t_k}^2 \leq \delta$. Assume that there exists $\eta > 0$ such that $A(\delta, \mathtt{M}, \eta, \mathrm{diam}(\mathcal{M})) > 0$ with*

$$\begin{aligned}
A(\delta, \mathtt{M}, \eta, \mathrm{diam}(\mathcal{M})) &= 2 - 2\delta - 32\delta(1 + \mathtt{M}^2) - 8\mathtt{M} - 4\eta\mathrm{diam}(\mathcal{M}), \\
B(\delta, \mathtt{M}, \eta, \mathrm{diam}(\mathcal{M})) &= 32\delta(\mathtt{M}^2 + \mathrm{diam}(\mathcal{M})^2) + 2(1 + \delta)(\mathrm{diam}(\mathcal{M})/\eta + \mathtt{M}) + 4d.
\end{aligned}$$

*Then, we have for any $k \in \{0, \ldots, K\}$*

$$\mathbb{E}[\|Y_k\|^2] \leq \mathtt{K} = d + B(\delta, \mathtt{M}, \eta, \mathrm{diam}(\mathcal{M}))(1/A(\delta, \mathtt{M}, \eta, \mathrm{diam}(\mathcal{M})) + \delta). \tag{44}$$

*In particular if $\mathtt{M} \leq 1/32$ and $\delta \leq 1/32$ then for any $k \in \{0, \ldots, K\}$*

$$\mathbb{E}[\|Y_k\|^2] \leq \mathtt{K}_0 = 5d + 320(1 + \mathrm{diam}(\mathcal{M}))^2.$$

*Proof.* Recall that using (5), we have that for any $k \in \{0, \ldots, K-1\}$

$$Y_{k+1} = Y_k + (\exp[\textstyle\int_{T-t_{k+1}}^{T-t_k} \beta_s \mathrm{d}s] - 1)(Y_k + 2\boldsymbol{s}(T - t_k, Y_k)) + (\exp[2\textstyle\int_{T-t_{k+1}}^{T-t_k} \beta_s \mathrm{d}s] - 1)^{1/2}Z_k, \tag{45}$$

For simplicity, we denote

$$\gamma_{1,k} = (\exp[\textstyle\int_{T-t_{k+1}}^{T-t_k} \beta_s \mathrm{d}s] - 1)/\beta_{T-t_k}, \qquad \gamma_{2,k} = (\exp[2\textstyle\int_{T-t_{k+1}}^{T-t_k} \beta_s \mathrm{d}s] - 1)/(2\beta_{T-t_k}).$$

Then, (45) can be rewritten for any $k \in \{0, \ldots, K-1\}$ as

$$Y_{k+1} = Y_k + \gamma_{1,k}\beta_{T-t_k}(Y_k + 2\boldsymbol{s}(T - t_k, Y_k)) + \sqrt{2\gamma_{2,k}\beta_{T-t_k}}Z_k. \tag{46}$$

In what follows, we denote $\bar{\gamma}_{1,k} = \gamma_{1,k}\beta_{T-t_k}$ and $\bar{\gamma}_{2,k} = \gamma_{2,k}\beta_{T-t_k}$. In addition, using that $\gamma_k\beta_{T-t_k} \leq \delta \leq 1/4$, we have that $\gamma_{1,k} \leq \gamma_{2,k} \leq 2\gamma_{1,k}$. Indeed, we have that for any $k \in \{0, \ldots, K-1\}$

$$\begin{aligned}
\gamma_{2,k}/\gamma_{1,k} &= (1/2)(\exp[\textstyle\int_{T-t_{k+1}}^{T-t_k} \beta_s \mathrm{d}s] + 1) \geq 1, \\
\gamma_{2,k}/\gamma_{1,k} &= (1/2)(\exp[\textstyle\int_{T-t_{k+1}}^{T-t_k} \beta_s \mathrm{d}s] + 1) \leq (1/2)(\exp[\gamma_k\beta_{T-t_k}] + 1) \leq 2,
\end{aligned}$$

Using Lemma C.1, we have that for any $t \in [0, T]$, $x_t \in \mathbb{R}^d$ and $\eta > 0$

$$
\begin{aligned}
\langle x_t, \boldsymbol{s}(t, x_t) \rangle &\leq -\|x_t\|^2/\sigma_t^2 + m_t \mathrm{diam}(\mathcal{M})\|x_t\|/\sigma_t^2 + \mathtt{M}(1 + \|x_t\|)\|x_t\|/\sigma_t^2 \\
&\leq (-1 + 2\mathtt{M} + \eta m_t \mathrm{diam}(\mathcal{M})) \|x_t\|^2/\sigma_t^2 + (m_t \mathrm{diam}(\mathcal{M})/\eta + \mathtt{M})/\sigma_t^2,
\end{aligned} \tag{47}
$$

where we have used that for any $a, b \geq 0$, $2ab \leq \eta a^2 + b^2/\eta$ in the last line. In addition, using Lemma C.1, for any $t \in [0, T]$ and $x_t \in \mathbb{R}^d$ we have

$$
\begin{aligned}
\|\boldsymbol{s}(t, x_t)\|^2 &\leq 2\|\boldsymbol{s}(t, x_t) - \nabla \log p_t(x_t)\|^2 + 2\|\nabla \log p_t(x_t)\|^2 \\
&\leq 4\mathtt{M}^2(1 + \|x_t\|^2)/\sigma_t^4 + 4\|x_t\|^2/\sigma_t^4 + 4m_t^2\mathrm{diam}(\mathcal{M})^2/\sigma_t^4 \\
&\leq 4(1 + \mathtt{M}^2)\|x_t\|^2/\sigma_t^4 + 4(\mathtt{M}^2 + m_t^2\mathrm{diam}(\mathcal{M})^2)/\sigma_t^4.
\end{aligned} \tag{48}
$$

Combining (46), (47) and (48) we have for any $k \in \{0, \ldots, K-1\}$

$$
\begin{aligned}
\mathbb{E}[\|Y_{k+1}\|^2] &= (1 + \bar{\gamma}_{1,k})^2\mathbb{E}[\|Y_k\|^2] + 4\bar{\gamma}_{1,k}^2\mathbb{E}[\|\boldsymbol{s}(T - t_k, Y_k)\|^2] \\
&\quad + 4\bar{\gamma}_{1,k}(1 + \bar{\gamma}_{1,k})\mathbb{E}[\langle Y_k, \boldsymbol{s}(T - t_k, Y_k)\rangle] + 2\bar{\gamma}_{2,k}d \\
&\leq (1 + 2\bar{\gamma}_{1,k} + \bar{\gamma}_{1,k}^2)\mathbb{E}[\|Y_k\|^2] + 16(\bar{\gamma}_{1,k}/\sigma_{T-t_k}^2)^2(1 + \mathtt{M}^2)\mathbb{E}[\|Y_k\|^2] \\
&\quad + 16(\bar{\gamma}_{1,k}/\sigma_{T-t_k}^2)^2(\mathtt{M}^2 + m_{T-t_k}^2\mathrm{diam}(\mathcal{M})^2) \\
&\quad + 4\bar{\gamma}_{1,k}(1 + \bar{\gamma}_{1,k})\mathbb{E}[\langle Y_k, \boldsymbol{s}(T - t_k, Y_k)\rangle] + 4\bar{\gamma}_{1,k}d \\
&\leq (1 + 2\bar{\gamma}_{1,k} + \bar{\gamma}_{1,k}^2)\mathbb{E}[\|Y_k\|^2] + 16(\bar{\gamma}_{1,k}/\sigma_{T-t_k}^2)^2(1 + \mathtt{M}^2)\mathbb{E}[\|Y_k\|^2] \\
&\quad + 16(\bar{\gamma}_{1,k}/\sigma_{T-t_k}^2)^2(\mathtt{M}^2 + m_{T-t_k}^2\mathrm{diam}(\mathcal{M})^2) \\
&\quad + 4(\bar{\gamma}_{1,k}/\sigma_{T-t_k}^2)(1 + \bar{\gamma}_{1,k})(-1 + 2\mathtt{M} + \eta m_{T-t_k}\mathrm{diam}(\mathcal{M}))\mathbb{E}[\|Y_k\|^2] \\
&\quad + (\bar{\gamma}_{1,k}/\sigma_{T-t_k}^2)(1 + \bar{\gamma}_{1,k})(m_{T-t_k}\mathrm{diam}(\mathcal{M})/\eta + \mathtt{M}) + 4\bar{\gamma}_{1,k}d.
\end{aligned}
$$

In what follows, we let $\delta_k = \bar{\gamma}_{1,k}/\sigma_{T-t_k}^2$. Using that for any $t \in [0, T]$, $m_t, \sigma_t \in [0, 1]$ we have

$$
\begin{aligned}
\mathbb{E}[\|Y_{k+1}\|^2] &\leq (1 + 2\delta_k + \delta_k^2)\mathbb{E}[\|Y_k\|^2] + 16\delta_k^2(1 + \mathtt{M}^2)\mathbb{E}[\|Y_k\|^2] \\
&\quad + 16\delta_k^2(\mathtt{M}^2 + \mathrm{diam}(\mathcal{M})^2) \\
&\quad + 4\delta_k(1 + \delta_k)(-1 + 2\mathtt{M} + \eta\mathrm{diam}(\mathcal{M}))\mathbb{E}[\|Y_k\|^2] \\
&\quad + \delta_k(1 + \delta_k)(\mathrm{diam}(\mathcal{M})/\eta + \mathtt{M}) + 4\delta_k d.
\end{aligned}
$$

Since $s \mapsto \beta_s$ is non-decreasing, that $\beta_{T-t_k}\gamma_k \leq \delta \leq 1/4$ and that for any $w \in [0, 1/2]$, $\mathrm{e}^w - 1 \leq 1 + 2w$, we have

$$
\exp[\textstyle\int_{T-t_{k+1}}^{T-t_k} \beta_s \mathrm{d}s] - 1 \leq \exp[\beta_{T-t_k}\gamma_k] - 1 \leq 2\beta_{T-t_k}\gamma_k.
$$

Therefore, we get that $\delta_k \leq 2\gamma_k\beta_{T-t_k}/\sigma_{T-t_k}^2 \leq 2\delta$. Since $\delta_k \leq 2\delta$ we have

$$
\begin{aligned}
\mathbb{E}[\|Y_{k+1}\|^2] &\leq (1 + 2\delta_k + 2\delta_k\delta)\mathbb{E}[\|Y_k\|^2] + 32\delta_k\delta(1 + \mathtt{M}^2)\mathbb{E}[\|Y_k\|^2] \\
&\quad + 4\delta_k(-1 + 2\mathtt{M} + \eta\mathrm{diam}(\mathcal{M}))\mathbb{E}[\|Y_k\|^2] \\
&\quad + 32\delta_k\delta(\mathtt{M}^2 + \mathrm{diam}(\mathcal{M})^2) \\
&\quad + 2\delta_k(1 + \delta)(\mathrm{diam}(\mathcal{M})/\eta + \mathtt{M}) + 4\delta_k d.
\end{aligned}
$$

Hence, we have that

$$
\begin{aligned}
\mathbb{E}[\|Y_{k+1}\|^2] &\leq (1 + \delta_k[-2 + 2\delta + 32\delta(1 + \mathtt{M}^2) + 8\mathtt{M} + 4\eta\mathrm{diam}(\mathcal{M})])\mathbb{E}[\|Y_k\|^2] \\
&\quad + \delta_k[32\delta(\mathtt{M}^2 + \mathrm{diam}(\mathcal{M})^2) + 2(1 + \delta)(\mathrm{diam}(\mathcal{M})/\eta + \mathtt{M}) + 4d].
\end{aligned}
$$

Denote $A = 2 - 2\delta - 32\delta(1 + \mathtt{M}^2) - 8\mathtt{M} - 4\eta\mathrm{diam}(\mathcal{M})$, $B = 32\delta(\mathtt{M}^2 + \mathrm{diam}(\mathcal{M})^2) + 2(1 + \delta)(\mathrm{diam}(\mathcal{M})/\eta + \mathtt{M}) + 4d$. Then, we have

$$
\mathbb{E}[\|Y_{k+1}\|^2] \leq (1 - \delta_k A)\mathbb{E}[\|Y_k\|^2] + \delta_k B.
$$

Hence, if $\mathbb{E}[\|Y_k\|^2] \geq B/A$ we have that $\mathbb{E}[\|Y_{k+1}\|^2] \leq \mathbb{E}[\|Y_k\|^2]$. In addition, if $\mathbb{E}[\|Y_k\|^2] \leq B/A$ then, $\mathbb{E}[\|Y_{k+1}\|^2] \leq B/A + \delta B$. Therefore, we conclude by recursion that for any $k \in \{0, \ldots, K\}$

$$\mathbb{E}[\|Y_k\|^2] \leq d + B(1/A + \delta),$$

which concludes the first part of the proof. If $\delta, \mathtt{M} \leq 1/32$ then, $A(\delta, \mathtt{M}, \eta, \mathrm{diam}(\mathcal{M})) \geq 1/2 - 4\eta\mathrm{diam}(\mathcal{M})$. We conclude upon setting $\eta = 1/(16\mathrm{diam}(\mathcal{M}))$. In that case $A(\delta, \mathtt{M}, \eta, \mathrm{diam}(\mathcal{M})) \geq 1/4$ and

$$B(\delta, \mathtt{M}, \eta, \mathrm{diam}(\mathcal{M})) \leq 32\delta(\mathtt{M}^2 + \mathrm{diam}(\mathcal{M})^2) + 2(1 + \delta)(16\mathrm{diam}(\mathcal{M}) + \mathtt{M}) + 4d$$
$$\leq 64(1 + \mathrm{diam}(\mathcal{M}) + \mathrm{diam}(\mathcal{M})^2) + 4d,$$

which concludes the proof. $\qquad\square$

Note that the same result holds for $(\bar{\mathbf{Y}}_t)_{t \in [0, t_K]}$.

**Lemma D.6.** *Assume* **A**1, **A**2 *and* **A**3. *In addition, assume that for any* $k \in \{0, \ldots, K-1\}$, $\gamma_k \beta_{T-t_k}/\sigma_{T-t_k}^2 \leq \delta \leq 1/4$. *Then, we have for any* $k \in \{0, \ldots, K-1\}$ *and* $t \in [t_k, t_{k+1}]$

$$\mathbb{E}[\|\bar{\mathbf{Y}}_t - \bar{\mathbf{Y}}_{t_k}\|^2] \leq \mathtt{L}\beta_{T-t_k}\gamma_k,$$

*with* $\mathtt{L} = 8(1 + \delta)(16(5 + \mathtt{M}^2)\mathtt{K} + 16(4\mathrm{diam}(\mathcal{M})^2 + \mathtt{M}^2)) + 4$ *and where* $\mathtt{K}$ *is defined in* (44). *In particular if* $\mathtt{M} \leq 1/32$ *and* $\delta \leq 1/32$

$$\mathbb{E}[\|\bar{\mathbf{Y}}_t - \bar{\mathbf{Y}}_{t_k}\|^2] \leq \mathtt{L}_0\beta_{T-t_k}\gamma_k = (64d + 20544(1 + \mathrm{diam}(\mathcal{M}))^2)\beta_{T-t_k}\gamma_k.$$

*Proof.* Recall that

$$\bar{\mathbf{Y}}_t = \bar{\mathbf{Y}}_{t_k} + (\exp[\textstyle\int_{T-t}^{T-t_k} \beta_s \mathrm{d}s] - 1)(\bar{\mathbf{Y}}_{t_k} + 2\boldsymbol{s}(T - t_k, \bar{\mathbf{Y}}_{t_k})) + (\exp[2\textstyle\int_{T-t}^{T-t_k} \beta_s \mathrm{d}s] - 1)^{1/2}Z,$$

where $Z$ is a Gaussian random variable with zero mean and identity covariance and the equality holds in distribution independent from $\bar{\mathbf{Y}}_{t_k}$. Therefore, we get that

$$\mathbb{E}[\|\bar{\mathbf{Y}}_t - \bar{\mathbf{Y}}_{t_k}\|^2] = 2(\exp[\textstyle\int_{T-t}^{T-t_k} \beta_s \mathrm{d}s] - 1)^2(\mathbb{E}[\|\bar{\mathbf{Y}}_{t_k}\|^2] + 4\mathbb{E}[\|\boldsymbol{s}(T - t_k, \bar{\mathbf{Y}}_{t_k})\|^2])$$
$$+ (\exp[2\textstyle\int_{T-t}^{T-t_k} \beta_s \mathrm{d}s] - 1)d. \tag{49}$$

In addition, using **A**3, Lemma C.1 and that for any $t \in [0, T]$, $m_t \in [0, 1]$, we get that for any $u \in [0, T]$ and $x_u \in \mathbb{R}^d$

$$\|\boldsymbol{s}(u, x_u)\| \leq \mathtt{M}(1 + \|x_u\|)/\sigma_u^2 + 2\|x_u\|/\sigma_u^2 + 2\mathrm{diam}(\mathcal{M})/\sigma_u^2$$
$$\leq (1/\sigma_u^2)\{(\mathtt{M} + 2)\|x_u\| + (\mathtt{M} + 2\mathrm{diam}(\mathcal{M}))\}.$$

Combining this result and (49), we get that

$$\mathbb{E}[\|\bar{\mathbf{Y}}_t - \bar{\mathbf{Y}}_{t_k}\|^2] = 2(\exp[\textstyle\int_{T-t}^{T-t_k} \beta_s \mathrm{d}s] - 1)^2(\mathbb{E}[\|\bar{\mathbf{Y}}_{t_k}\|^2]$$
$$+ 4\mathbb{E}[\|\boldsymbol{s}(T - t_k, \bar{\mathbf{Y}}_{t_k})\|^2]) + (\exp[2\textstyle\int_{T-t}^{T-t_k} \beta_s \mathrm{d}s] - 1)d$$
$$\leq 2(\exp[\textstyle\int_{T-t}^{T-t_k} \beta_s \mathrm{d}s] - 1)^2\mathbb{E}[\|\bar{\mathbf{Y}}_{t_k}\|^2]$$
$$+ 32(4 + \mathtt{M}^2)(\exp[\textstyle\int_{T-t}^{T-t_k} \beta_s \mathrm{d}s] - 1)^2\mathbb{E}[\|\bar{\mathbf{Y}}_{t_k}\|^2]/\sigma_{T-t_k}^2$$
$$+ 32(\exp[\textstyle\int_{T-t}^{T-t_k} \beta_s \mathrm{d}s] - 1)^2(4\mathrm{diam}(\mathcal{M})^2 + \mathtt{M}^2)/\sigma_{T-t_k}^2$$
$$+ (\exp[2\textstyle\int_{T-t}^{T-t_k} \beta_s \mathrm{d}s] - 1)d. \tag{50}$$

Since $s \mapsto \beta_s$ is non-decreasing, that $\beta_{T-t_k}\gamma_k \leq \delta \leq 1/4$ and that for any $w \in [0, 1/2]$, $\mathrm{e}^{2w} - 1 \leq 1 + 4w$, we have for any $\alpha \in \{1, 2\}$

$$\exp[\alpha \textstyle\int_{T-t}^{T-t_k} \beta_s \mathrm{d}s] - 1 \leq \exp[\alpha\beta_{T-t_k}\gamma_k] - 1 \leq 2\alpha\beta_{T-t_k}\gamma_k.$$

Combining this result and (51), we get that

$$
\begin{aligned}
\mathbb{E}[\|\bar{\mathbf{Y}}_t - \bar{\mathbf{Y}}_{t_k}\|^2] &\leq 8\beta_{T-t_k}^2 \gamma_k^2 \mathbb{E}[\|\bar{\mathbf{Y}}_{t_k}\|^2 + 32(4+\mathtt{M}^2)\|\bar{\mathbf{Y}}_{t_k}\|^2/\sigma_{T-t_k}^2 + 32(4\mathrm{diam}(\mathcal{M})^2 + \mathtt{M}^2)/\sigma_{T-t_k}^2] \\
&\quad + 4\beta_{T-t_k}\gamma_k d \\
&\leq 8(\beta_{T-t_k}^2 \gamma_k^2/\sigma_{T-t_k}^2)(32(5+\mathtt{M}^2)\mathbb{E}[\|\bar{\mathbf{Y}}_{t_k}\|^2] + 32(4\mathrm{diam}(\mathcal{M})^2 + \mathtt{M}^2)) + 4\beta_{T-t_k}\gamma_k d.
\end{aligned}
\tag{51}
$$

Therefore, using Lemma D.5 and (44), we get that

$$
\begin{aligned}
\mathbb{E}[\|\bar{\mathbf{Y}}_t - \bar{\mathbf{Y}}_{t_k}\|^2] &\leq 8(\beta_{T-t_k}^2 \gamma_k^2/\sigma_{T-t_k}^2)(32(5+\mathtt{M}^2)\mathtt{K} + 32(4\mathrm{diam}(\mathcal{M})^2 + \mathtt{M}^2)) + 4\beta_{T-t_k}\gamma_k d \\
&\leq \{256\delta((5+\mathtt{M}^2)\mathtt{K} + 4\mathrm{diam}(\mathcal{M})^2 + \mathtt{M}^2) + 4d\}\beta_{T-t_k}\gamma_k
\end{aligned}
$$

which concludes the first part of the proof. Now assuming that $\delta, \mathtt{M} \leq 1/32$ we have

$$
\begin{aligned}
\mathbb{E}[\|\bar{\mathbf{Y}}_t - \bar{\mathbf{Y}}_{t_k}\|^2] &\leq (64\mathtt{K} + 64\mathrm{diam}(\mathcal{M})^2 + 16d)\beta_{T-t_k}\gamma_k \\
&\leq 64d + 20544(1 + \mathrm{diam}(\mathcal{M}))^2,
\end{aligned}
$$

which concludes the proof. $\qquad\square$

## D.4  Control of the tangent backward process

We now introduce the tangent process associated with $(\mathbf{Y}_t)_{t\in[0,T]}$. We have

$$
\mathrm{d}\nabla\mathbf{Y}_t = \beta_{T-t}(\mathrm{Id} + 2\nabla^2 \log p_{T-t}(\mathbf{Y}_t))\nabla\mathbf{Y}_t \mathrm{d}t, \qquad \nabla\mathbf{Y}_0 = \mathrm{Id}.
\tag{52}
$$

We recall that controlling the tangent process allows to control the Wasserstein distance between the original process and its target measure. Indeed, let $(\mathbf{Y}_t^x)_{t\in[0,T]}$ and $(\mathbf{Y}_t^y)_{t\in[0,T]}$ be the processes given by (39) with initial condition $x$ and $y$ respectively. Then we have that for any $t \in [0,T]$

$$
\|\mathbf{Y}_t^x - \mathbf{Y}_t^y\| \leq \int_0^1 \|\nabla\mathbf{Y}_t^{z_\lambda}\|\mathrm{d}\lambda\|x-y\|,
\tag{53}
$$

where $(\nabla\mathbf{Y}_t^x)_{t\in[0,T]}$ is the tangent process given by (52) and associated with $(\mathbf{Y}_t^{z_\lambda})_{t\in[0,T]}$, where $z_\lambda = \lambda x + (1-\lambda)y$. Before providing controls in the general setting, we take a detour and focus on the case where $\mathrm{diam}(\mathcal{M}) = 0$, i.e. $\pi = \delta_0$. In the following proposition, we show that in this case the backward process converges in finite-time. This highlights the role of the diameter of the manifold in the subsequent analysis.

**Proposition D.7.** *Assume* **A1** *and that* $\mathrm{diam}(\mathcal{M}) = 0$, *i.e.* $\pi = \delta_0$. *Then, we have that for any* $x, y \in \mathbb{R}^d$ *and* $t \in [0,T]$

$$
\mathbf{W}_1(\delta_x\mathrm{Q}_t, \delta_y\mathrm{Q}_y) \leq 2\exp[(1/2)\textstyle\int_{T-t}^T \beta_s\mathrm{d}s](\exp[2\int_0^T \beta_s\mathrm{d}s] - 1)^{-1/2}(\exp[2\int_0^{T-t} \beta_s\mathrm{d}s] - 1)^{1/2}\|x-y\|.
$$

*In particular, we have that for any* $x \in \mathbb{R}^d$, $\delta_x\mathrm{Q}_T = \pi$, *i.e. the backward diffusion converges in finite time no matter the initialization distribution.*

*Proof.* Let $t \in [0,T]$. Using Lemma C.2, we have that for any $\mathtt{M} \in \mathcal{M}_d(\mathbb{R})$

$$
\langle\mathtt{M}, \nabla^2 \log q_t(x_t)\mathtt{M}\rangle \leq -\sigma_t^{-2}\|\mathtt{M}\|^2.
$$

In particular, we have that for any $\mathtt{M} \in \mathcal{M}_d(\mathbb{R})$

$$
\beta_{T-t}\langle\mathtt{M}, \mathrm{Id} + 2\nabla^2 \log q_{T-t}(x_t)\mathtt{M}\rangle \leq (\beta_{T-t} - 2\beta_{T-t}\sigma_{T-t}^{-2})\|\mathtt{M}\|^2.
\tag{54}
$$

Using Lemma D.1, we have

$$
\begin{aligned}
\textstyle\int_0^t (\beta_{T-s} - 2\beta_{T-s}\sigma_{T-s}^{-2})\mathrm{d}s \\
&= \textstyle\int_0^t \beta_{T-s}\mathrm{d}s + \log(\exp[2\int_0^{T-t}\beta_s\mathrm{d}s] - 1) - \log(\exp[2\int_0^T \beta_s\mathrm{d}s] - 1) \\
&= \textstyle\int_{T-t}^T \beta_s\mathrm{d}s + \log(\exp[2\int_0^{T-t}\beta_s\mathrm{d}s] - 1) - \log(\exp[2\int_0^T \beta_s\mathrm{d}s] - 1).
\end{aligned}
$$

Finally, we have that

$$\exp[\int_0^t (\beta_{T-s} - 2\beta_{T-s}\sigma_{T-s}^{-2})\mathrm{d}s]$$
$$\leq \exp[\int_{T-t}^T \beta_s \mathrm{d}s](\exp[2\int_0^T \beta_s \mathrm{d}s] - 1)^{-1}(\exp[2\int_0^{T-t} \beta_s \mathrm{d}s] - 1).$$

Hence, using this result, (52) and (54), we get that

$$(1/2)\|\nabla \mathbf{Y}_t\|^2 \leq \exp[\int_{T-t}^T \beta_s \mathrm{d}s](\exp[2\int_0^T \beta_s \mathrm{d}s] - 1)^{-1}(\exp[2\int_0^{T-t} \beta_s \mathrm{d}s] - 1),$$

which concludes the first part of the proof, using (53). For the second part of the proof, we first remark that for any $x, y \in \mathbb{R}^d$, $\mathbf{W}_1(\delta_x Q_T, \delta_y Q_T) = 0$. Therefore, for any probability measures $\mu, \nu$ such that $\int_{\mathbb{R}^d} \|x\|\mathrm{d}\mu(x) < +\infty$ and $\int_{\mathbb{R}^d} \|x\|\mathrm{d}\nu(x) < +\infty$, we have $\mathbf{W}_1(\mu Q_T, \nu Q_T) = 0$ We conclude upon combining this result and that $(\delta_x P_T)Q_T = \delta_x$. □

Note that it is also possible to explicitly write down the backward stochastic process in this case since $\nabla \log p_t$ is available in close form. One can remark that in this case we recover an Ornstein–Uhlenbeck bridge.

In the case where the diameter is non-zero, we cannot recover such a contraction. However, it is possible to obtain a contraction up to a certain point. The following lemma will allow us to control the growth of the tangent process.

**Lemma D.8.** *Assume* **A**2 *and that* $T \geq 2\bar{\beta}(1 + \log(1 + \mathrm{diam}(\mathcal{M})))$. *Let* $t^\star \in [0, T]$ *given by*

$$t^\star = T - 2\bar{\beta}(1 + \log(1 + \mathrm{diam}(\mathcal{M}))).$$

*Then, for any* $t \in [0, t^\star]$ *we have*

$$\int_0^t \beta_{T-s}(1 - 2/\sigma_{T-s}^2 + m_{T-s}^2 \mathrm{diam}(\mathcal{M})^2/\sigma_{T-s}^4)\mathrm{d}s \leq -(1/2)\int_0^t \beta_{T-t}\mathrm{d}s.$$

*In addition, for any* $t \in [t^\star, T]$

$$\int_{t^\star}^t \beta_{T-s}(1 - 2/\sigma_{T-s}^2 + m_{T-s}^2 \mathrm{diam}(\mathcal{M})^2/\sigma_{T-s}^4)\mathrm{d}s \leq (\mathrm{diam}(\mathcal{M})^2/2)(\sigma_{T-t}^{-2} - \sigma_{T-t^\star}^{-2}).$$

*Proof.* Let $s \in [0, T]$. Note that we have

$$1 - 2/\sigma_{T-s}^2 + m_{T-s}^2 \mathrm{diam}(\mathcal{M})^2/\sigma_{T-s}^4 \leq -1/2, \tag{55}$$

if and only if

$$3\sigma_{T-t}^4 - 4\sigma_{T-s}^2 + 2m_{T-s}^2 \mathrm{diam}(\mathcal{M})^2 \leq 0.$$

Hence, using this result and the fact that $\sigma_{T-s}^2 = 1 - m_{T-s}^2$ we have that (55) is satisfied if and only if

$$3m_{T-s}^4 + 2(\mathrm{diam}(\mathcal{M})^2 - 1)m_{T-s}^2 - 1 \leq 0.$$

Introduce $P(u) = 3u^2 + 2(\mathrm{diam}(\mathcal{M})^2 - 1)u - 1$. We have that $P(u) \leq 0$ for $u \in [0, u_0]$ with

$$u_0 = [-(\mathrm{diam}(\mathcal{M})^2 - 1) + ((\mathrm{diam}(\mathcal{M})^2 - 1)^2 + 3)^{1/2}]/3 = (\delta + (\delta^2 + 3)^{1/2})/3,$$

where $\delta = \mathrm{diam}(\mathcal{M})^2 - 1 \in [-1, +\infty)$. If $\mathrm{diam}(\mathcal{M})^2 - 1 \geq 1$ then, using this result and the fact that for any $a \in [0, +\infty)$, $(1 + a)^{1/2} \geq 1 + a/2 - a^2/8$ we have

$$u_0 \geq \delta(-1 + (1 + 3/\delta^2)^{1/2})/3 \geq \delta(1/(2\delta^2) - 3/(8\delta^4)) \geq (1/2 - 3/8)/\delta \geq 1/(8\delta). \tag{56}$$

In addition, if $\delta \leq 1$ then $\delta^2 \in [0, 1]$. Using this result and the fact that for any $a \in [0, 1]$, $\sqrt{3 + a} \geq (2 - \sqrt{3})a + \sqrt{3}$ we have

$$u_0 \geq (-\delta + (3 + \delta^2)^{1/2})/3$$
$$\geq (-|\delta| + (3 + |\delta|^2)^{1/2})/3 \geq ((1 - \sqrt{3})|\delta| + \sqrt{3})/3 \geq 1/3.$$

Combining this result and (56), we get that

$$u_0 \geq 1/(8(1+|\delta|)).$$

Therefore, we get that for any $t \in [0, T]$ such that $m_{T-t}^2 \leq 1/(8(1+|\delta|))$

$$1 - 2/\sigma_{T-s}^2 + m_{T-s}^2 \text{diam}(\mathcal{M})^2/\sigma_{T-s}^4 \leq -1/2.$$

Hence, for any $t \in [0, T]$ such that $\exp[-2(T-t)/\bar{\beta}] \leq 1/(8(1+|\delta|))$

$$1 - 2/\sigma_{T-s}^2 + m_{T-s}^2 \text{diam}(\mathcal{M})^2/\sigma_{T-s}^4 \leq -1/2.$$

Let $t_0^\star$ such that $\exp[-2(T-t_0^\star)/\bar{\beta}] = 1/(8(1+|\delta|))$. We get that

$$t_0^\star = T - (\bar{\beta}/2)\log(8(1+|\delta|)).$$

Using that for any $a \geq 0$, $1 + |a^2 - 1| \leq 2(1+a)^2$ and that $\log(16) \leq 4$, we get that

$$t_0^\star \geq T - (\bar{\beta}/2)(\log(16(1+\text{diam}(\mathcal{M}))^2)) \geq t^\star = T - 2\bar{\beta}(1+\log(1+\text{diam}(\mathcal{M}))).$$

Hence, since $t \mapsto m_{T-t}$ is non-decreasing, we get that for any $t \in [0, t^\star]$, $m_{T-t} \leq 1/(8(1+|\delta|))$ and therefore

$$1 - 2/\sigma_{T-s}^2 + m_{T-s}^2 \text{diam}(\mathcal{M})^2/\sigma_{T-s}^4 \leq -1/2,$$

which concludes the first part of the proof. The second part of the proof follows from Lemma D.1 and

$$\int_{t^\star}^t \beta_{T-s}(1 - 2/\sigma_{T-s}^2 + m_{T-s}^2 \text{diam}(\mathcal{M})^2/\sigma_{T-s}^4)\mathrm{d}s$$
$$\leq (\text{diam}(\mathcal{M})^2/2)[(1 - \exp[-2\int_0^{T-t}\beta_s\mathrm{d}s])^{-1} - (1 - \exp[-2\int_0^{T-t^\star}\beta_s\mathrm{d}s])^{-1}]$$
$$\leq (\text{diam}(\mathcal{M})^2/2)(\sigma_{T-t}^{-2} - \sigma_{T-t^\star}^{-2}),$$

which concludes the proof. $\qquad\square$

The control of the tangent process in the case where $\text{diam}(\mathcal{M}) > 0$ is given in Proposition 6.

**Proposition D.9.** *Assume* **A**1 *and* $T \geq 2\bar{\beta}(1+\log(1+\text{diam}(\mathcal{M}))$. *Then, for any* $x, y \in \mathbb{R}^d$ *and* $t \in [0, t_K]$

$$\mathbf{W}_1(\delta_x Q_t, \delta_y Q_t) \leq \exp[(\text{diam}(\mathcal{M})^2/2)\sigma_{T-t_K}^{-2}]\|x - y\|.$$

*Proof.* This is a direct consequence of (53) and Proposition 6. $\qquad\square$

# E A stochastic interpolation formula

In this section, we present a formula first introduced in Del Moral and Singh (2019) which is a stochastic extension of the Alekseev–Gröbner formula (Alekseev, 1961). We recall the definition of the stochastic flows $(\mathbf{Y}_{s,t}^x)_{s,t \in [0,T]}$ and the interpolation of its discretization $(\bar{\mathbf{Y}}_{s,t}^x)_{s,t \in [0,T]}$, for any $x \in \mathbb{R}^d$

$$\mathrm{d}\mathbf{Y}_{s,t}^x = \beta_{T-t}\{\mathbf{Y}_{s,t}^x + 2\nabla \log q_{T-t}(\mathbf{Y}_{s,t}^x)\}\mathrm{d}t + \sqrt{2\beta_{T-t}}\mathrm{d}\mathbf{B}_t, \qquad \mathbf{Y}_{s,s}^x = x.$$

and

$$\mathrm{d}\bar{\mathbf{Y}}_{s,t}^x = \beta_{T-t}\{\bar{\mathbf{Y}}_{s,t}^x + 2s(T - t_k, \bar{\mathbf{Y}}_{s,t_k}^x)\}\mathrm{d}t + \sqrt{2\beta_{T-t}}\mathrm{d}\mathbf{B}_t, \qquad \bar{\mathbf{Y}}_{s,s}^x = x.$$

The following proposition is a straightforward application of (Del Moral and Singh, 2019, Theorem 1.2). Note that these results apply since the drift and the volatility of the backward processes have bounded differential up to order three, see Lemma C.5.

**Proposition E.1.** *Assume* **A**1. *Then, for any* $s, t \in [0, T)$ *with* $s < t$ *and* $x \in \mathbb{R}^d$

$$\mathbf{Y}_{s,t}^x - \bar{\mathbf{Y}}_{s,t}^x = \int_s^t (\nabla \mathbf{Y}_{u,t}^{\bar{\mathbf{Y}}_{s,u}})^\top \Delta b_u((\bar{\mathbf{Y}}_{s,v})_{v \in [s,u]})\mathrm{d}u,$$

*where for any* $u \in [0, T)$ *such that* $u \in [t_k, t_{k+1})$ *for some* $k \in \{0, \ldots, K-1\}$ *and* $(\omega_v)_{v \in [s,u]} \in \mathrm{C}([s,u], \mathbb{R}^d)$ *we have*

$$b_u(\omega) = \beta_{T-u}(\omega_u + 2\nabla \log q_{T-u}(\omega_u)), \qquad \bar{b}_u(\omega) = \beta_{T-t_u}(\omega_u + 2s(T - t_k, \omega_{t_k})),$$
$$\Delta b_u(\omega) = b_u(\omega) - \bar{b}_u(\omega).$$

## F    Additional comments on Theorem 1

In this section, we discuss the validity of **A3** and then comment the suboptimality of the bound of Theorem 1.

### F.1    Validity of A3

First, we highlight that under **A1**, **A3** is satisfied if for any $t \in (0, T]$ and $x_t \in \mathbb{R}^d$ we have $\|\boldsymbol{s}(t, x_t) - \nabla \log p_t(x_t)\| \leq \mathtt{M}_r \|\nabla \log p_t(x_t)\|$ with $\mathtt{M}_r \geq 0$. Indeed, using this condition **A1**, Lemma C.1 and letting $\mathtt{M} = 4\mathtt{M}_r(1 + \mathrm{diam}(\mathcal{M}))$, we get that **A3** is satisfied. Hence, **A3** is implied by a control on the *relative* error between the score function and its approximation.

Assume that $\pi = \delta_0$. Then in that case $\nabla \log p_t = \nabla \log p_{t|0}(\cdot|0)$ and we can compute explicitly the error $\|\boldsymbol{s}(t, x_t) - \nabla \log p_t(x_t)\|$ at given query points $(t, x_t)$. In Figure 1, we analyze this error in a two-dimensional setting. In particular, we recover that the behavior of the error is explosive as $\|x\| \to +\infty$ and $t \to 0$.

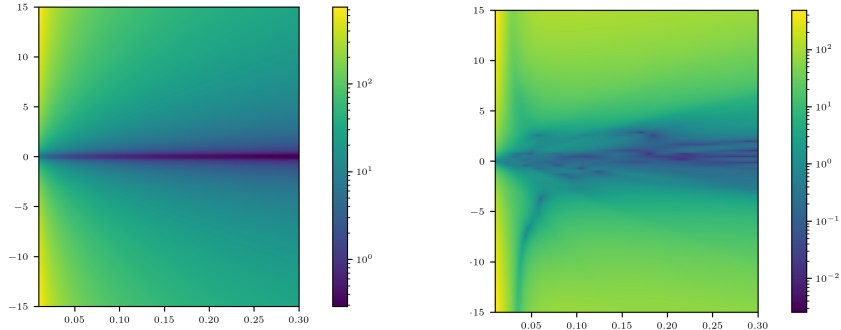

Figure 1: Target is a Dirac mass at 0. Left: norm of the true score $\|\nabla \log p_{t|0}(x)\|$ ($x$-axis: time evolution ($T = 1$), $y$-axis: spatial evolution (along the first coordinate, second coordinate is fixed to 0). Right: norm of the error between the estimated score and the true score $\|\boldsymbol{s}(t, x) - \nabla \log p_{t|0}(x)\|$ ($x$-axis: time evolution ($T = 1$), $y$-axis: spatial evolution (along the first coordinate, second coordinate is fixed to 0).

Finally, we conclude this study by illustrating the explosive behavior of the norm of the score in a two-dimensional setting (we restrict ourselves to this small dimensional setting so that we can get a dense grid of query points without encountering memory issues), see Figure 2. We emphasize that the norm of the score has a similar behavior as the error term, *i.e.* it is explosive as $\|x\| \to +\infty$ and $t \to 0$.

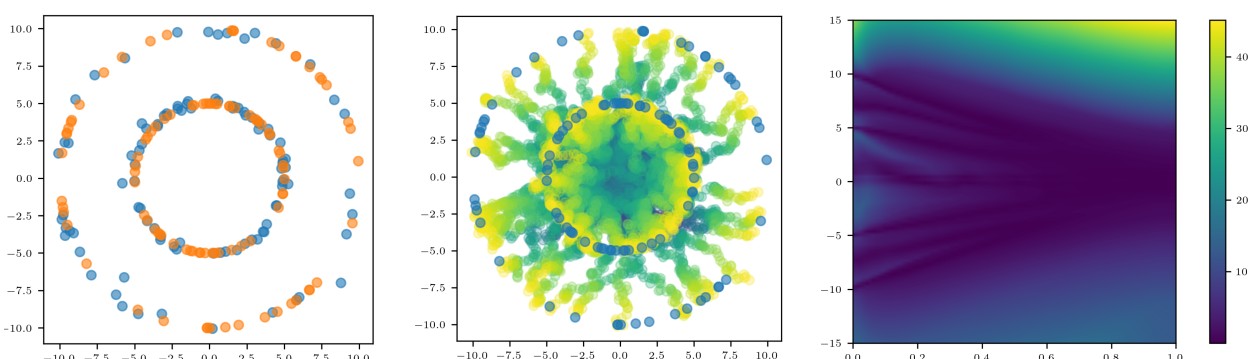

Figure 2: Target is the uniform distribution on two concentric circles. Left: samples from the target distribution (orange) and samples from the diffusion model (blue). Middle: trajectories of the diffusion model. Right: norm of the estimated score $\|\boldsymbol{s}(t, x)\|$ ($x$-axis: time evolution ($T = 1$), $y$-axis: spatial evolution (along the first coordinate, second coordinate is fixed to 0).

In both settings, the score function is learned by minimizing the Denoising Score Matching objective (3) using the ADAM optimizer. The architecture of the network and training settings are similar to the ones used in De Bortoli et al. (2021a).

### F.2 Suboptimality of the bound

Recall that the generative model is given by $\mathcal{L}(Y_K)$. In Theorem 1, we provide an upper bound on $\mathbf{W}_1(\mathcal{L}(Y_K), \pi)$. In this section, we compare the obtained bound with a bound on $\mathbf{W}_1(\mathcal{L}(Y_0), \pi)$. Recall that $Y_0 \sim \mathrm{N}(0, \mathrm{Id})$. Considering any coupling $(X, Y_0)$ such that $X \sim \pi$ and $Y_0 \sim \mathrm{N}(0, \mathrm{Id})$, we have

$$\mathbf{W}_1(\mathcal{L}(Y_0), \pi) \leq \mathbb{E}[\|X\|] + \mathbb{E}[\|Y_0\|] \leq \mathrm{diam}(\mathcal{M}) + \sqrt{d}.$$

This naive bound can be better than the one obtained in Theorem 1, especially for large values of $\mathtt{D}_0$. If that is the case then the derived bound seems to be vacuous at first sight since diffusing the process backward does *not* improve the Wasserstein distance of order one between the obtained model and the target measure. However, we argue that such cases are possible especially if we are using a poor estimation of the score, *i.e.* a large value of $\mathtt{M}$ in **A**3. Indeed, neglecting the discretization error and setting $\beta = 1$ for simplicity, the backward process is given by

$$\mathrm{d}\hat{\mathbf{Y}}_t = \{\hat{\mathbf{Y}}_t + 2\boldsymbol{s}(T - t, \hat{\mathbf{Y}}_t)\}\mathrm{d}t + \sqrt{2}\mathrm{d}\mathbf{B}_t, \qquad \hat{\mathbf{Y}}_0 \sim \pi_\infty = \mathrm{N}(0, \mathrm{Id}). \tag{57}$$

Assume that $\boldsymbol{s} = 0$ (which is approximately the case at initialization if we parameterize $\boldsymbol{s}$ with a neural network with a fully connected last layer and no non-linearity), then the dynamics becomes

$$\mathrm{d}\hat{\mathbf{Y}}_t = \hat{\mathbf{Y}}_t \mathrm{d}t + \sqrt{2}\mathrm{d}\mathbf{B}_t, \qquad \hat{\mathbf{Y}}_0 \sim \pi_\infty = \mathrm{N}(0, \mathrm{Id}). \tag{58}$$

In that case, for any $t \in [0, T]$, $\hat{\mathbf{Y}}_t$ is Gaussian and we have that $\mathcal{L}(\hat{\mathbf{Y}}_t) = \mathrm{N}(0, ((3\exp[2t] - 1)/2)\,\mathrm{Id})$. In addition, we have that for any $f : \mathbb{R}^d \to \mathbb{R}$ which is 1-Lipschitz we have

$$\mathbf{W}_1(\mathcal{L}(\hat{\mathbf{Y}}_T), \pi) \geq \mathbb{E}[f(\hat{\mathbf{Y}}_t)] - \int_{\mathbb{R}^d} f(x)\mathrm{d}\pi(x).$$

Choosing $f(x) = |x_1|$ for any $x = (x_1, \ldots, x_d) \in \mathbb{R}^d$ in the previous inequality we get,

$$\mathbf{W}_1(\mathcal{L}(\hat{\mathbf{Y}}_T), \pi) \geq (3\exp[2T] - 1)^{1/2} - \mathrm{diam}(\mathcal{M}).$$

Therefore, choosing $T \geq 0$ large enough and using (57), we get that $\mathbf{W}_1(\mathcal{L}(\hat{\mathbf{Y}}_T), \pi) \geq \mathbf{W}_1(\mathcal{L}(\hat{\mathbf{Y}}_0), \pi)$. This result implies that even in idealized setting, the backward process might steer the Gaussian distribution *away* from the target distribution $\pi$. This is due to the *explosive* property of the Ornstein-Uhlenbeck process (58) which should be compared to the *contractive* behavior of the forward Ornstein-Uhlenbeck process (1).

## G  Assumptions on the schedule

In what follows, we consider three schedules commonly used in practice: (a) the constant schedule, (b) the linear schedule, (c) the cosine schedule. We show that **A**2 is satisfied in all these cases. We consider a generalized version of the cosine schedule which makes it differentiable by replacing the hard clamping by a soft version with level $r > 0$ (note that letting $r \to 0$ we recover the original cosine schedule). The constant schedule is defined by $\beta_s = \beta_0$ for all $s \in [0, T]$. The linear schedule was introduced in Ho et al. (2020) and is defined by $\beta_s = \beta_0 + (\beta_T - \beta_0)t/T$ with $\beta_T > \beta_0 > 0$. Finally, the cosine schedule was introduced in (Nichol and Dhariwal, 2021, Equation (17)) in discrete-time and can be defined as follows in continuous-time

$$\beta_t = \mathrm{softmin}_r(1, \lim_{h \to 0}(\bar{\alpha}_{t-h} - \bar{\alpha}_t)/(\bar{\alpha}_{t-h}h)) = \mathrm{softmin}_r(1, f)_t, \qquad f(t) = -\bar{\alpha}'_t/\bar{\alpha}_t,$$

with $\bar{\alpha}$ defined as

$$\bar{\alpha}_t = \cos((t/T + \eta)/(1 + \eta)(\pi/2))^2 / \cos(\eta/(1 + \eta)(\pi/2))^2,$$

and where $\eta \geq 0$, $r > 0$ are parameters and for any $f_1, f_2 : [0, T] \to \mathbb{R}_+$

$$\mathrm{softmin}_r(f_1, f_2)_t = -r\log(\exp[-f_1(t)/r] + \exp[-f_2(t)/r]).$$

In the special case where $f_1 = 1$ we have

$$\text{softmin}_r(1, f_2)_t = 1 - r\log(1 + \exp[(1 - f_2(t))/r]),$$

We have

$$\bar{\alpha}'(t)/\bar{\alpha}(t) = -\pi/(T(1 + \eta))\tan((t/T + \eta)/(1 + \eta)(\pi/2)).$$

In particular, $t \mapsto \beta_t$ is increasing and bounded above and below on $[0, T]$.

Finally, we end this section by remarking that if one aims at studying the Euler-Maruyama discretization of the approximate backward, *i.e.* the process given by (26) then one needs to also assume some Lipschitz property on the schedule $s \mapsto \beta_s$.

## H  A short proof of the results of Franzese et al. (2022)

In (Franzese et al., 2022, Equation (9)), the authors show that (under mild regularity assumptions[9])

$$\int_{\mathbb{R}^d} \log p_{\theta,T}(x)p_0(x)\mathrm{d}x \geq \int_{\mathbb{R}^d} \log p_0(x)p_0(x)\mathrm{d}x - \mathcal{G}(\boldsymbol{s}_\theta, T) - \mathrm{KL}(p_T|p_\infty). \tag{59}$$

To do so, they rearrange the ELBO result from Huang et al. (2021). We have that (59) is equivalent to

$$\mathrm{KL}(p_0|p_{\theta,T}) \leq \mathcal{G}(\boldsymbol{s}_\theta, T) + \mathrm{KL}(p_T|p_\infty). \tag{60}$$

The definition of $\mathcal{G}(\boldsymbol{s}_\theta, T)$ is given by

$$\mathcal{G}(\boldsymbol{s}_\theta, T) = (1/2)(\int_0^T \beta_t^2 \mathbb{E}[\|\boldsymbol{s}_\theta(t, \mathbf{X}_t) - \nabla\log p_{t|0}(\mathbf{X}_t|\mathbf{X}_0)\|^2]\mathrm{d}t$$
$$- \int_0^T \beta_t^2 \mathbb{E}[\|\nabla\log p_t(\mathbf{X}_t) - \nabla\log p_{t|0}(\mathbf{X}_t|\mathbf{X}_0)\|^2]).$$

Developing the square and using that $\mathbb{E}[\nabla\log p_{t|0}(\mathbf{X}_t|\mathbf{X}_0)|\mathbf{X}_t] = \nabla\log p_t(\mathbf{X}_t)$, we get that

$$\mathcal{G}(\boldsymbol{s}_\theta, T) = \int_0^T \beta_t^2 \mathbb{E}[\|\boldsymbol{s}_\theta(t, \mathbf{X}_t) - \nabla\log p_t(\mathbf{X}_t)\|^2]\mathrm{d}t.$$

Hence, combining this result and (60), we have that (59) is equivalent to (Song et al., 2021a, Theorem 1) which is obtained upon combining the data-processing inequality, the decomposition of the Kullback-Leibler via conditioning and the Girsanov theorem.

## I  Wasserstein controls under $\mathrm{L}^2$ errors

In this section, we replace the assumption **A3** by the following weaker control.

**A5.** *There exist $\boldsymbol{s} \in \mathrm{C}([0, T] \times \mathbb{R}^d, \mathbb{R}^d)$ and $\mathtt{M} \geq 0$ such that for any $k \in \{0, \ldots, K\}$ and $x_t \in \mathbb{R}^d$,*

$$\mathbb{E}[\|\boldsymbol{s}(T - t_k, Y_k) - \nabla\log p_{T-t_k}(Y_k)\|^2] \leq \mathtt{M}^2 \mathbb{E}[(1 + \|Y_k\|^2)]/\sigma_{T-t_k}^4,$$

*where we recall that $(Y_k)_{k \in \{0,\ldots,K\}}$ is given by (5).*

Note that this assumption is different from the one of Lee et al. (2022) as the expectation is considered w.r.t. $\{Y_k\}_{k=0}^K$ and not $\{\mathbf{Y}_{t_k}\}_{k=0}^K$. In order to control the $\mathrm{L}^2$ error, Lee et al. (2022) use a change of measure and control the $\chi^2$ divergence between the density of $Y_k$ and the one $\mathbf{Y}_{t_k}$ for any $k \in \{0, \ldots, K\}$. These controls are obtained using a logarithmic Sobolev assumption on the target measure $\pi$. Adapting these results to our Wasserstein distance setting is not straightforward and is left for future work. Under **A5**, we have the following theorem, which is an extension of Theorem 1. To prove this theorem, we extend (Lee et al., 2022, Theorem 4.1) to the Wasserstein distance of order one and weaker growth conditions.

---

[9]We assume that all probability measures admit densities w.r.t. the Lebesgue measure and that all the integrals we consider are well-defined.

**Theorem I.1.** *Assume* **A**1, **A**2, **A**4, **A**5 *that* $T \geq 2\bar{\beta}(1+\log(1+\mathrm{diam}(\mathcal{M})))$, $t_K = T-\varepsilon$ *and* $\varepsilon, \mathtt{M}, \mathtt{M}/\zeta, \delta \leq 1/32$. *Then, there exists* $\mathtt{D}_0 \geq 0$ *such that*

$$\mathbf{W}_1(\mathcal{L}(Y_K), \pi) \leq \mathtt{D}_0(K\zeta + \exp[\kappa/\varepsilon](\mathtt{M}/\zeta + \delta^{1/2})/\varepsilon^2 + \exp[\kappa/\varepsilon]\exp[-T/\bar{\beta}] + \varepsilon^{1/2}),$$

*with* $\kappa = \mathrm{diam}(\mathcal{M})^2(1+\bar{\beta})/2$ *and*

$$\mathtt{D}_0 = D(1+\bar{\beta})^5(1+d+\mathrm{diam}(\mathcal{M})^4)(1+\log(1+\mathrm{diam}(\mathcal{M}))),$$

*and* $D$ *is a numerical constant.*

We start with the following lemma, which is an extension of Lemma D.5 to the setting where **A**3 is replaced by **A**5.

**Lemma I.2.** *Assume* **A**1, **A**2 *and* **A**5. *Assume that there exists* $\delta > 0$ *such that for any* $k \in \{0, \ldots, K\}$, $\gamma_k\beta_{T-t_k}/\sigma_{T-t_k}^2 \leq \delta$. *Assume that there exists* $\eta > 0$ *such that* $A(\delta, \mathtt{M}, \eta, \mathrm{diam}(\mathcal{M})) > 0$ *with*

$$A(\delta, \mathtt{M}, \eta, \mathrm{diam}(\mathcal{M})) = 2 - 2\delta - 32\delta(1+\mathtt{M}^2) - 8\mathtt{M} - 4\eta\mathrm{diam}(\mathcal{M}),$$
$$B(\delta, \mathtt{M}, \eta, \mathrm{diam}(\mathcal{M})) = 32\delta(\mathtt{M}^2 + \mathrm{diam}(\mathcal{M})^2) + 2(1+\delta)(\mathrm{diam}(\mathcal{M})/\eta + \mathtt{M}) + 4d.$$

*Then, we have for any* $k \in \{0, \ldots, K\}$

$$\mathbb{E}[\|Y_k\|^2] \leq \mathtt{K} = d + B(\delta, \mathtt{M}, \eta, \mathrm{diam}(\mathcal{M}))(1/A(\delta, \mathtt{M}, \eta, \mathrm{diam}(\mathcal{M})) + \delta).$$

*In particular if* $\mathtt{M} \leq 1/32$ *and* $\delta \leq 1/32$ *then for any* $k \in \{0, \ldots, K\}$

$$\mathbb{E}[\|Y_k\|^2] \leq \mathtt{K}_0 = 5d + 320(1 + \mathrm{diam}(\mathcal{M}))^2.$$

*Proof.* Recall that using (5), we have that for any $k \in \{0, \ldots, K-1\}$

$$Y_{k+1} = Y_k + (\exp[\int_{T-t_{k+1}}^{T-t_k}\beta_s\mathrm{d}s] - 1)(Y_k + 2\boldsymbol{s}(T-t_k, Y_k)) + (\exp[2\int_{T-t_{k+1}}^{T-t_k}\beta_s\mathrm{d}s] - 1)^{1/2}Z_k, \qquad (61)$$

For simplicity, we denote

$$\gamma_{1,k} = (\exp[\int_{T-t_{k+1}}^{T-t_k}\beta_s\mathrm{d}s] - 1)/\beta_{T-t_k}, \qquad \gamma_{2,k} = (\exp[2\int_{T-t_{k+1}}^{T-t_k}\beta_s\mathrm{d}s] - 1)/(2\beta_{T-t_k}).$$

Then, (61) can be rewritten for any $k \in \{0, \ldots, K-1\}$ as

$$Y_{k+1} = Y_k + \gamma_{1,k}\beta_{T-t_k}(Y_k + 2\boldsymbol{s}(T-t_k, Y_k)) + \sqrt{2\gamma_{2,k}\beta_{T-t_k}}Z_k. \qquad (62)$$

In what follows, we denote $\bar{\gamma}_{1,k} = \gamma_{1,k}\beta_{T-t_k}$ and $\bar{\gamma}_{2,k} = \gamma_{2,k}\beta_{T-t_k}$. In addition, using that $\gamma_k\beta_{T-t_k} \leq \delta \leq 1/4$, we have that $\gamma_{1,k} \leq \gamma_{2,k} \leq 2\gamma_{1,k}$. Indeed, we have that for any $k \in \{0, \ldots, K-1\}$

$$\gamma_{2,k}/\gamma_{1,k} = (1/2)(\exp[\int_{T-t_{k+1}}^{T-t_k}\beta_s\mathrm{d}s] + 1) \geq 1,$$
$$\gamma_{2,k}/\gamma_{1,k} = (1/2)(\exp[\int_{T-t_{k+1}}^{T-t_k}\beta_s\mathrm{d}s] + 1) \leq (1/2)(\exp[\gamma_k\beta_{T-t_k}] + 1) \leq 2,$$

In what follows, for any $t \in [0, T]$ and $x_t \in \mathbb{R}^d$, we denote $\Delta_t = \|\boldsymbol{s}(t, x_t) - \nabla\log p_t(x_t)\|$. Using Lemma C.1, we have that for any $t \in [0, T]$, $x_t \in \mathbb{R}^d$ and $\eta > 0$

$$\langle x_t, \boldsymbol{s}(t, x_t)\rangle \leq -\|x_t\|^2/\sigma_t^2 + m_t\mathrm{diam}(\mathcal{M})\|x_t\|/\sigma_t^2 + \Delta_t(x_t)\|x_t\|$$
$$\leq (-1 + \eta m_t\mathrm{diam}(\mathcal{M}))\|x_t\|^2/\sigma_t^2 + (m_t\mathrm{diam}(\mathcal{M})/\eta)/\sigma_t^2 + \Delta_t(x_t)\|x_t\|, \qquad (63)$$

where we have used that for any $a, b \geq 0$, $2ab \leq \eta a^2 + b^2/\eta$ in the last line. In addition, using Lemma C.1, for any $t \in [0, T]$ and $x_t \in \mathbb{R}^d$ we have

$$\|\boldsymbol{s}(t, x_t)\|^2 \leq 2\|\boldsymbol{s}(t, x_t) - \nabla\log p_t(x_t)\|^2 + 2\|\nabla\log p_t(x_t)\|^2$$
$$\leq 2\Delta_t(x_t)^2 + 4\|x_t\|^2/\sigma_t^4 + 4m_t^2\mathrm{diam}(\mathcal{M})^2/\sigma_t^4, \qquad (64)$$

In addition, using **A**5, the Cauchy-Schwarz inequality and (63) we have

$$
\begin{aligned}
\mathbb{E}[\langle Y_k, \boldsymbol{s}(T-t_k, Y_k)\rangle] &\leq (-1 + \eta m_{T-t_k}\mathrm{diam}(\mathcal{M}))\mathbb{E}[\|Y_k\|^2]/\sigma_{T-t_k}^2 + (m_{T-t_k}\mathrm{diam}(\mathcal{M})/\eta)/\sigma_{T-t_k}^2 \\
&\quad + \mathbb{E}[\Delta_{T-t_k}(Y_k)\|Y_k\|] \\
&\leq (-1 + \eta m_{T-t_k}\mathrm{diam}(\mathcal{M}))\mathbb{E}[\|Y_k\|^2]/\sigma_{T-t_k}^2 + (m_{T-t_k}\mathrm{diam}(\mathcal{M})/\eta)/\sigma_{T-t_k}^2 \\
&\quad + \mathbb{E}^{1/2}[\Delta_{T-t_k}(Y_k)^2]\mathbb{E}^{1/2}[\|Y_k\|^2] \\
&\leq (-1 + \eta m_{T-t_k}\mathrm{diam}(\mathcal{M}))\mathbb{E}[\|Y_k\|^2]/\sigma_{T-t_k}^2 + (m_{T-t_k}\mathrm{diam}(\mathcal{M})/\eta)/\sigma_{T-t_k}^2 \\
&\quad + \mathtt{M}(1 + \mathbb{E}^{1/2}[\|Y_k\|^2])\mathbb{E}^{1/2}[\|Y_k\|^2]/\sigma_{T-t_k}^2 \\
&\leq (-1 + \eta m_{T-t_k}\mathrm{diam}(\mathcal{M}) + \mathtt{M})\mathbb{E}[\|Y_k\|^2]/\sigma_{T-t_k}^2 + (m_{T-t_k}\mathrm{diam}(\mathcal{M})/\eta)/\sigma_{T-t_k}^2 + \mathtt{M}\mathbb{E}^{1/2}[\|Y_k\|^2]/\sigma_{T-t_k}^2 \\
&\leq (-1 + \eta m_{T-t_k}\mathrm{diam}(\mathcal{M}) + 2\mathtt{M})\mathbb{E}[\|Y_k\|^2]/\sigma_{T-t_k}^2 + (m_{T-t_k}\mathrm{diam}(\mathcal{M})/\eta + \mathtt{M})/\sigma_{T-t_k}^2. \quad (65)
\end{aligned}
$$

Finally, using **A**5 and (64) we have

$$
\begin{aligned}
\mathbb{E}[\|\boldsymbol{s}(T-t_k, Y_k)\|^2] &\leq 4\mathbb{E}[\Delta_{T-t_k}(Y_k)^2] + 4\mathbb{E}[\|Y_k\|^2]/\sigma_{T-t_k}^4 + 4m_{T-t_k}^2\mathrm{diam}(\mathcal{M})^2/\sigma_{T-t_k}^4 \\
&\leq 4\mathtt{M}^2(1 + \mathbb{E}[\|Y_k\|^2])/\sigma_{T-t_k}^4 + 4\mathbb{E}[\|Y_k\|^2]/\sigma_{T-t_k}^4 + 4m_{T-t_k}^2\mathrm{diam}(\mathcal{M})^2/\sigma_{T-t_k}^4 \\
&\leq 4(1 + \mathtt{M}^2)\mathbb{E}[\|Y_k\|^2])/\sigma_{T-t_k}^4 + 4(\mathtt{M}^2 + m_{T-t_k}^2\mathrm{diam}(\mathcal{M})^2)/\sigma_{T-t_k}^4. \quad (66)
\end{aligned}
$$

Combining (62), (65) and (66) we have for any $k \in \{0, \ldots, K-1\}$

$$
\begin{aligned}
\mathbb{E}[\|Y_{k+1}\|^2] &= (1 + \bar{\gamma}_{1,k})^2\mathbb{E}[\|Y_k\|^2] + 4\bar{\gamma}_{1,k}^2\mathbb{E}[\|\boldsymbol{s}(T-t_k, Y_k)\|^2] \\
&\quad + 4\bar{\gamma}_{1,k}(1 + \bar{\gamma}_{1,k})\mathbb{E}[\langle Y_k, \boldsymbol{s}(T-t_k, Y_k)\rangle] + 2\bar{\gamma}_{2,k}d \\
&\leq (1 + 2\bar{\gamma}_{1,k} + \bar{\gamma}_{1,k}^2)\mathbb{E}[\|Y_k\|^2] + 16(\bar{\gamma}_{1,k}/\sigma_{T-t_k}^2)^2(1 + \mathtt{M}^2)\mathbb{E}[\|Y_k\|^2] \\
&\quad + 16(\bar{\gamma}_{1,k}/\sigma_{T-t_k}^2)^2(\mathtt{M}^2 + m_{T-t_k}^2\mathrm{diam}(\mathcal{M})^2) \\
&\quad + 4\bar{\gamma}_{1,k}(1 + \bar{\gamma}_{1,k})\mathbb{E}[\langle Y_k, \boldsymbol{s}(T-t_k, Y_k)\rangle] + 4\bar{\gamma}_{1,k}d \\
&\leq (1 + 2\bar{\gamma}_{1,k} + \bar{\gamma}_{1,k}^2)\mathbb{E}[\|Y_k\|^2] + 16(\bar{\gamma}_{1,k}/\sigma_{T-t_k}^2)^2(1 + \mathtt{M}^2)\mathbb{E}[\|Y_k\|^2] \\
&\quad + 16(\bar{\gamma}_{1,k}/\sigma_{T-t_k}^2)^2(\mathtt{M}^2 + m_{T-t_k}^2\mathrm{diam}(\mathcal{M})^2) \\
&\quad + 4(\bar{\gamma}_{1,k}/\sigma_{T-t_k}^2)(1 + \bar{\gamma}_{1,k})(-1 + 2\mathtt{M} + \eta m_{T-t_k}\mathrm{diam}(\mathcal{M}))\mathbb{E}[\|Y_k\|^2] \\
&\quad + (\bar{\gamma}_{1,k}/\sigma_{T-t_k}^2)(1 + \bar{\gamma}_{1,k})(m_{T-t_k}\mathrm{diam}(\mathcal{M})/\eta + \mathtt{M}) + 4\bar{\gamma}_{1,k}d.
\end{aligned}
$$

The rest of the proof is identical to the one of Lemma D.5. $\qquad \square$

Let $\zeta > 0$. For any $k \in \{0, \ldots, K\}$ we define $\mathsf{A}_k$ such that

$$
\mathsf{A}_k = \{y \in \mathbb{R}^d : \|\boldsymbol{s}(T-t_k, y) - \nabla \log p_{T-t_k}(y)\| > (\mathtt{M}/\zeta)/\sigma_{T-t_k}^2\}.
$$

We define the process $(Y_k^\star)_{k\in\{0,\ldots,K\}}$ such that $Y_0^\star = Y_0$ and for any $k \in \{0, \ldots, K-1\}$, if $Y_k = Y_k^\star$ and $Y_k \in \mathsf{A}_k$ then $Y_{k+1}^\star = Y_{k+1}$. Otherwise, we define

$$
Y_{k+1}^\star = Y_k^\star + (\exp[\textstyle\int_{T-t_{k+1}}^{T-t_k} \beta_s \mathrm{d}s] - 1)(Y_k^\star + 2\nabla \log p_{T-t_k}(Y_k^\star)) + (\exp[2\textstyle\int_{T-t_{k+1}}^{T-t_k} \beta_s \mathrm{d}s] - 1)^{1/2}Z_k.
$$

This is similar to assuming that there exists $\boldsymbol{s}^\star$ such that for any $k \in \{0, \ldots, K-1\}$

$$
Y_{k+1}^\star = Y_k^\star + (\exp[\textstyle\int_{T-t_{k+1}}^{T-t_k} \beta_s \mathrm{d}s] - 1)(Y_k^\star + 2\boldsymbol{s}^\star(T-t_k, Y_k^\star)) + (\exp[2\textstyle\int_{T-t_{k+1}}^{T-t_k} \beta_s \mathrm{d}s] - 1)^{1/2}Z_k,
$$

with $\boldsymbol{s}^\star$ which satisfies[10] **A**3 with $\mathtt{M}$ replaced by $\mathtt{M}/\zeta$ and while $Y_k \in \mathsf{A}_k$, $Y_{k+1}^\star = Y_{k+1}$.

We have the following lemma which is an extension of (Lee et al., 2022, Theorem 4.1) to the Wasserstein setting. Note that contrary to (Lee et al., 2022, Theorem 4.1) which states results in total variation we also need control on the moments of the distribution under a $L^2$ error, which is precisely Lemma I.2.

---

[10]Note that we slightly abuse since $\boldsymbol{s}^\star$ is random (depending on the behavior of $Y_k$) but one can check that all our proofs remain unchanged in this slightly larger setting

**Lemma I.3.** *Assume* **A**1*,* **A**2 *and* **A**5*. Assume that there exists* $\delta > 0$ *such that for any* $k \in \{0, \ldots, K\}$*,* $\gamma_k \beta_{T-t_k}/\sigma_{T-t_k}^2 \leq \delta$ *and that* $\mathtt{M}, \mathtt{M}/\zeta, \delta \leq 1/32$*. Then, we have for any* $k \in \{0, \ldots, K\}$

$$\mathbb{E}[\|Y_k^\star - Y_k\|] \leq 4(1 + \mathtt{K}_0)\zeta k,$$

*where* $\mathtt{K}_0$ *is defined in Lemma I.2.*

*Proof.* Using the Cauchy-Schwarz inequality we have

$$\begin{aligned}
\mathbb{E}[\|Y_k^\star - Y_k\|] &= \mathbb{E}[\|Y_k^\star - Y_k\|\mathbb{1}_{Y_k \neq Y_k^\star}] \\
&\leq \sqrt{2}(\mathbb{E}^{1/2}[\|Y_k^\star\|^2] + \mathbb{E}^{1/2}[\|Y_k\|^2])(\textstyle\sum_{j=1}^k \mathbb{P}(Y_j \in \mathsf{A}_j))^{1/2} \\
&\leq \sqrt{2}(\mathbb{E}^{1/2}[\|Y_k^\star\|^2] + \mathbb{E}^{1/2}[\|Y_k\|^2]) \\
&\quad \times (\textstyle\sum_{j=1}^k \mathbb{P}(\|\boldsymbol{s}(T - t_j, Y_j) - \nabla \log p_{T-t_j}(Y_j)\| > (\mathtt{M}/\zeta)/\sigma_{T-t_j}^2))^{1/2}.
\end{aligned} \tag{67}$$

Using the Markov inequality, we have for any $j \in \{0, \ldots, K\}$

$$\begin{aligned}
\mathbb{P}(\|\boldsymbol{s}(T - t_j) &- \nabla \log p_{T-t_j}(Y_j)\| > (\mathtt{M}/\zeta)/\sigma_{T-t_j}^2) \\
&\leq \mathbb{E}[\|\boldsymbol{s}(T - t_j, Y_j) - \nabla \log p_{T-t_j}(Y_j)\|^2]\sigma_{T-t_j}^4 \zeta^2/\mathtt{M}^2 \leq \zeta^2 \mathbb{E}[1 + \|Y_j\|^2].
\end{aligned}$$

Therefore, combining this result, (67) and Lemma I.2 we have $\mathbb{E}[\|Y_k^\star - Y_k\|] \leq 4(1 + \mathtt{K}_0)\zeta k$. $\qquad\square$

We are now ready to complete the proof of Theorem I.1

*Proof.* We have

$$\mathbf{W}_1(\mathcal{L}(Y_K), \pi) \leq \mathbf{W}_1(\mathcal{L}(Y_K), \mathcal{L}(Y_k^\star)) + \mathbf{W}_1(\mathcal{L}(Y_k^\star), \pi). \tag{68}$$

Note that using Theorem 1 we have

$$\mathbf{W}_1(\mathcal{L}(Y_k^\star), \pi) \leq \mathtt{D}_0(\exp[\kappa/\varepsilon](\mathtt{M}/\zeta + \delta^{1/2})/\varepsilon^2 + \exp[\kappa/\varepsilon]\exp[-T/\bar{\beta}] + \varepsilon^{1/2}). \tag{69}$$

In addition, using Lemma I.3 we have

$$\mathbf{W}_1(\mathcal{L}(Y_K), \mathcal{L}(Y_k^\star)) \leq 4(1 + \mathtt{K}_0)\zeta K.$$

Combining this result and (69) in (68) concludes the proof. $\qquad\square$

## J  Improved bounds under Hessian conditions

In Appendix J.1, we prove Theorem 3 which is an improvement upon Theorem 1 under tighter conditions on the Hessian $\nabla^2 \log p_t$. In Appendix J.2, we show that this condition is satisfied in the case of a uniform measure over $[-1/2, 1/2]^p$ for some $p \in \{1, \ldots, d\}$. Finally, in Appendix J.3, we show, under appropriate smoothness conditions, that the condition is never satisfied on non-convex sets.

### J.1  Proof of Theorem 3

In this section, we prove Theorem 3. We start by deriving an improvement on Proposition 6. The main difference between Proposition 6 and Proposition J.1 lies into the dependency w.r.t. $\sigma_{T-t_K}^{-2}$. In Proposition 6, we have an exponential dependency $\exp[(\mathrm{diam}(\mathcal{M})^2/2)\sigma_{T-t_K}^{-2}]$ whereas in Proposition J.1, we have a polynomial dependency $\sigma_{T-t_K}^{-2\Gamma}$.

For ease of notation we introduce the following assumption.

**A 6.** *There exists* $\Gamma \geq 0$ *such that for any* $t \in (0, T]$ *and* $x_t \in \mathbb{R}^d$*,* $\|\nabla^2 \log p_t(x_t)\| \leq \Gamma/\sigma_t^2$*.*

We start with the following proposition.

**Proposition J.1.** *Assume* **A**1*,* **A**6 *and that* $T \geq 2\bar{\beta}(1 + \log(1 + \operatorname{diam}(\mathcal{M})))$*. Let* $t_K \in [0, T)$*. Then, for any* $t \in [0, t_K]$ *and* $x \in \mathbb{R}^d$ *we have*

$$\|\nabla \mathbf{Y}_{t,t_K}^x\| \leq \exp[-(1/2) \int_{T-t^\star}^{T-t} \beta_s \mathrm{d}s \mathbb{1}_{[0,t^\star)}(t)] \sigma_{T-t_K}^{-2\Gamma} \exp[(\Gamma + 1) \int_{T-t_K}^{T-t^\star} \beta_u \mathrm{d}u].$$

*Proof.* Let $x \in \mathbb{R}^d$. First, using (13) and Lemma C.2 we have that for any $s, t \in [0, T]$ with $s \leq t$

$$\mathrm{d}\|\nabla \mathbf{Y}_{s,t}^x\|^2 \leq 2\beta_{T-t}(\|\nabla \mathbf{Y}_{s,t}^x\|^2 - 2(1 - m_{T-t}^2 \operatorname{diam}(\mathcal{M})^2/(2\sigma_{T-t}^2))/\sigma_{T-t}^2 \|\nabla \mathbf{Y}_{s,t}^x\|^2)\mathrm{d}t.$$

First, assume that $s \leq t^\star$ and that $t \geq t^\star$. In that case, using Lemma D.8 we have that

$$\int_s^{t^\star} \beta_{T-u}(1 - 2/\sigma_{T-u}^2 + m_{T-u}^2 \operatorname{diam}(\mathcal{M})^2/\sigma_{T-u}^4)\mathrm{d}u \leq -(1/2) \int_s^{t^\star} \beta_{T-u}\mathrm{d}u.$$

Therefore, using that result and the fact that $\nabla \mathbf{Y}_{s,s}^x = \operatorname{Id}$ we get that

$$\|\nabla \mathbf{Y}_{s,t^\star}^x\| \leq \exp[-(1/2) \int_{T-t^\star}^{T-s} \beta_u \mathrm{d}u]. \tag{70}$$

In addition, using that for any $t \in (0, T]$ and $x_t \in \mathbb{R}^d$, $\|\nabla^2 \log p_t(x_t)\| \leq \Gamma/\sigma_t^2$ we have

$$\mathrm{d}\|\nabla \mathbf{Y}_{s,t}^x\|^2 \leq 2\beta_{T-t}(1 + 2\Gamma/\sigma_{T-t}^2)\|\nabla \mathbf{Y}_{s,t}^x\|^2 \mathrm{d}t.$$

In addition, using Lemma D.1 we have that

$$
\begin{aligned}
\int_{t^\star}^t \beta_{T-u}(1 + 2\Gamma/\sigma_{T-u}^2)\mathrm{d}u \leq {}& \Gamma[\log(\exp[2\int_0^{T-t^\star} \beta_{T-u}\mathrm{d}u] - 1) \\
& - \log(\exp[2\int_0^{T-t} \beta_{T-u}\mathrm{d}u] - 1)] + \int_{T-t}^{T-t^\star} \beta_u \mathrm{d}u \\
\leq {}& \Gamma[\log(\sigma_{T-t^\star}^2) - \log(\sigma_{T-t}^2)] + (\Gamma + 1) \int_{T-t}^{T-t^\star} \beta_u \mathrm{d}u \\
\leq {}& -\Gamma \log(\sigma_{T-t}^2) + (\Gamma + 1) \int_{T-t}^{T-t^\star} \beta_u \mathrm{d}u.
\end{aligned}
$$

Therefore, combining this result and (70), we get that

$$
\begin{aligned}
\|\nabla \mathbf{Y}_{s,t}\| &\leq \sigma_{T-t}^{-2\Gamma} \exp[(\Gamma + 1) \int_{T-t}^{T-t^\star} \beta_u \mathrm{d}u] \|\nabla \mathbf{Y}_{s,t^\star}\| \\
&\leq \sigma_{T-t}^{-2\Gamma} \exp[(\Gamma + 1) \int_{T-t}^{T-t^\star} \beta_u \mathrm{d}u] \exp[-(1/2) \int_{T-t^\star}^{T-s} \beta_u \mathrm{d}u].
\end{aligned}
$$

The proof in the cases where $s \geq t^\star$, $t \geq t^\star$ and $s \leq t^\star$, $t \leq t^\star$ are similar and left to the reader. $\square$

The rest of the proof follows the proof of Section 4. The following proposition is the counterpart of Proposition 10. Again note that the exponential dependency w.r.t. $1/\varepsilon$ has been replaced by a polynomial dependency.

**Proposition J.2.** *. Assume* **A**1*,* **A**2*,* **A**3*,* **A**4*,* **A**6 *and* $t_K = T - \varepsilon$*. In addition, assume that* $\varepsilon, \delta, \mathtt{M} \leq 1/32$*. Then*

$$\mathbf{W}_1(\pi_\infty \mathrm{Q}_{t_K}, \pi_\infty \mathrm{R}_K) \leq \mathtt{D}_0(\mathtt{M} + \delta^{1/2})/\varepsilon^{\Gamma+2},$$

*where*

$$\mathtt{D}_0 = 4(4 + 256d + 43664(1 + \operatorname{diam}(\mathcal{M}))^4) \exp[3(1 + \bar{\beta})^2(\Gamma + 2)(1 + \log(1 + \operatorname{diam}(\mathcal{M}))))].$$

*Proof.* Using Proposition 5, we have

$$\|\mathbf{Y}_{t_K} - Y_K\| \leq \int_0^{t_K} \|\nabla \mathbf{Y}_{u,t_K}(\bar{\mathbf{Y}}_{0,u})\| \|\Delta b_u((\bar{\mathbf{Y}}_{0,v})_{v \in [0,T]})\| \mathrm{d}u.$$

Combining this result, recalling that $t^\star$ is defined in (14) and Proposition J.1, we get

$$\|\mathbf{Y}_{t_K} - Y_K\| \leq \int_0^{t_K} \exp[-(1/2) \int_{T-t^\star}^{T-u} \beta_s \mathrm{d}s \mathbb{1}_{[0,t^\star)}(u)] \sigma_{T-t_K}^{-2\Gamma}$$

$$\times \exp[(\Gamma+1) \int_{T-t_K}^{T-t^\star} \beta_s \mathrm{d}s] \|\Delta b_u((\bar{\mathbf{Y}}_{0,v})_{v\in[0,T]})\| \mathrm{d}u$$
$$\leq \sigma_{T-t_K}^{-2\Gamma} \exp[(\Gamma+1) \int_{T-t_K}^{T-t^\star} \beta_s \mathrm{d}s] (\int_0^{t^\star} \exp[-(1/2) \int_{T-t^\star}^{T-u} \beta_s \mathrm{d}s] \Delta b_u((\bar{\mathbf{Y}}_{0,v})_{v\in[0,T]}) \mathrm{d}u$$
$$+ \int_{t^\star}^{t_K} \Delta b_u((\bar{\mathbf{Y}}_{0,v})_{v\in[0,T]}) \mathrm{d}u).$$

Using this result and Proposition 9 we get

$$\mathbf{W}_1(\pi_\infty \mathrm{Q}_{t_K}, \pi_\infty \mathrm{R}_K) \leq \mathbb{E}[\|\mathbf{Y}_{t_K} - Y_K\|]$$
$$\leq \sigma_{T-t_K}^{-2\Gamma} \exp[(\Gamma+1) \int_{T-t_K}^{T-t^\star} \beta_s \mathrm{d}s](\int_0^{t^\star} \exp[-(1/2) \int_{T-t^\star}^{T-u} \beta_s \mathrm{d}s] \mathbb{E}[\|\Delta b_u((\bar{\mathbf{Y}}_{0,v})_{v\in[0,T]})\|] \mathrm{d}u$$
$$+ \int_{t^\star}^{t_K} \mathbb{E}[\|\Delta b_u((\bar{\mathbf{Y}}_{0,v})_{v\in[0,T]})\|] \mathrm{d}u).$$
$$\leq \sigma_{T-t_K}^{-2\Gamma} \exp[(\Gamma+1) \int_{T-t_K}^{T-t^\star} \beta_s \mathrm{d}s] \mathtt{C}_0 (T - t_K + \bar{\beta})^2 (\mathtt{M} + \delta^{1/2})/(T - t_K)^2$$
$$\times(\int_0^{t^\star} \exp[-(1/2) \int_{T-t^\star}^{T-u} \beta_s \mathrm{d}s] \mathrm{d}u + t_K - t^\star)$$
$$\leq \sigma_{T-t_K}^{-2\Gamma} \exp[(\Gamma+1) \bar{\beta}(t_K - t^\star) \mathrm{d}s] \mathtt{C}_0 (T - t_K + \bar{\beta})^2 (\mathtt{M} + \delta^{1/2})/(T - t_K)^2$$
$$\times(\int_0^{t^\star} \exp[-(1/2) \int_{T-t^\star}^{T-u} \beta_s \mathrm{d}s] \mathrm{d}u + t_K - t^\star). \tag{71}$$

We have that
$$\int_0^{t^\star} \exp[-(1/2) \int_{T-t^\star}^{T-u} \beta_s \mathrm{d}s] \mathrm{d}u \leq \int_0^{t^\star} \exp[-(t^\star - u)/(2\bar{\beta})] \mathrm{d}u \leq 2\bar{\beta}. \tag{72}$$

In addition, using (14) we have

$$t_K - t^\star = T - \varepsilon - T + 2\bar{\beta}(1 + \log(1 + \mathrm{diam}(\mathcal{M}))) \leq 2\bar{\beta}(1 + \log(1 + \mathrm{diam}(\mathcal{M}))). \tag{73}$$

Finally, using Lemma D.2, we have that $\sigma_{T-t_K}^{-2} \leq (1 + \bar{\beta})/\varepsilon$. Combining this result, (72) and (73) in (71) and that for any $a \geq 0$, $1 + a \leq \exp[a]$, we get

$$\mathbf{W}_1(\pi_\infty \mathrm{Q}_{t_K}, \pi_\infty \mathrm{R}_K) \leq 4(1 + \bar{\beta})^{\Gamma+3}(1 + \log(1 + \mathrm{diam}(\mathcal{M})))$$
$$\times \exp[2\bar{\beta}^2(\Gamma+1)(1 + \log(1 + \mathrm{diam}(\mathcal{M})))] \mathtt{C}_0 (\mathtt{M} + \delta^{1/2}) \varepsilon^{-(\Gamma+2)}$$
$$\leq 4(1 + \bar{\beta})^{\Gamma+3}(1 + \log(1 + \mathrm{diam}(\mathcal{M})))$$
$$\times \exp[2\bar{\beta}^2(\Gamma+1)(1 + \log(1 + \mathrm{diam}(\mathcal{M})))] \mathtt{C}_0 (\mathtt{M} + \delta^{1/2}) \varepsilon^{-(\Gamma+2)}$$
$$\leq 4 \exp[3\bar{\beta}^2(\Gamma+2)(1 + \log(1 + \mathrm{diam}(\mathcal{M})))] \mathtt{C}_0 (\mathtt{M} + \delta^{1/2}) \varepsilon^{-(\Gamma+2)},$$

which concludes the proof. $\qquad \square$

We now state the equivalent of Proposition D.9.

**Proposition J.3.** *Assume* **A**1, **A**6 *and* $T \geq 2\bar{\beta}(1 + \log(1 + \mathrm{diam}(\mathcal{M}))$. *Then, for any* $x, y \in \mathbb{R}^d$ *and* $t \in [0, t_K]$
$$\mathbf{W}_1(\delta_x \mathrm{Q}_t, \delta_y \mathrm{Q}_t) \leq \exp[2(\Gamma+1)\bar{\beta}^2(1 + \log(1 + \mathrm{diam}(\mathcal{M})))] \sigma_{T-t_K}^{-2\Gamma} \|x - y\|.$$

*Proof.* This is a direct consequence of (53), Proposition J.1 and (73). $\qquad \square$

Finally, we control $\mathbf{W}_1(\pi_\infty \mathrm{Q}_{t_K}, \pi \mathrm{P}_{T-t_K})$. First, we have

$$\mathbf{W}_1(\pi_\infty \mathrm{Q}_{t_K}, \pi \mathrm{P}_{T-t_K}) = \mathbf{W}_1(\pi_\infty \mathrm{Q}_{t_K}, \pi \mathrm{P}_T \mathrm{Q}_{t_K}) \tag{74}$$
$$\leq \exp[2(\Gamma+1)\bar{\beta}^2(1 + \log(1 + \mathrm{diam}(\mathcal{M})))] \sigma_{T-t_K}^{-2\Gamma} \mathbf{W}_1(\pi \mathrm{P}_T, \pi_\infty).$$

To control $\mathbf{W}_1(\pi \mathrm{P}_T, \pi_\infty)$, we use a synchronous coupling, *i.e.* we set $(\mathbf{Y}_t, \mathbf{Z}_t)_{t\in[0,T]}$ such that

$$\mathrm{d}\mathbf{Y}_t = -\beta_t \mathbf{Y}_t \mathrm{d}t + \sqrt{2\beta_t} \mathrm{d}\mathbf{B}_t, \qquad \mathrm{d}\mathbf{Z}_t = -\beta_t \mathbf{Z}_t \mathrm{d}t + \sqrt{2\beta_t} \mathrm{d}\mathbf{B}_t,$$

where $(\mathbf{B}_t)_{t\in[0,T]}$ is a $d$-dimensional Brownian motion and $\mathbf{Y}_0 \sim \pi$, $\mathbf{Z}_0 \sim \pi_\infty$. We have that for any $t \in [0, T]$, $\mathbf{Z}_t \sim \pi_\infty$. In addition, denoting $u_t = \mathbb{E}[\|\mathbf{Y}_t - \mathbf{Z}_t\|]$ for any $t \in [0, T]$, we have that

$$\mathrm{d}u_t \leq u_0 \exp[-\int_0^t \beta_s \mathrm{d}s].$$

Therefore, combining this result and (74), we get that

$$\mathbf{W}_1(\pi_\infty Q_{t_K}, \pi P_{T-t_K}) \leq \exp[2(\Gamma+1)\bar\beta^2(1+\log(1+\mathrm{diam}(\mathcal{M})))]\sigma_{T-t_K}^{-2\Gamma}\exp[-\int_0^T \beta_t \mathrm{d}t]\mathbf{W}_1(\pi, \pi_\infty).$$

Therefore, similarly as in the proof of Proposition 10, we have

$$\mathbf{W}_1(\pi_\infty Q_{t_K}, \pi P_{T-t_K}) \leq \mathtt{D}_1 \exp[-T/\bar\beta]/\varepsilon^\Gamma,$$

with

$$\mathtt{D}_1 = \exp[2(\Gamma+2)(1+\bar\beta)^2(1+\log(1+\mathrm{diam}(\mathcal{M})))](\sqrt{d}+\mathrm{diam}(\mathcal{M})).$$

We conclude the proof of Theorem 3 upon combining this result, Proposition J.2 and (24).

## J.2 Hessian bounds for the uniform distribution

The goal of this section is to prove the following result.

**Proposition J.4.** *Assume that $\pi$ is the uniform distribution over $[-1/2, 1/2]^p$ for some $p \in \{1, \ldots, d\}$. Then, there exists $\Gamma \geq 0$ such that for any $t \in (0, T]$ and $x \in \mathbb{R}^d$, $\|\nabla^2 \log p_t(x_t)\| \leq \Gamma/\sigma_t^2$.*

*Proof.* Let $t \in (0, T]$. We start by deriving a closed form expression for $p_t$. We have for any $x \in \mathbb{R}^d$

$$\begin{aligned}
p_t(x) &= \int_{\mathcal{M}} \exp[-\|x-m_t z\|^2/(2\sigma_t^2)]\mathrm{d}\pi(z)(2\pi\sigma_t)^{-d/2} \\
&= \exp[-\textstyle\sum_{i=p+1}^d x_i^2/(2\sigma_t^2)]\textstyle\prod_{i=1}^p \int_{-1/2}^{1/2}\exp[-(x_i-m_t z_i)^2/(2\sigma_t^2)]\mathrm{d}z_i(2\pi\sigma_t)^{-d/2} \\
&= \exp[-\textstyle\sum_{i=p+1}^d x_i^2/(2\sigma_t^2)]\textstyle\prod_{i=1}^p \int_{x_i-m_t/2}^{x_i+m_t/2}\exp[-z_i^2/(2\sigma_t^2)]\mathrm{d}z_i(2\pi\sigma_t)^{-d/2}m_t^{-p} \\
&= \exp[-\textstyle\sum_{i=p+1}^d x_i^2/(2\sigma_t^2)](2\pi\sigma_t)^{-d/2}\times\textstyle\prod_{i=1}^p \int_{(x_i-m_t/2)/\sigma_t}^{(x_i+m_t/2)/\sigma_t}\exp[-z_i^2/2]\mathrm{d}z_i(\sigma_t/m_t)^p \\
&= (2\pi)^{d/2}(2\pi\sigma_t)^{-d/2}(\sigma_t/m_t)^p\textstyle\prod_{i=p+1}^d \varphi(x_i/\sigma_t) \\
&\quad \times \textstyle\prod_{i=1}^p \{\Phi((x_i+m_t/2)/\sigma_t) - \Phi((x_i-m_t/2)/\sigma_t)\},
\end{aligned}$$

where $\varphi(t) = \exp[-t^2/2]/\sqrt{2\pi}$ and $\Phi(t) = \int_{-\infty}^t \varphi(s)\mathrm{d}s$. Note that from this expression, $\nabla^2 \log p_t$ is diagonal. Hence, we only need to compute $\partial_i^2 \log p_t(x)$ for any $x \in \mathbb{R}^d$ and $i \in \{1, \ldots, d\}$. Let $i \in \{p+1, \ldots, d\}$. We have for any $x \in \mathbb{R}^d$

$$\partial_i^2 \log p_t(x) = -\sigma_t^{-2}.$$

We now turn to the case where $i \in \{1, \ldots, p\}$. In this case, we denote $F_t(\Phi, a, b) = \Phi(a+b) - \Phi(a-b)$ and we have

$$\partial_i^2 \log p_t(x) = (1/\sigma_t^2)\partial_2^2 \log F_t(\Phi, x_i/\sigma_t, m_t/\sigma_t). \tag{75}$$

We have that for any $a, b \in \mathbb{R}$ with $a \neq b$

$$\partial_2^2 \log F_t(\Phi, a, b) = F_t(\varphi', a, b)/F_t(\Phi, a, b) - F_t(\varphi, a, b)^2/F_t(\Phi, a, b)^2. \tag{76}$$

In what follows, we assume that $a > b$ and we define for any $t \in \mathbb{R}$

$$\mathrm{erfc}(t) = 1 - (2/\sqrt{\pi})\int_0^t \exp[-s^2]\mathrm{d}s.$$

Note that $F_t(\Phi, a, b) = (1/2)(\mathrm{erfc}((a-b)/2) - \mathrm{erfc}((a+b)/2))$. In addition, there exists $C > 0$ such that for any $t \neq 0$ we have

$$\mathrm{erfc}(t) \geq \exp[-t^2]/(\sqrt{\pi}t)(1+C/t^2), \qquad \mathrm{erfc}(t) \leq \exp[-t^2]/(\sqrt{\pi}t)(1-/(Ct^2)).$$

In particular, we have

$$\begin{aligned}
F_t(\Phi, a, b) &\leq \exp[-(a-b)^2/2]/(\sqrt{2\pi}(a-b))(1+R_0(a, b)), \\
F_t(\Phi, a, b) &\geq \exp[-(a-b)^2/2]/(\sqrt{2\pi}(a-b))(1-\bar R_0(a, b)),
\end{aligned} \tag{77}$$

where

$$R_0(a, b) = C/(a - b)^2 + \exp[+(a - b)^2/2 - (a + b)^2/2](a - b)/(a + b)(1 - /(C(a + b)^2)),$$
$$\bar{R}_0(a, b) = -C/(a - b)^2 + \exp[+(a - b)^2/2 - (a + b)^2/2](a - b)/(a + b)(1 + C/(a + b)^2),$$

Note that there exists $C_0 \geq 0$ such that for any $a, b \in \mathbb{R}$ with $a \geq b + 1$

$$(R_0(a, b) + \bar{R}_0(a, b))(a - b)^2 \leq C_0. \tag{78}$$

In particular, there exists $a_0 \geq 0$ such that if $a \geq b + a_0$, $\bar{R}_0(a, b) + R_0(a, b) \leq 1/2$. Similarly, we have

$$F_t(\varphi, a, b) = (2\pi)^{-1/2} \exp[-(a - b)^2/2](-1 + \exp[-((a + b)^2 - (a - b)^2)/2]). \tag{79}$$

We denote $R_1(a, b) = \exp[-((a + b)^2 - (a - b)^2)/2]$ and note that there exists $C_1 \geq 0$ such that for any $a, b \in \mathbb{R}$ with $a \geq b + 1$

$$R_1(a, b)(a - b)^2 \leq C_1. \tag{80}$$

Finally, we have

$$F_t(\varphi', a, b) = (2\pi)^{-1/2} \exp[-(a - b)^2/2](a - b)(1 + (a + b)/(a - b) \exp[-((a + b)^2 - (a - b)^2)/2]). \tag{81}$$

We denote $R_2(a, b) = (a + b)/(a - b) \exp[-((a + b)^2 - (a - b)^2)/2]$ and note that there exists $C_2 \geq 0$ such that for any $a, b \in \mathbb{R}$ with $a \geq b + 1$

$$R_2(a, b)(a - b)^2 \leq C_2. \tag{82}$$

Combining (76), (77), (79) and (81), we get that for any $a, b \in \mathbb{R}$ with $a \geq b + 1$

$$\partial_2^2 \log F_t(\Phi, a, b) \leq (a - b)^2[(1 + R_2(a, b))/(1 + \bar{R}_0(a, b)) - (1 + R_1(a, b))^2/(1 + R_0(a, b))^2].$$

In addition, we have for any $a, b \in \mathbb{R}$ with $a \geq b + 1$

$$(1 + R_2(a, b))/(1 + \bar{R}_0(a, b)) - (1 + R_1(a, b))^2/(1 + R_0(a, b))^2$$
$$= (R_2(a, b)(1 + R_0(a, b))^2 - R_1(a, b)(1 + \bar{R}_0(a, b)))/[(1 + \bar{R}_0(a, b))^2(1 + \bar{R}_0(a, b))].$$

Combining this result, (78), (80) and (82), we get that for any $a, b \in \mathbb{R}$ with $a \geq b + a_0$

$$R_2(a, b)(1 + R_0(a, b))^2 - R_1(a, b)(1 + \bar{R}_0(a, b)) \leq 4(C_1 + C_2)/(a - b)^2.$$

Therefore, there exists $C_3 \geq 0$ such that for any $a \geq b + a_0$, $|\partial_2^2 \log F_t(\Phi, a, b)| \leq C_3$. Similarly there exists $C_4 \geq 0$ such that if $a \leq -b - a_0$, $|\partial_2^2 \log F_t(\Phi, a, b)| \leq C_4$. By symmetry, there exists $C_5 \geq 0$ such that for any $b \geq a + a_0$ or $-b \geq -a - a_0$, we have $|\partial_2^2 \log F_t(\Phi, a, b)| \leq C_5$ and we conclude by continuity that there exists $C \geq 0$ such that for any $a, b \in \mathbb{R}$,

$$\partial_2 \log F_t(\Phi, a, b) \leq C,$$

which concludes the proof upon combining this result with (75). □

### J.3 The role of convexity

In this section, for any $x \in \mathbb{R}^d$, we define $f_x : \mathcal{M} \to \mathbb{R}_+$ given for any $y \in \mathcal{M}$ by $f_x(y) = \|x - y\|^2$. Before giving our main result we need to introduce a few useful tools.

For any subset $\mathsf{X} \subset \mathbb{R}^d$ and $x \in \mathbb{R}^d$ we define $\mathrm{P}(x) = \{y \in \mathsf{X} : d(x, \mathsf{X}) = d(x, y)\}$. Note that $\mathrm{P}(x)$ can be empty. We say that a set $\mathsf{X}$ is *Chebyshev* if for all $x \in \mathbb{R}^d$, there exists $p(x) \in \mathsf{X}$ such that $\mathrm{P}(x) = \{p(x)\}$, *i.e.* Chebyshev sets are the subsets of $\mathbb{R}^d$ such that each point admits a unique projection on $\mathsf{X}$. It is clear that all closed and convex sets are Chebyshev sets. Note that all Chebyshev sets are closed since for any Chebyshev set $\mathsf{X}$ and $x \in \partial \mathsf{X}$ (the frontier of $\mathsf{X}$) we have that there exists $p(x) \in \mathsf{X}$ such that $d(x, p(x)) = d(x, \mathsf{X}) = 0$, *i.e.* $x \in \mathsf{X}$. In addition, Chebyshev sets are also convex, see (Kritikos, 1938; Motzkin, 1935; Bundt, 1934). This result implies the following proposition.

**Proposition J.5.** *Let $\mathsf{X} \subset \mathbb{R}^d$. $\mathsf{X}$ is a closed convex set if and only if $\mathsf{X}$ is a Chebyshev set.*

In order to prove our main result, we introduce some basics on *Morse theory*. Assume that $\mathcal{M}$ is a smooth manifold and $f \in \mathrm{C}^\infty(\mathcal{M}, \mathbb{R})$. We say that $x \in \mathcal{M}$ is a non-degenerate minimizer if $x \in \mathcal{M}$ is a minimizer of $f$ and the Hessian of $f$ at $x$ is not singular. We then have the following proposition (see (Matsumoto, 2002) for instance).

**Proposition J.6.** *Let $f \in \mathrm{C}^\infty(\mathcal{M}, \mathbb{R})$ and $x \in \mathcal{M}$ a non degenerate minimizer of $f$. Then there exist $\mathsf{U} \subset \mathcal{M}$ open and $\varphi : \mathsf{U} \to \varphi(\mathsf{U}) \subset \mathbb{R}^p$ a local chart such that $\varphi(x) = 0$ and for any $\hat{y} \in \varphi(\mathsf{U})$*

$$f(\varphi^{-1}(\hat{y})) = f(x) + \|\hat{y}\|^2.$$

*In addition, we have $\mathrm{D}\varphi(x) = \nabla^2 f(x)$.*

Note that upon considering $\varphi^{-1}(\varphi(\mathsf{U})/2)$ instead of $\mathsf{U}$ we can always assume that $\varphi^{-1}$ has bounded derivatives.

Finally, we introduce the concept of *shape operator* (also called *second fundamental form* or *Weingarten form*), see (Bishop and Crittenden, 2011). We assume that the metric on $\mathcal{M}$ is the induced Euclidean metric. Let $N \in \Gamma(\mathrm{T}\mathcal{M}^\top)$ a section on the normal bundle $\mathrm{T}\mathcal{M}^\top$. We define $\mathfrak{A}^N : \Gamma(\mathrm{T}\mathcal{M})^2 \to \mathrm{C}^\infty(\mathcal{M})$ such that for any $V_1, V_2 \in \Gamma(\mathrm{T}\mathcal{M})$

$$\mathfrak{A}^N(V_1, V_2) = -\langle V_1, \nabla_{V_2} N\rangle.$$

Note that $\mathfrak{A}^N$ is symmetric, linear and the scalar product and covariant derivative $\nabla$ are considered w.r.t. the ambient Euclidean metric. The shape operator encodes the local geometry of $\mathcal{M}$. For example in the case of $\mathbb{S}^{d-1}$, we have that for any $N \in \Gamma(\mathrm{T}\mathcal{M}^\top)$ and $V_1, V_2 \in \Gamma(\mathrm{T}\mathcal{M})$

$$\mathfrak{A}^N(V_1, V_2) = -\langle V_1, V_2\rangle\langle N_0, N\rangle, \tag{83}$$

where $N_0$ is the normal vector field pointing outward of the sphere, see (Absil et al., 2013). We have the following result, see (Bishop and Crittenden, 2011, Theorem 3) and (Bishop, 1974).

**Proposition J.7.** *If $\mathcal{M}$ is convex then for any $N \in \Gamma(\mathrm{T}\mathcal{M}^\top)$ such that for any $x, y \in \mathcal{M}$, $\langle y - x, N(x)\rangle \geq 0$, $\mathfrak{A}^N$ is non-negative.*

Finally, let $\bar{f} \in \mathrm{C}^\infty(\mathbb{R}^d, \mathbb{R})$ and $f$ its restriction to $\mathcal{M}$. Using (Absil et al., 2013), we have for any $V_1, V_2 \in \Gamma(\mathrm{T}\mathcal{M})$

$$\nabla^2 f(V_1, V_2) = \langle V_1, \Pi(\nabla^2 \bar{f}(V_2))\rangle + \mathfrak{A}^{\Pi^\top(\nabla \bar{f})}(V_1, V_2), \tag{84}$$

where for any $x \in \mathcal{M}$, $\Pi_x$ is the orthogonal projection operator on $\mathrm{T}_x\mathcal{M}$. We are now ready to state our main result.

**Theorem J.8.** *Assume that $\mathcal{M} \subset \mathbb{R}^d$ is a smooth manifold and that $\pi$ admits a smooth density w.r.t. the Hausdorff measure on $\mathcal{M}$. The following hold:*

*(a) If $\mathcal{M}$ is convex then for any $x \in \mathcal{M}$ we have $\limsup_{t \to 0} \sigma_t^2 \|\nabla^2 \log p_t(m_t x)\| < +\infty$.*

*(b) If there exists $x \in \mathbb{R}^d$ such that $|\mathrm{P}(x)| > 1$ and for any $p(x) \in \mathrm{P}(x)$, $\mathfrak{A}_{p(x)}^{p(x)-x} \succ -\mathrm{Id}$. Then, we have $\liminf_{t \to 0} \sigma_t^4 \|\nabla^2 \log p_t(m_t x)\| > 0$.*

Theorem J.8 implies that one can obtain information about the geometry of $\mathcal{M}$ by computing the Hessian of the logarithmic gradient of the densities of $(\mathcal{L}(\mathbf{X}_t))_{t \in [0,T]}$. In the convex case the scaling w.r.t. $\sigma_t$ is of order $\sigma_t^{-2}$ whereas in the second scenario the scaling is of order $\sigma_t^{-4}$. Note that the condition "there exists $x \in \mathbb{R}^d$ such that $|\mathrm{P}(x)| > 1$" is equivalent to assuming that $\mathcal{M}$ is not a Chebyshev set and hence a non convex set in virtue of Proposition J.5. Therefore in Theorem J.8-(b), we assume that $\mathcal{M}$ is non convex and a curvature condition. The condition $\mathfrak{A}_x^{p(x)-x} \succ -\mathrm{Id}$ implies that the manifold is not too "negatively curved" at the projection points. The non-strict inequality is always true, *i.e.* for any $p(x) \in \mathrm{P}(x)$, $\mathfrak{A}_x^{p(x)-x} \succeq -\mathrm{Id}$ since $\nabla^2 f^x(p(x)) \succeq 0$.

We conjecture that this curvature condition can be relaxed. Indeed, it is not satisfied in the case where $\mathcal{M} = \{x \in \mathbb{R}^d : \|x\| = 1\}$, since the only point $x \in \mathbb{R}^d$ such that $|\mathrm{P}(x)| > 1$ is $x = 0$ and in that case

$P(x) = \mathcal{M}$ and therefore $\nabla^2 f^0 = 0$, which implies that for any $x \in \mathcal{M}$, $\mathfrak{A}_x^x = -\operatorname{Id}$. This formula could also have been obtained from (83). However, one can show that we still have $\liminf_{t \to 0} \sigma_t^4 \|\nabla^2 \log p_t(0)\| > 0$. In future works, we would like to relax these curvature conditions assuming that the manifold has an analytic structure and using results from (Combet, 2006).

Finally, we highlight that Theorem J.8-(a) is weaker than the condition **A**6. To bridge the gap between Theorem J.8-(a) and **A**6 one would need to strengthen Theorem J.8-(a) to derive *uniform in space* bounds. This would require to use *quantitative* version of the Morse lemmas, see (Le Loi and Phien, 2014) for instance. We postpone this study to future works. However, Theorem J.8-(b) implies that **A**6 does not hold. Hence, any non convex set which is not too "negatively curved" does not satisfy **A**6.

*Proof.* (a) First, we assume that $\mathcal{M}$ is convex. We show that for any $x \in \mathbb{R}^d$ and $p(x) \in P(x)$, $\nabla^2 f^x(p(x)) \succ 0$, *i.e.* the Hessian of $f^x$ is not degenerate. For any $x \in \mathbb{R}^d$, we define $\bar{f}^x$ such that for any $y \in \mathbb{R}^d$, $\bar{f}^x(y) = \|x - y\|^2$. Using (84), we have for any $x \in \mathbb{R}^d$

$$\nabla^2 f^x(p(x)) = \operatorname{Id} + \mathfrak{A}^{p(x)-x} \succeq \operatorname{Id} \succ 0. \tag{85}$$

Since $\mathcal{M}$ is convex for any $x \in \mathbb{R}^d$, $P(x) = \{p(x)\}$ and note that $P(x)$ is the set of minimizers of $f^x$. Let $x \in \mathbb{R}^d$. Using Proposition J.6 and (85), there exist $U \subset \mathcal{M}$ open and $\varphi : U \to \varphi(U) \subset \mathbb{R}^p$ a local chart such that $\varphi(p(x)) = 0$ and for any $\hat{y} \in \varphi(U)$

$$\|x - \varphi^{-1}(\hat{y})\|^2 = \|x - p(x)\|^2 + \|\hat{y}\|^2, \tag{86}$$

with $\varphi^{-1}(0) = p(x)$. Note that $D\varphi^{-1}(0) = (\nabla^2 f^x(p(x)))^{-1}$. For any $t \in (0, T]$, denote $\lambda_t = m_t / \sigma_t$. Using (86), we have that

$$\int_{U^2} (y_0 - y_1)^{\otimes 2} \exp[-(m_t/\sigma_t)^2 \|x - y_0\|^2/2] \exp[-(m_t/\sigma_t)^2 \|x - y_1\|^2/2] \mathrm{d}\pi(y_0)\mathrm{d}\pi(y_1)$$
$$= \int_{\varphi^{-1}(U)^2} (\varphi^{-1}(\hat{y}_0) - \varphi^{-1}(\hat{y}_1))^{\otimes 2} \exp[-(m_t/\sigma_t)^2 \|\hat{y}_0\|^2/2] \exp[-(m_t/\sigma_t)^2 \|\hat{y}_1\|^2/2] \mathrm{d}\hat{\pi}(\hat{y}_0)\mathrm{d}\hat{\pi}_1(\hat{y}_1)$$
$$\quad \times \exp[-(m_t/\sigma_t)^2 \|x - p(x)\|^2]$$
$$= (1/\lambda_t)^{2p} \int_{\lambda_t \varphi^{-1}(U)^2} (\varphi^{-1}(\hat{y}_0/\lambda_t) - \varphi^{-1}(\hat{y}_1/\lambda_t))^{\otimes 2} \exp[-\|\hat{y}_0\|^2/2] \exp[-\|\hat{y}_1\|^2/2] \mathrm{d}\hat{\pi}(\hat{y}_0)\mathrm{d}\hat{\pi}_1(\hat{y}_1)$$
$$\quad \times \exp[-(m_t/\sigma_t)^2 \|x - p(x)\|^2],$$

where $\hat{\pi} = \varphi_\# \pi$ admits a positive density w.r.t. the Lebesgue measure. Therefore, there exists $C_0 \geq 0$ such that for any $t \in (0, T]$

$$(2\pi/\lambda_t^2)^{-p} \|\int_{U^2} (y_0 - y_1)^{\otimes 2} \exp[-(m_t/\sigma_t)^2 \|x - y_0\|^2/2] \exp[-(m_t/\sigma_t)^2 \|x - y_1\|^2/2] \mathrm{d}\pi(y_0)\mathrm{d}\pi(y_1)\|$$
$$\leq C_0 (2\pi)^{-p} \sigma_t^2 \int_{(\mathbb{R}^p)^2} \|\hat{y}_0 - \hat{y}_1\|^2 \exp[-\|\hat{y}_0\|^2/2] \exp[-\|\hat{y}_1\|^2/2] \mathrm{d}\hat{y}_0 \mathrm{d}\hat{y}_1$$
$$\quad \times \exp[-(m_t/\sigma_t)^2 \|x - p(x)\|^2]$$
$$\leq 2C_0 p \sigma_t^2 \exp[-(m_t/\sigma_t)^2 \|x - p(x)\|^2], \tag{87}$$

where we have used that $\|a - b\|^2 \leq 2(\|a\|^2 + \|b\|^2)$ for any $a, b \in \mathbb{R}^p$ in the last line. In addition, since $U$ is open we have that $\mathcal{M} \cap U^c$ is compact and since for any $y \in \mathcal{M} \cap U^c$, $\|y - x\|^2 > \|p(x) - x\|^2$, there exists $\varepsilon > 0$ such that for any $y \in \mathcal{M} \cap U^c$, $\|y - x\|^2 \geq \|p(x) - x\|^2 + \varepsilon$. Therefore, we have

$$\|\int_{(\mathcal{M} \cap U^c)^2} (y_0 - y_1)^{\otimes 2} \exp[-(m_t/\sigma_t)^2 \|x - y_0\|^2/2] \exp[-(m_t/\sigma_t)^2 \|x - y_1\|^2/2] \mathrm{d}\pi(y_0)\mathrm{d}\pi(y_1)\|$$
$$\leq \operatorname{diam}(\mathcal{M})^2 \exp[-(m_t/\sigma_t)^2 \|x - p(x)\|^2] \exp[-\varepsilon(m_t/\sigma_t)^2]. \tag{88}$$

Combining (87) and (88) there exists $C_1 \geq 0$ such that for any $t \in (0, T]$

$$(2\pi/\lambda_t^2)^{-p} \|\int_{\mathcal{M}^2} (y_0 - y_1)^{\otimes 2} \exp[-(m_t/\sigma_t)^2 \|x - y_0\|^2/2] \exp[-(m_t/\sigma_t)^2 \|x - y_1\|^2/2] \mathrm{d}\pi(y_0)\mathrm{d}\pi(y_1)\|$$
$$\leq C_1 \sigma_t^2 \exp[-(m_t/\sigma_t)^2 \|x - p(x)\|^2]. \tag{89}$$

In addition, there exists $C_2 > 0$ such that for any $t \in (0, T]$

$$(2\pi/\lambda_t^2)^{-p} \int_U \exp[-(m_t/\sigma_t)^2 \|x - y\|^2/2] \mathrm{d}\pi(y) \geq (1/C_2) \exp[-(m_t/\sigma_t)^2 \|x - p(x)\|^2]. \tag{90}$$

Therefore, combining (89) and (90), we get that there exists $C_3 \geq 0$ such that for any $t \in (0, T]$

$$\| \int_{\mathcal{M}} (y_0 - y_1)^{\otimes 2} \exp[-(m_t/\sigma_t)^2 \|x - y_0\|^2 / 2] \exp[-(m_t/\sigma_t)^2 \|x - y_1\|^2 / 2] \mathrm{d}\pi(y_0) \mathrm{d}\pi(y_1) \|$$
$$/ (\int_{\mathcal{M}} \exp[-(m_t/\sigma_t)^2 \|x - y\|^2 / 2] \mathrm{d}\pi(y))^2 \leq C_3 \sigma_t^2.$$

We conclude the proof in the convex case upon combining this result and Lemma C.2.

(b) Second, we assume that there exists $x \in \mathbb{R}^d$ such that $|\mathrm{P}(x)| > 1$ and for any $p(x) \in \mathrm{P}(x)$, $\mathfrak{A}^{p(x)-x} \succ -\mathrm{Id}$. Using (84), we have for any $p(x) \in \mathrm{P}(x)$

$$\nabla^2 f^x(p(x)) = \mathrm{Id} + \mathfrak{A}^{p(x)-x} \succ 0.$$

Using this fact and that $\mathrm{P}(x)$ is the set of minimizers of $f^x$ and is compact, we get that $|\mathrm{P}(x)| < +\infty$. Hence, we assume that $\mathrm{P}(x) = \{p_i(x)\}_{i=1}^N$ with $N > 1$. Using Proposition J.6, for any $i \in \{1, \ldots, N\}$, there exist $\mathsf{U}_i \subset \mathcal{M}$ open and $\varphi_i : \mathsf{U}_i \to \varphi_i(\mathsf{U}_i) \subset \mathbb{R}^p$ a local chart such that $\varphi_i(p_i(x)) = 0$ and for any $\hat{y} \in \varphi_i(\mathsf{U}_i)$

$$\|x - \varphi_i^{-1}(\hat{y})\|^2 = \|x - p_i(x)\|^2 + \|\hat{y}\|^2,$$

with $\varphi_i^{-1}(0) = p_i(x)$. Note that $\mathrm{D}\varphi_i^{-1}(0) = (\nabla^2 f^x(p_i(x)))^{-1}$. Without loss of generality we assume that for any $i, j \in \{1, \ldots, N\}$, $\mathsf{U}_i \cap \mathsf{U}_j = \emptyset$. We have that

$$\int_{\mathsf{U}_0 \times \mathsf{U}_1} (y_0 - y_1)^{\otimes 2} \exp[-(m_t/\sigma_t)^2 \|x - y_0\|^2 / 2] \exp[-(m_t/\sigma_t)^2 \|x - y_1\|^2 / 2] \mathrm{d}\pi(y_0) \mathrm{d}\pi(y_1)$$
$$= \int_{\varphi_0^{-1}(\mathsf{U}_0) \times \varphi_1^{-1}(\mathsf{U}_1)} (\varphi_0^{-1}(\hat{y}_0) - \varphi_1^{-1}(\hat{y}_1))^{\otimes 2} \exp[-(m_t/\sigma_t)^2 \|\hat{y}_0\|^2 / 2]$$
$$\times \exp[-(m_t/\sigma_t)^2 \|\hat{y}_1\|^2 / 2] \mathrm{d}\hat{\pi}(\hat{y}_0) \mathrm{d}\hat{\pi}_1(\hat{y}_1) \exp[-(m_t/\sigma_t)^2 \|x - p(x)\|^2]$$
$$= (1/\lambda_t)^{2p} \int_{\lambda_t \varphi_0^{-1}(\mathsf{U}_0) \times \lambda_t \varphi_1^{-1}(\mathsf{U}_1)} (\varphi_0^{-1}(\hat{y}_0/\lambda_t) - \varphi_1^{-1}(\hat{y}_1/\lambda_t))^{\otimes 2}$$
$$\times \exp[-\|\hat{y}_0\|^2 / 2] \exp[-\|\hat{y}_1\|^2 / 2] \mathrm{d}\hat{\pi}(\hat{y}_0) \mathrm{d}\hat{\pi}_1(\hat{y}_1) \exp[-(m_t/\sigma_t)^2 \|x - p(x)\|^2]. \tag{91}$$

In addition, using the dominated convergence theorem we have

$$\lim_{t \to +\infty} (2\pi)^{-p} \int_{\lambda_t \varphi_0^{-1}(\mathsf{U}_0) \times \lambda_t \varphi_1^{-1}(\mathsf{U}_1)} (\varphi_0^{-1}(\hat{y}_0/\lambda_t) - \varphi_1^{-1}(\hat{y}_1/\lambda_t))^{\otimes 2} \tag{92}$$
$$\times \exp[-\|\hat{y}_0\|^2 / 2] \exp[-\|\hat{y}_1\|^2 / 2] \mathrm{d}\hat{\pi}(\hat{y}_0) \mathrm{d}\hat{\pi}_1(\hat{y}_1) = \hat{h}_0(p_0(x)) \hat{h}_1(p_1(x)) (p_0(x) - p_1(x))^{\otimes 2},$$

where $\hat{h}_i$ is the density of $\hat{\pi}_i = (\varphi_i)_\# \pi$ w.r.t. the Lebesgue measure. In addition, there exists $C_4 \geq 0$ such that for any $t \in (0, T]$

$$(2\pi)^{-p/2} \int_{\mathsf{U}_0} \exp[-(m_t/\sigma_t)^2 \|x - y_0\|^2 / 2] \mathrm{d}\pi(y_0)$$
$$= (2\pi)^{-p/2} \int_{\varphi_0^{-1}(\mathsf{U}_0)} \exp[-(m_t/\sigma_t)^2 \|\hat{y}_0\|^2 / 2] \mathrm{d}\hat{\pi}(\hat{y}_0) \exp[-(m_t/\sigma_t)^2 \|x - p(x)\|^2]$$
$$\leq C_4 \exp[-(m_t/\sigma_t)^2 \|x - p(x)\|^2] \lambda_t^{p/2}. \tag{93}$$

In addition, since $\mathsf{U}_0$ is open we have that $\mathcal{M} \cap \mathsf{U}_0^c$ is compact and since for any $y \in \mathcal{M} \cap \mathsf{U}_0^c$, $\|y - x\|^2 > \|p(x) - x\|^2$, there exists $\varepsilon > 0$ such that for any $y \in \mathcal{M} \cap \mathsf{U}_0^c$, $\|y - x\|^2 \geq \|p(x) - x\|^2 + \varepsilon$. Therefore, we have

$$\int_{\mathcal{M} \cap \mathsf{U}_0^c} \exp[-(m_t/\sigma_t)^2 \|x - y_0\|^2 / 2] \mathrm{d}\pi(y_0) \leq C_5 \exp[-(m_t/\sigma_t)^2 \|x - p(x)\|^2 / 2] \exp[-\varepsilon (m_t/\sigma_t)^2].$$

Combining this result and (93), we get that there exists $C_6 > 0$ such that

$$(2\pi/\lambda_t^2)^{-p/2} \int_{\mathcal{M}} \exp[-(m_t/\sigma_t)^2 \|x - y_0\|^2 / 2] \mathrm{d}\pi(y_0) \leq C_6 \exp[-(m_t/\sigma_t)^2 \|x - p(x)\|^2 / 2].$$

Combining this result, (91) and (92), there exists $C_7 > 0$ such that

$$\liminf_{t \to 0} \int_{\mathcal{M}} (y_0 - y_1)^{\otimes 2} \exp[-(m_t/\sigma_t)^2 \|x - y_0\|^2 / 2] \exp[-(m_t/\sigma_t)^2 \|x - y_1\|^2 / 2] \mathrm{d}\pi(y_0) \mathrm{d}\pi(y_1)$$
$$/ (\int_{\mathcal{M}} \exp[-(m_t/\sigma_t)^2 \|x - y\|^2 / 2] \mathrm{d}\pi(y))^2 \geq (x_0 - x_1)^{\otimes 2} / C_6.$$

We conclude upon combining this result and Lemma C.2.

$\square$

