# OpenReview forum: "Convergence of denoising diffusion models under the manifold hypothesis"
_TMLR — Accepted by TMLR_

### Review · Reviewer_RSJB · 2022-08-22

**Summary Of Contributions:**

The authors theoretically study denoising diffusion models, which have recently gained immense popularity for generating images. The main theoretical result is a quantitative convergence guarantee that ensures that if the various parameters are chosen correctly (e.g., the step size and score estimation error should be small), then the denoising diffusion model outputs an approximate sample from the data distribution in $W_1$ distance.

Diffusion models have been studied before, but prior works assumed that the score function of the law of the forward process is Lipschitz, uniformly in time. The authors of the present work point out that such an assumption is typically violated in practice, especially if the data distribution is not absolutely continuous w.r.t. Lebesgue measure (e.g., the data distribution is supported on a lower dimensional submanifold, as predicted by the so-called manifold hypothesis). The main novelty of this work, therefore, is to perform the analysis assuming that the data distribution is supported on an arbitrary compact subset of Euclidean space, which necessitates an analysis in a Wasserstein distance (rather than total variation or KL).

I have not checked the proofs in detail but to the best of my knowledge they appear to be correct. The techniques are more or less standard.

**Broader Impact Concerns:**

None.

**Requested Changes:**

As discussed above, I would like the authors to soften their claims about extending the results of Lee et al. under an $L^2$ error assumption, as I believe the results of this paper are substantially weaker in this regard. Also, please acknowledge the exponential dependencies as limitations of the work.

I have also found minor typos:
- Ornstein–Uhlenbeck is consistently misspelled throughout the paper.
- Pg. 3, “on a lower dimensional manifolds”
- Throughout the paper, in the condition $T \ge \bar \beta\\, (1 + \log(1 + \text{diam}(\mathcal M)))$, the closing parenthesis is missing.
- Pg. 5, in the discussion of the first term, it should say $\delta^{1/2}$, not $\delta$. Also, “the second term correspond” should read “the second term corresponds”.
- Eq. (9), the square should be on the $\nabla$, not on $p_t$
- Could the authors please check the definition of the Minkowski dimension in eq. (11)? It makes no sense to me. Why is there a $d - \cdots$? And should $\mathcal M_\varepsilon$ be intersected with $\mathcal M$ itself? (Does $d$ refer to the intrinsic distance on the manifold, or to the distance on $\mathbb R^d$? Thus far it has not been assumed that $\mathcal M$ is a manifold, only a compact set.)
- Proof of Lemma D.2, there is an extra “$\le$”.
- In the display equation below eq. (68), the square should be on the norm, not on $\nabla \mathbf Y_{s,t}$. In the chain of display equations below this, there is a missing square on $\sigma_{T-t}$.

**Strengths And Weaknesses:**

The strength of the work is that it works under a realistic assumption on the data distribution. In particular, it is a welcome feature of the analysis that dissipativity and bounds on the score function are derived from the compact support assumption, rather than assumed (as is typical for these analyses).

There are also some weaknesses of the work:
- The quantitative bounds depend exponentially on parameters such as the diameter of the support, which can be quite pessimistic.
- Although the assumption on the score estimation error is weaker than assuming an $L^\infty$ error bound, it is still unrealistic; it is preferable to work with an $L^2$ error guarantee, especially since the score matching objective is the $L^2$ error. Towards this end, the authors show in Appendix H that an $L^2$ error assumption suffices, but here there is an important distinction from the work of Lee et al. From my understanding, the crucial difficulty of Lee et al. is that they only assume an $L^2$ error bound with respect to the law of the true backwards process (BP), which is realistic as it is based on the score matching objective, whereas in this paper the $L^2$ error bound is with respect to the law of the algorithm (EI discretization of SBP). The latter assumption is far easier to handle as it simply contributes an extra term to the discretization error, whereas the assumption in Lee et al. is more subtle, requiring some delicate change of measure arguments.

---

> ### Author Response · Authors · 2022-09-11
> **Response to Reviewer RSJB**
>
> Thank you for taking the time to review our submission and for your constructive feedback. We answer your main concerns below.
>
> * The quantitative bounds depend exponentially on parameters such as the diameter of the support, which can be quite pessimistic.
>
> We agree with the reviewer that one limitation of this works lies into the exponential dependency w.r.t. $\varepsilon$. It is an important question to try to find conditions under which this dependency is relaxed. Currently under global bounds on the Hessian such relaxations are possible but such Lipschitzness conditions seem hard to obtain in full generality as they seem linked with the convexity of the manifold. We agree however that there is no clear answer to this problem yet. Note also that the this exponential dependency is directly linked to the explosion of the Lipschitz constant of $\nabla \log p_t$. In our work we noticed that an explosion of order $O(1/t)$ leads to polynomial rates w.r.t. $\varepsilon$ whereas an explosion of order $O(1/t^2)$ leads to exponential rates w.r.t. $\varepsilon$. We emphasize that such conditions on the Lipschitz constant of $\nabla \log p_t$ were also identified in [1, Proposition 1]. In the revised version of our paper we have added a discussion on this topic. In particular, we emphasize that one of the limitation of our current work lies in the exponential dependency.
>
> * Although the assumption on the score estimation error is weaker than assuming an $L^\infty$ error bound, it is still unrealistic; it is preferable to work with an  error guarantee, especially since the score matching objective is the $L^2$ error. Towards this end, the authors show in Appendix H that an $L^2$ error assumption suffices, but here there is an important distinction from the work of Lee et al. From my understanding, the crucial difficulty of Lee et al. is that they only assume an  error bound with respect to the law of the true backwards process (BP), which is realistic as it is based on the score matching objective, whereas in this paper the $L^2$ error bound is with respect to the law of the algorithm (EI discretization of SBP). The latter assumption is far easier to handle as it simply contributes an extra term to the discretization error, whereas the assumption in Lee et al. is more subtle, requiring some delicate change of measure arguments.
>
> We fully agree with the reviewer and thank them for pointing out this key difference. In the revised version of our work we have tamed our claims accordingly. In particular, we highlight that the work of [2] takes place in a more realistic setting than our bounds. We are currently working towards extending our theory so that the bounds derived in Assumption H are also given w.r.t. the distribution of the true backward process. Such an extension would be based on the tools introduced in [2].
>
> [1] Block, Mroueh, Rakhlin, Ross -- Fast Mixing of Multi-Scale Langevin Dynamics under the Manifold Hypothesis (2020)
>
> [2] Lee, Lu, Tan -- Convergence for score-based generative modeling with polynomial complexity (2022)

---

### Review · Reviewer_Fneb · 2022-09-03

**Summary Of Contributions:**

In Theorem 1, the paper upper bounds the approximation error when a) the target is a compactly-supported probability measure and b) the approximation is a discretization of a continuous-time process coming from the diffusion modelling literature. The idealized continuous-time process (Equation 2) is a time-reversed Ornstein-Ulhenbeck (Equation 1). As Equation 2 depends on an analytically-unavailable family of functions, the first part of the approximation is replacing the family of functions with empirical estimates (replacing Equation 2 with Equation 4). The second part of the approximation is discretizing the continuous time process: the final approximation is Equation 5. The remaining theoretical results (Corollary 2, Theorem 3 and Proposition 4) are either implications of Theorem 1 or improvements of Theorem 1 under more assumptions.

**Requested Changes:**

Please address my questions from the "Strengths and Weaknesses" section. In addition, there are small changes that I would like to see.

The following are typos that should be fixed
- After Equation 8, currently it says $Q_{t}$ is associated with the distribution of $\hat{Y}$ at $t$, but it should be associated with the distribution of $Y$ at $t$ instead.
- Theorem 3 has a typo in Equation 9. Should be Hessian of log density. Not gradient of the square of log density.
- Proposition 4 should say $\epsilon^{1/2}$ rather than $\epsilon$ to be consistent with Theorem 1

I am confused about the meaning of these following sentences
- What is the purpose of the sentence “setting $\beta_0 \leq \beta_T$ allows for better control of the backward diffusion”? Sounds redundant given what’s in the previous sentence “$\beta_t$ is a non-decreasing weight function”



**Strengths And Weaknesses:**

# Strengths

The submission analyzes the approximation error under weaker assumptions that those currently assumed in the literature. Whereas the existing literature only apply to target distributions that are absolutely continuous with the d-dimensional Lebesgue measure (Equation 12), the submission's key result (Theorem 1) applies to such target distributions and also more kinds of targets, such as those supported on a lower-dimensional manifold.

The proof of Theorem 1 is sound. I can verify the correctness of the results in Section 4 (except for Proposition 5 as it is discussed in an external reference).

# Weaknesses

It is unclear what is the space that realizations of the tangent process of Proposition 5 live in. Based on the initial condition $\nabla Y_{s,s}^x = \text{Id}$, it seems like each $\nabla Y_{s,t}^x$ would be a $d \times d$ matrix.
But based on the equality in Proposition 5, where it is written $\nabla Y_{u,t}^x (Y_{s,u}^x)$, it seems like $\nabla Y_{u,t}^x $ is a function that takes in $\mathbb{R}^d$ vectors and outputs $\mathbb{R}^d$ vectors.

There is a small mistake in the statement of Corollary 2. The actual relationship between $T$, $M$, $\delta$ and $\gamma_K$ on $\eta$ is likely something like what’s been written, but can’t be exactly this. For instance, plugging $\gamma_K = \eta^2$ into the $D_0 \epsilon^{1/2}$ part of Theorem 1 already evaluates to $D_0 \eta$, but we haven’t even evaluated the contributions of the other parts of the bound.

The claim "letting $\epsilon \to 0, T \to \infty, \delta \to 0, M \to 0$, we get that $W_1 \to 0$" is not correct. There must be some relationship between the rates at which these quantities approach the limit for the final upper bound of $W_1$ to go to zero. I think Corollary 2 can be used to make the rates more explicit.

There are some small typos that should be addressed (please see Requested Changes section)

---

> ### Author Response · Authors · 2022-09-11
> **Response to Reviewer Fneb**
>
> We thank the reviewer for their thorough review of our work. Below, we address the concerns of the reviewer.
>
> * It is unclear what is the space that realizations of the tangent process of Proposition 5 live in. Based on the initial condition $\nabla Y_{s,s}^x = \mathrm{Id}$, it seems like each $\nabla Y_{s,t}^x$
>  would be a $d \times $  matrix. But based on the equality in Proposition 5, where it is written $\nabla Y_{u,t}^x(Y_{s,u}^x)$, it seems like $\nabla Y_{u,t}^x$ is a function that takes in $\mathbb{R}^d$ vectors and outputs $\mathbb{R}^d$ vectors.
>
> We thank the reviewer for bringing this notation problem to our attention and apologize for the inconvenience. In what follows, we try to give a brief explanation of the tangent process. As the reviewer points out $(\nabla Y_{s,t}^x)$ is a $d \times $d stochastic process. This stochastic process satisfies $\mathrm{d} \nabla Y_{s,t}^x = (\mathrm{Id} + \nabla^2 \log p_{T-t}(Y_{s,t}^x)) \mathrm{d} t$ (we have omitted the speed $\beta_t$ for clarity here), with $\nabla Y_{s,s}^x = \mathrm{Id}$. Of course this process depends on $(Y_{s,t}^x)$ which is a $d$-dimensional stochastic process satisfying the time-reversal dynamics $\mathrm{d} Y_{s,t}^x = (Y_{s,t}^x + 2 \nabla \log p_{T-t}(Y_{s,t}^x) \mathrm{d} t + \sqrt{2} \mathrm{d} B_t$ (again we omit the speed $\beta_t$) with initial condition $Y_{s,s}^x = x$. The current notation $\nabla Y_{u,t}^x(Y_{s,u}^x)$ does not make sense and should be instead $\nabla Y_{u,t}^{Y_{s,u}^x}$. We apologize for the error and have revised the manuscript accordingly.
>
> * There is a small mistake in the statement of Corollary 2. The actual relationship between $T$, $M$, $\delta$  and $\gamma_K$  on $\eta$  is likely something like what’s been written, but can’t be exactly this.
>
> We agree with the reviewer. There is a typo here and a numerical constant is missing. $D_0$ should be replaced by $4D_0$.
>
> * There must be some relationship between the rates at which these quantities approach the limit for the final upper bound of $W_1$ to go to zero
>
> We agree with the reviewer that the current sentence is hand-wavy here as we do not precise *how* the different constants go to zero. This issue was also highlighted by Reviewer W6qc. We have replaced the sentence by "First, we note that letting $T \to +\infty$,
>   $\delta,M \to 0$ and then $\varepsilon \to 0$ we get that
>   $W_1(\mathcal{L}(Y_{K}), \pi) \to 0$. An explicit dependency
>   between these parameters is given in Corollary 2". We refer to Corollary 2 for a quantitative analysis of the phenomenon.
>
> Finally, we have fixed the typos and removed the redundant sentence regarding the schedule $\beta_t$.

---

> > ### Comment · Reviewer_Fneb · 2022-09-19
> > **Reply to Authors' Response**
> >
> > I thank the authors for addressing my concerns. I don't have any issue with latest revision.

---

### Review · Reviewer_W6qc · 2022-09-03

**Summary Of Contributions:**

The author shows a bound in the Wasserstein-1 distance between the distribution output by an DDM and the true data generating distribution. The bound is the applied to the empirical measure using the triangle inequality. A tigther bound is derived in the case that the hessian of the log-likelihood satisfies some more conditions.

**Broader Impact Concerns:**

As a theoretical study of DDMs I do not think that this work needs a Broader Impact section.

**Requested Changes:**

## Major
1. The sentence directly after Theorem 1 seems misleading. It depends very much on the order in which one takes these limits. If one first takes $M \to 0$ and $\delta$ to 0, one is treating the exact case with no error and the statement is trivial. If one lets $\epsilon \to 0$ before any of these two, the bound exponentially diverges towards infinity and the statement is not true. Therefore one only sees that the Wasserstein distance actually converges to 0 in the exact case, which is known already, since then $\mathcal{L}(Y_T) = \pi$.
2. In Section 3.2 generalization is treated. For that one applies the earlier Theorems to approximate the empirical distribution and then bounds the Wasserstein distance between the empirical distribution and $\pi$ by applying other results. Using the triangle inequality one now gets a bound between the law output by a DDM started in the empirical distribution and $\pi$. It is not really discussed in which way these results relate to generalization. First of all, due to employing the triangle inequality, the distance bound between $\mathcal{L}(Y_K)$ and $\pi$ will always be worse than the distance between the empirical measure in which the DDM is started and $\pi$. Therefore, it can not be deduced from this if the output distribution of the DDM is closer to the empirical measure, the true data generating distribution or anything in between, even if the first terms in the RHS of the bound in Proposition 4 could be made rather small. To be clear, I think the Proposition is interesting, since the result than actually entails all the approximations made by an DDM into one bound, I just think it is not strongly related to generalization, or if it is indeed the case, the authors should provide some discussion on the connection.
3. In the comparison sections, the authors state that this work goes one step further than [2]. It seems to be that the results are complementary, rather then building on each other, since they neither contain nor employ the results from [2]. The similarity rather seems to be the setting of studying DDMs under the manifold hypothesis.
4. As stated in the "Weaknesses" section the most interesting results from this work is some intuition on how the different parameters influence the Wasserstein distance. Therefore, I would wish the work to focus on this a bit more and discuss these points in more detail. In particular, Assumption 3 seems to be the right kind of Assumption to me, since one gets linear dependence on $M$ in the error bounds. Nevertheless, the proofs rely on the drift not being evaluated on $[T - \epsilon, T]$. If one wants to choose a small $\epsilon$ and evaluate the drift approximation also on $[T - \epsilon, T]$, one would need $M$ to be chosen of order $\frac{1}{\epsilon^2}\exp(\kappa/\epsilon)$. Therefore, if I understand it right, if the Wasserstein distance is to be kept constant for different values of $\epsilon$, the drift approximation quality would actually need to get exponentially better as one lets $\epsilon \to 0$ and also evaluates the drift on the final part of the interval. These are some important implications of the results and deserve more treatment from my point of view.
5. Could it be discussed how realistic the Assumptions of Theorem 3 are or when they could be expected to hold/not hold?

## Minor
6. Maybe use a different letter than $\kappa$ in (3), since $\kappa$ will be used later for another quantity.
7. "Our result could be extended to W_p". Maybe that could be justified a bit more, since especially the Wasserstein-1 distance has some different properties to the other Wasserstein distances.
8.  In general I think it would help the clarity and readability of the results a lot if the results could be split up in a part that explicitly bounds the distance between the continuous SDE with the true drift and the continuous SDE with the approximate drift, and then another result for the discretization, but that might also be to much work.
9. In the Proof of Lemma C.2, on the first line it says $p_t^N = \hat{p}_t^N/(2\pi\sigma_t^2)^{d/2}$. This is the definition of $\hat{p}_t^N$, right? Maybe this could be written as $\hat{p}_t^N := p_t^N (2 \pi\sigma_t^2)^{d/2}$?


Overall, I think this work is a great contribution to literature and should be published if the above points are addressed. Also as it is, I think a main interesting point is that Assumption 3 measures the error in a way, that at least for fixed $\epsilon$, the Wasserstein distance depends linearly on $M$, which is a rather interesting result.

[2] Score-Based Generative Models Detect Manifolds, Jakiw Pidstrigach, https://arxiv.org/abs/2206.01018

**Strengths And Weaknesses:**

## Strengths
- The work considers all the approximations made by DDMs in practice, which are: the approximation of initial condition by $N(0, I)$, the approximation of the drift, the approximation of the $\pi$ by an empirical measure and the discretization of the SDE.
- One can read off the  dependence of the bounds on different parameters and approximations.
- The constants in front of the bounds are made explicit in the proofs.
- Assumption 3 furthermore is a novel way to measure the error between the approximation, which could guide further research in the area, since it fits the problem well and is more realistic than expecting uniform error bounds. Also the results show, that by measuring the error in this way, one is able to make the Wasserstein depend linearly on $M$, further emphasizing that this a good way to measure the approximation error.

## Weaknesses
If the drift approximation is not perfect, the bounds are of the form $\exp(1/\epsilon)/\epsilon^2$, where $\epsilon$ is the size of the last step. The last step is often chosen as small as possible in practice, e.g. of the order $\epsilon = 10^{-3}$ or $\epsilon = 10^{-5}$. To put the derived bounds into context, I want to quickly derive a bound between $\pi$ and the initial condition of the reverse SDE, $\mathcal{N}(0, I)$.

To that end the triangle inequality can be employed to derive
$$W(N(0, I), \pi) \le W(N(0, I), \delta_0) + W(\delta_0, \pi).$$
The first term can be bounded by
$$W(N(0, I), \delta_0) \le \sqrt{d},$$
see Lemma 2.4 of [1]. The second term can be upper bounded by $\text{diam}(\mathcal{M})$. Therefore, the distance between the initial condition of the reverse SDE and $\pi$ is of order
$$W(N(0, I), \pi) \le \sqrt{d} + \text{diam}(\mathcal{M}).$$
The constant $D_0$ exceeds this bound, and the exponential terms afterwards will amplify this term if the approximation error is not of a very small magnitude.
Therefore, for realistic choices of the parameters, this bound is probably larger than $\sqrt{d} + \text{diam}(\mathcal{M})$. Nevertheless, I think this work is an important contribution to the literature, since one can read off the relationship between the different parameters and how they influence the Wasserstein distance, see also point 4 in the "Requested Changes" section.

Furthermore, some claims and formulations made in this work are not justified from my viewpoint, see the "Requested changes" section.

[1] On fine properties of mixtures with respect to concentration of measure and Sobolev type inequalities, Djalil Chafaı and Florent Malrieu, https://arxiv.org/abs/0805.0987

---

> ### Author Response · Authors · 2022-09-11
> **Response to Reviewer W6qc**
>
> Thank you for your positive comment and feedback. In what follows, we answer to the comments of the reviewer in details.
>
> * The constant  $D_0$ exceeds this bound, and the exponential terms afterwards will amplify this term if the approximation error is not of a very small magnitude. Therefore, for realistic choices of the parameters, this bound is probably larger than $\sqrt{d} + \mathrm{diam}(\mathcal{M})$.
>
> We agree with the reviewer and want to highlight two points:
> 1) The bound $D_0$ is quite loose and we don't claim this bound to be tight. For instance numerical constants and the stepsize dependency could be improved.
> 2) At initialization (i.e. for large values of $M$) the denoising diffusion model can lead to worse Wasserstein bounds than not doing anything (i.e. only considering a Gaussian random variable). This can be seen in the following example. Assume that the target measure is the unit Gaussian and that the forward process is given by the Ornstein-Uhlenbeck process, i.e. $\mathrm{d} X_t = -X_t \mathrm{d}t + \sqrt{2} \mathrm{d} B_t$. Then since the unit Gaussian is invariant for the forward process, we have that for any $t$, $p_t$ (the density of the forward process) is also the density of the unit Gaussian.The time reversal is given by $\mathrm{d} Y_t = (Y_t + 2 \nabla \log p_t(Y_t) ) \mathrm{d}t + \sqrt{2} \mathrm{d} B_t$. Of course in that case the backward process is also a Ornstein-Uhlenbeck process, i.e. $\mathrm{d} Y_t = -Y_t\mathrm{d}t + \sqrt{2} \mathrm{d} B_t$ and no approximation is needed. However, if one tries to apply the denoising diffusion model methodology then we must consider the backward diffusion $\mathrm{d} Y_t = (Y_t + 2 s_t(Y_t) ) \mathrm{d}t + \sqrt{2} \mathrm{d} B_t$, where $s_t$ is some approximation of the Stein score. However, at initialization (since the last layer of the network is not subjected to any non-linearity in practice), we have that $s_t(x) \approx 0$, which implies a large value of $M$. In that case the backward diffusion process is approximately given by  $\mathrm{d} Y_t = Y_t \mathrm{d}t + \sqrt{2} \mathrm{d} B_t$ which has an *explosive* behavior opposite to the *contractive* behavior of the Ornstein-Uhlenbeck process. This toy example illustrates that even in simple cases, a bad approximation of the score can have dramatic consequences. We have added this discussion in Appendix F.2.
>
> * The sentence directly after Theorem 1 seems misleading. It depends very much on the order in which one takes these limits. If one first takes $M \to 0$ and $\delta$  to 0, one is treating the exact case with no error and the statement is trivial. If one lets $\varepsilon \to 0$  before any of these two, the bound exponentially diverges towards infinity and the statement is not true. Therefore one only sees that the Wasserstein distance actually converges to 0 in the exact case, which is known already, since then $\mathcal{L}(Y_T)=0$.
>
> We agree with the reviewer that this sentence is misleading. This concern is shared with Reviewer Fneb and we have modified our manuscript accordingly. We have replaced the sentence by "First, we note that letting $T \to +\infty$,
>   $\delta, M \to 0$ and then $\varepsilon \to 0$ we get that
>   $W_1(\mathcal{L}(Y_{K}), \pi) \to 0$. An explicit dependency
>   between these parameters is given in Corollary 2". We refer to Corollary 2 for a quantitative analysis of the phenomenon.
>
> * In Section 3.2 generalization is treated. For that one applies the earlier Theorems to approximate the empirical distribution and then bounds the Wasserstein distance between the empirical distribution and $\pi$ by applying other results [...].
>
> We agree with the reviewer that the word "generalization" might be misleading here. By generalization, we refer here to the ability of the generative model (trained on an empirical dataset) to be close (in expectation) to the true underlying measure. Note that this definition of generalization is slighlty different from the one used in supervised discriminative models (like in [1]) as explained in the introduction of [2], see [3,4] for example of empirical studies of the generalization *in generative models*. We agree that a strong result on generalization would require to check that the samples obtained while training on the empirical dataset are close to the underlying measure and are not mere copies of the training dataset. We have added a discussion after our result asserting that our current result covers one *approximation* made during the training of the diffusion model but do not provide any result on the *innovation* of generative models. In our opinion quantifying the innovation of diffusion models (and generative models) in general is a difficult but central question that would deserve an investigation of its own and we leave it for future work.

---

> > ### Author Response · Authors · 2022-09-11
> > **Response to Reviewer W6qc (continued)**
> >
> > * In the comparison sections, the authors state that this work goes one step further than [2]. It seems to be that the results are complementary, rather then building on each other, since they neither contain nor employ the results from [2].
> >
> > By "going one step further" than [5] we meant that our results highlight that indeed diffusion models detect manifolds but also provide quantitative bounds. However, we agree that the settings and assumptions are different and that these two works can be viewed as complementary. We have modified the wording of the corresponding sentences in the revised version of the manuscript.
> >
> > * As stated in the "Weaknesses" section the most interesting results from this work is some intuition on how the different parameters influence the Wasserstein distance [...]
> >
> > We fully agree with the reasoning of the reviewer. While the bounds on $M$ ($\exp[\kappa/\varepsilon]/\varepsilon^2$) might be too loose and could likely be improved we believe that this behavior is the correct one in the sense that due to the explosive nature of the drift around time $0$, small errors can have a huge impact. Such a situation can also be seen when one tries to approximate diffusion bridges which requires a careful handling of the drift near the boundaries, see [6] and the references therein. We have added a discussion on this as well as a discussion on diffusion bridges in the revised version of the manuscript.
> >
> > * Could it be discussed how realistic the Assumptions of Theorem 3 are or when they could be expected to hold/not hold?
> >
> > In our opinion, Assumption 3 and its limitations can be discussed in two different fashions. First, the bound must hold for all $t$ and $x$, i.e. even if we allow for an explosion for small times and large spatial variables we still need the bound to hold *everywhere*. In practice this hard to check in the sense that bounds can often be obtained w.r.t. $L^2$ error as in [7]. We provide more discussion on that particular aspect of the assumption in our answer to Reviewer RSJB. The second aspect of the bound we would like to emphasize is the explosion of the approximation as time goes to $0$ and the space variable goes to $+\infty$ in norm. We highlight that an explosion of the score as the time goes to $0$ makes sense especially if we parameterize the score in an unconditional manner, see [8] for details about unconditional parameterization. A discussion on Assumption 3 has been postponed to Appendix F.1.
> >
> > * Maybe use a different letter than $\kappa$ in (3), since $\kappa$  will be used later for another quantity.
> >
> > We now use another letter ($\phi$).
> >
> > * "Our result could be extended to W_p". Maybe that could be justified a bit more [...]
> >
> > We agree that $W_1$ distances play a special role among Wasserstein distances. However, we do not use these properties (in particular we do not use that the Wasserstein distance of order one is an instance of an Integral Probability Metric) an only rely on the coupling formulation of the Wasserstein distance of order one. As such our results can be extended to other Wasserstein distances. We have clarified this in the revised version of the paper.
> >
> > * In general I think it would help the clarity and readability of the results a lot if the results could be split up in a part that explicitly bounds the distance between the continuous SDE with the true drift and the continuous SDE with the approximate drift, and then another result for the discretization, but that might also be to much work.
> >
> > Whenever possible we have tried to disentangle the results which are specific to the continuous setting and the ones which are specific to the approximation of the SDE. In particular, Section 4.1. is only concerned with the quality of the approximation of the discretized (an approximate) SDE  to the ideal SDE. Section 4.2 and Section 4.3 only deal with the continuous idealized version of the SDE.
> >
> > * In the Proof of Lemma C.2, on the first line it says [...]
> >
> > Agreed, we have changed this line according to the reviewer remark.
> >
> > [1] Zhang, Bengio, Hardt, Recht, Vinyals -- Understanding deep learning requires rethinking generalization (2017)
> >
> > [2] Gili, Mauri, Perdomo-Ortiz -- Evaluating Generalization in Classical and Quantum Generative Models (2022)
> >
> > [3] Alaa, van Breugel, Saveliev, van der Schaar -- How Faithful is your Synthetic Data? Sample-level Metrics for Evaluating and Auditing Generative Models (2022)
> >
> > [4] Zhao, Ren, Yuan, Song, Goodman, Ermon -- Bias and Generalization in Deep Generative Models:
> > An Empirical Study (2018)
> >
> > [5] Pidstrigach -- Score-Based Generative Models Detect Manifolds (2022)
> >
> > [6] De Bortoli, Doucet, Heng, Thornton -- Simulating Diffusion Bridges with Score Matching (2021)
> >
> > [7] Lee, Lu, Tan -- Convergence for score-based generative modeling with polynomial complexity (2022)
> >
> > [8] Jolicoeur-Martineau, Piche-Taillefer, Mitliagkas, Tachet des Combes -- Adversarial score matching and improved sampling for image generation (2020)

---

> > > ### Comment · Reviewer_W6qc · 2022-09-15
> > > **Answer to the Authors**
> > >
> > > Thank you for the detailed response to my review and the updates made to the paper. These are the remaining remarks:
> > > - I still do not understand the first sentence after Theorem 1. To my understanding, taking the limits in this order only means that the reverse SDE recovers the target distribution in the exact case. This is a fundamental property of reverse SDEs. This is a known result, and therefore the purpose of this sentence after the Theorem is not clear to me.
> > > - About the _vacuous_ bounds: (Second paragraph after Corollary 2). The bounds are only vacuous if they are large in TV distance. If they are small in TV distance, they are actually the exact opposite. Contrary to the Wasserstein-distance, they do actually imply that the DDM algorithm outputs samples that are from the support of the target measure $\pi$, which is a very strong statement under the manifold hypothesis. In the KL-Divergence, any finite bound actually implies the above, and only an infinite KL-Divergence bound would be _vacuous_ (which it would be under any circumstances).
> > > - I guess this is just a mistake, but it still says "In this work, we go one step further"

---

> > > > ### Author Response · Authors · 2022-09-15
> > > > **Response to "Answer to Authors"**
> > > >
> > > > Thank you for your answer. We try to answer the reviewer comments in what follows:
> > > >
> > > > * We apologize for the confusion created by that sentence. There is no claim of novelty here. We totally agree with the reviewer that the backward SDE (with ideal drift) does converge to the original distribution (by definition of the backward SDE). This sentence was merely a sanity check intended to signal to the reader that we indeed recover this (expected) behavior. Of course the real contribution lies into the quantitative bounds (which are studied in Theorem 1 and Corollary 2). If the reviewer thinks that this sentence is not clear enough we can either highlight that this type of behavior is to be expected or remove that sentence altogether.
> > > >
> > > > * We agree that small TV bounds and finite KL bounds are immediately non-vacuous as they would signal a very strong property, i.e. the generative model is absolutely continuous w.r.t. the target distribution. However, we argue that this situation is unlikely as we add Gaussian noise at every step of the denoising diffusion model (including the last one). Hence, there is a small probability that the samples will not fall on the manifold rendering bounds in TV and KL vacuous not matter how small this probability (this can be observed in our toy examples in Figure 2 for instance where not all the samples fall on the manifold). However this is not the case with the Wasserstein distance which still yields meaningful bounds in that setting. If the reviewer agrees with this explanation we will modify the sentence accordingly to highlight that: 1) small TV bounds and finite KL bounds are non-vacuous and very informative 2) in practice the requirement to obtain small TV bounds and finite KL bounds is too strong as it requires to sample exactly on the manifold.
> > > >
> > > > * We apologize for the omission (we corrected this in the "Related work" section but not in the "Introduction") and have updated the paper accordingly.

---

> > > > > ### Comment · Reviewer_W6qc · 2022-09-16
> > > > > **Answer**
> > > > >
> > > > > Thank you for the answer. For both of the raised points, any of the suggested changes would resolve my concerns.

---

> > > > > > ### Author Response · Authors · 2022-09-16
> > > > > > **Answer**
> > > > > >
> > > > > > Thanks again for your very quick answer.
> > > > > > We have added two sentences to address both points in the revised version of the manuscript.

---

### Author Response · Authors · 2022-09-11
**General comment**

We thank the reviewers for their insightful comments and are encouraged by their positive feedback regarding the proposition and soudness of our work. We have uploaded a revised version of the paper with changes highlighted in red.

---

### Decision · Action_Editors · 2022-10-12

**Recommendation:** Accept as is

**Comment:**

Three expert reviewers think that the paper has solid contributions, which I agree with. The author have addressed a few minor concerns, and several claims have been properly worded, which improved the quality of the paper even further.

AE decision is "accept".


**Audience:**

This paper is of particular interest to machine learning theory community.

**Claims And Evidence:**

This is a theoretical work. The claims have been justified with rigorous analysis.